# Robustness in Text-Attributed Graph Learning: Insights, Trade-offs, and New Defenses

**Runlin Lei**[1]    **Lu Yi**[1]    **Mingguo He**[2]    **Pengyu Qiu**[3]
**Zhewei Wei**[1*]    **Yongchao Liu**[3*]    **Chuntao Hong**[3]

[1]Renmin University of China    [2]National University of Singapore    [3]Ant Group

`{runlin_lei, yilu, zhewei}@ruc.edu.cn, mingguo@nus.edu.sg`
`{pengyu.qpy, yongchao.ly, chuntao.hct}@antgroup.com`

## Abstract

While Graph Neural Networks (GNNs) and Large Language Models (LLMs) are powerful approaches for learning on Text-Attributed Graphs (TAGs), a comprehensive understanding of their robustness remains elusive. Current evaluations are fragmented, failing to systematically investigate the distinct effects of textual and structural perturbations across diverse models and attack scenarios. To address these limitations, we introduce a unified and comprehensive framework to evaluate robustness in TAG learning. Our framework evaluates classical GNNs, robust GNNs (RGNNs), and GraphLLMs across ten datasets from four domains, under diverse text-based, structure-based, and hybrid perturbations in both poisoning and evasion scenarios. Our extensive analysis reveals multiple findings, among which three are particularly noteworthy: 1) models have inherent robustness trade-offs between text and structure, 2) the performance of GNNs and RGNNs depends heavily on the text encoder and attack type, and 3) GraphLLMs are particularly vulnerable to training data corruption. To overcome the identified trade-offs, we introduce SFT-auto, a novel framework that delivers superior and balanced robustness against both textual and structural attacks within a single model. Our work establishes a foundation for future research on TAG security and offers practical solutions for robust TAG learning in adversarial environments. Our code is available at: `https://github.com/Leirunlin/TGRB`.

## 1 Introduction

Text-attributed graphs (TAGs), which integrate structural links with rich text features, are foundational to applications from social networks to citation graphs (Wu et al., 2023; Wang et al., 2025a). While Graph Neural Networks (GNNs) have long been the prevailing approach for learning on TAGs, Graph Large Language Models (GraphLLMs) are emerging as a compelling paradigm, leveraging their advanced reasoning capabilities directly on graph-structured text. However, the robustness of these models remains a critical challenge. For instance, in high-stakes domains such as social and financial networks, adversaries can manipulate both graph structures and textual content, significantly degrading model performance. For example, adversaries can deploy deceptive social bots with engineered biographies and network patterns to influence public opinion (Wang et al., 2023). Similarly, in recommendation systems, attackers may craft fake user profiles with misleading textual attributes to promote targeted items (Nguyen et al., 2024). This dual vulnerability makes it uniquely difficult to secure TAG learning.

Despite its importance, existing robustness analyses remain fragmented. Early analyses of GNNs and Robust GNNs (RGNNs) relied on naive embeddings, largely overlooking the rich semantic information in natural language (Zheng et al., 2021). Conversely, recent attempts start explorations of the robustness of GraphLLMs, yet lack comprehensive comparisons among model families and

---

*Zhewei Wei and Yongchao Liu are the Corresponding authors.

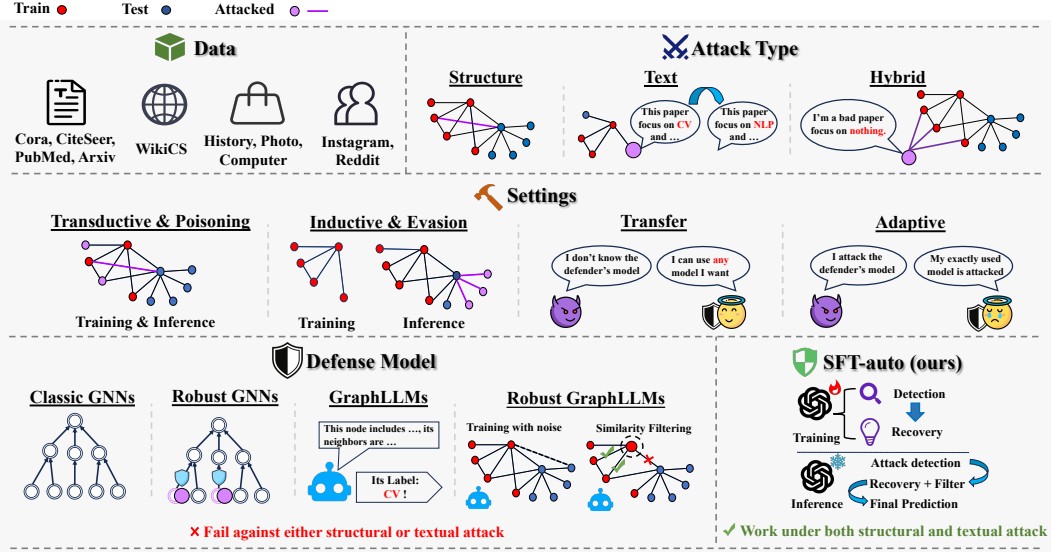

Figure 1: The overall framework for evaluating the robustness of TAG learning.

focus exclusively on limited attack settings (Guo et al., 2024a; Zhang et al., 2025a). This fragmented landscape has left the field without a comprehensive analysis of robust TAG learning.

To address this critical void, we perform a large-scale, systematic robustness analysis on TAGs. The overall framework is provided in Figure 1. Our evaluation spans ten datasets across four domains and a wide spectrum of models, including classical GNNs, RGNNs, and GraphLLMs. By subjecting these models to a unified threat model, we provide a unified evaluation methodology that serves as a foundation for understanding model vulnerabilities across diverse architectures and attack settings.

The extensive evaluation yields a series of empirical insights: 1) We uncover a crucial text-structure robustness trade-off, where models excel at defending against either textual or structural attacks but not both simultaneously. 2) We find that previously underrated methods, such as GNNGuard (Zhang & Zitnik, 2020), achieve surprising performance when re-evaluated in TAG settings with advanced text encoders. 3) GraphLLMs demonstrate higher vulnerability to poisoning attacks compared to GNNs. Specifically, when the training data is compromised, GraphLLMs experience a more significant decline in performance than GNNs. 4) Directly integrating existing robust GNN designs into LLM architectures fails to resolve the fundamental robustness trade-off.

Motivated by the proven effectiveness of noise-injection (Ennadir et al., 2024) and similarity-filtering strategies (Zhang & Zitnik, 2020; Hou et al., 2024) in RGNNs, we explore their adaptation to GraphLLMs to address the text-structure trade-off. While these variants show effectiveness against either textual or structural perturbations individually, they still struggle to achieve balanced robustness against both types of attacks. To achieve balanced robustness against both attacks, we propose a novel SFT (supervised fine-tuning) framework, SFT-auto, which employs multi-task training with a detection-prediction pipeline. This approach leverages the superior reasoning capabilities of LLMs to detect anomalies and make predictions within a single model. Our experiment results show that SFT-auto exhibits superior robustness in both modalities compared to the baselines.

To summarize, our **main contributions** are as follows:

- **A Comprehensive Evaluation Framework.** We propose a systematic robustness evaluation for learning in TAGs that benchmarks a wide spectrum of models, from classical GNNs, RGNNs, to GraphLLMs, against a diverse set of textual and structural attacks.

- **Abundant Empirical Insights.** Our large-scale analysis reveals critical vulnerabilities and trade-offs in robust TAG learning. We uncover a **text-structure robustness trade-off**, find that simple RGNNs with advanced text encoders can be surprisingly effective, and demonstrate the vulnerability of GraphLLMs to poisoning attacks.

- **An Effective Defense Framework.** To overcome the identified trade-off, we propose the SFT-auto model. It leverages the reasoning capabilities of LLMs to achieve superior and balanced robustness against both textual and structural attacks.

## 2 PRELIMINARIES: BACKGROUND AND EVALUATION PROTOCOL

### 2.1 BACKGROUND

We define a TAG as $\mathcal{G} = (\mathcal{V}, \mathcal{E}, \{s_i\}_{i=1}^N)$, where $\mathcal{V}$ is the set of nodes, $\mathcal{E}$ is the set of edges, and $N$ is the number of nodes. The adjacency matrix is $\mathbf{A} \in \{0,1\}^{N \times N}$, with each node $v_i \in \mathcal{V}$ associated with a sequential text $s_i$. Following prior work on graph adversarial learning (Jin et al., 2021; Zheng et al., 2021), we focus on the task of node classification, where the goal is to assign each node one of $C$ possible class labels. Denote the label vector as $\boldsymbol{y} \in \{0, \ldots, C-1\}^N$, the learning objective is to predict labels of target nodes $\boldsymbol{y}_{\text{target}}$, given $\mathcal{G}$ and ground-truth labels $\boldsymbol{y}_{\text{train}}$. In the **transductive setting**, all nodes are observed during training, while in the **inductive setting**, the model is required to generalize to previously unseen test nodes. Existing methods for TAG node classification include **GNNs** and **GraphLLMs**, denoted as a model $f(\{s_i\}_{i=1}^N, \mathbf{A})$. For GNNs, the model $f$ first employs a text encoder to transform each $s_i$ into a node-level feature matrix $\mathbf{X}$, and then processed jointly with the adjacency matrix $\mathbf{A}$. In contrast, some of the GraphLLMs directly input the raw node texts $\{s_i\}_{i=1}^N$ into the model $f$ and perform classification via prompt instructions.

**Graph Adversarial Attacks and Defenses.** In adversarial settings, an attacker seeks to degrade the performance of the defender's model on a target set of nodes $\mathcal{V}_{\text{target}}$. A typical graph attack is the Graph Modification Attack (GMA) (Zügner & Günnemann, 2019; Xu et al., 2019), which perturbs either the graph structure or the node texts. The objective of GMA is:

$$\min_{\mathbf{A}', \{s_i'\}_{i=1}^N} \text{Acc}\big(f_\theta(\{s_i'\}_{i=1}^N, \mathbf{A}'), \boldsymbol{y}_{\text{target}}\big), \text{ s.t.} \|\mathbf{A}' - \mathbf{A}\|_0 \leq 2\Delta_{\text{struct}} \text{ and } \sum_{i=1}^N \mathbf{1}\{s_i' \neq s_i\} \leq \Delta_{\text{text}},$$

where $\mathbf{A}'$ denotes the perturbed adjacency matrix, $\{s_i'\}_{i=1}^N$ represents the perturbed node texts, $\text{Acc}(\cdot)$ is the evaluation metric (e.g., accuracy) on the target nodes $\mathcal{V}_{\text{target}}$, and $\Delta_{\text{struct}}$ and $\Delta_{\text{text}}$ are the budget on the total number of perturbations. Specifically, $\|\mathbf{A}' - \mathbf{A}\|_0$ measures the number of edge modifications (additions or deletions), and $\sum_{i=1}^N \mathbf{1}\{s_i' \neq s_i\}$ counts the number of nodes with modified texts. Besides GMA, other paradigms targeting TAGs include Text-level Graph Injection Attacks (**Text-GIAs**), where new adversarial textual nodes are introduced into the graph, forming harmful connections to existing nodes (Lei et al., 2024).

Attacks can be categorized by their timing: **poisoning attacks** modify the training data to compromise the learned model, while **evasion attacks** alter test inputs to fool a fixed model at inference time. For the defender, the key objective is to maintain high performance even when the data may be under attack. Efforts have been made via RGNN design (Jin et al., 2021) and LLM as graph data purifiers (Zhang et al., 2025b).

Table 1: Comparisons between evaluations of robustness in TAG learning.

| Benchmark | Data | | Baselines | | | Evaluation Settings | | | |
|---|---|---|---|---|---|---|---|---|---|
| | Num. Datasets | Num. Domains | GNNs | RGNNs | GraphLLMs | Attack Types | Settings | Encoder Analysis | Adaptive Attack |
| GRB (Zheng et al., 2021) | 5 | 2 | ✓ | ✓ | × | GIA | Evasion & Inductive | × | × |
| Gosch et al. (Gosch et al., 2023) | 8 | 4 | ✓ | ✓ | × | GMA | Evasion & Both | × | ✓ |
| Guo et al. (Guo et al., 2024a) | 6 | 3 | ✓ | × | ✓ | GMA+Text | Evasion & Transductive | ✓ | × |
| TrustGLM (Zhang et al., 2025a) | 6 | 2 | × | × | ✓ | GMA+Text+Prompt | Evasion & Transductive | × | × |
| Olatunji et al. (2025) | 4 | 1 | × | × | ✓ | GMA+Text | Both & Transductive | × | × |
| **Ours** | **10** | **4** | ✓ | ✓ | ✓ | **GMA+Text+Text-GIA** | **All** | ✓ | ✓ |

### 2.2 EVALUATION PROTOCOL

Extensive evaluations of robustness in TAG learning have been developed. However, their analyses are limited, particularly with respect to data, baselines, and evaluation settings. As shown in Table 1, in terms of **data**, previous works suffer from limited dataset diversity and narrow domain

coverage, with some focusing exclusively on specific graph types or application domains. Regarding **baselines**, no previous work has achieved comprehensive integration of all three major graph learning paradigms (GNNs, RGNNs, and GraphLLMs), resulting in fragmented evaluation landscapes. Finally, **evaluation settings** in existing evaluations are often constrained by limited attack diversity.

To address these limitations, we include: (1) an extensive dataset diversity spanning multiple domains and graph types; (2) a unified comparison supporting major baseline categories (GNNs, RGNNs, GraphLLMs); and (3) extensive evaluation settings with comprehensive metrics. The detailed design of the evaluation follows the principles below:

- **Deploy sufficiently strong attacks.** We found that some attacks, such as Mettack (Zügner & Günnemann, 2019) and word-level textual attacks, don't generate sufficiently strong attacks or have poor transferability. Yet, weak perturbations fail to differentiate defense models, as performance rankings become dominated by clean accuracy rather than adversarial resilience (see Appendix D.4). Therefore, we employ more effective attacks with a sufficiently high perturbation ratio in the main paper to ensure a higher degree of differentiation. Results with smaller ratios are deferred to Appendix J.4.

- **Ensure fair baseline comparison.** We restrict evaluation to models with comparable clean performance to prevent stronger backbones from appearing artificially robust. Methods like Instruct-GLM, GPT zero-shot, and GraphPrompt in (Guo et al., 2024a; Zhang et al., 2025a; Olatunji et al., 2025) significantly underperform supervised baselines, invalidating robustness comparisons. Following established practices (Hou et al., 2024; Wu et al., 2025; Wang et al., 2025b), we select competitive GNNs, RGNNs, and GraphLLMs as baselines. The baselines are summarized in Table 2. The details of each baseline are provided in Appendix E. [1]

- **Adopt realistic evaluation protocols.** Prior benchmarks employ misaligned settings that compromise validity. As stated in (Gosch et al., 2023), poisoning attacks naturally pair with transductive learning, while evasion attacks suit inductive evaluation. Protocol misalignments enable trivial memorization-based defenses, undermining meaningful assessment. We strictly align attack types with appropriate learning paradigms across all experimental phases.

**Data.** We evaluate on ten datasets spanning four distinct domains following the LLMNodeBed benchmark (Wu et al., 2025): academic networks (Cora (Sen et al., 2008), CiteSeer (Giles et al., 1998), PubMed (Yang et al., 2016), ArXiv (Hu et al., 2020)), web links (WikiCS (Mernyei & Cangea, 2020)), social networks (Instagram, Reddit (Huang et al., 2024)), and e-commerce (History, Photo, Computer (Yan et al., 2023)). We adopt a supervised 60/20/20 split across training/validation/testing for the *inductive* setting and a semi-supervised 10/10/80 split for the *transductive* setting. All datasets are undirected graphs. Details of datasets are provided in Appendix C.

**Threat Model.** Our evaluation assesses perturbations to the graph structure, node texts, and also includes results against Text-GIAs (in Appendix F). We evaluate both poisoning attacks and evasion attacks. Our primary focus is on **transfer attacks** where the attacker has access to the victim's data but not their model directly. The perturbed graph is then transferred to test the defender's model, simulating a practical scenario where defenders can deploy their custom defense model. The specific attack configurations are detailed in the subsequent experimental sections, with comprehensive details provided in Appendix D. We also explore adaptive attacks in Appendix H.

**Other Setups.** We employ accuracy as the evaluation metric. Hyperparameters for all GNNs and RGNNs are optimized based on validation set performance. Following LLMNodeBed (Wu et al., 2025), we adopt RoBERTa (Liu et al., 2019) as the text encoder for GNN-based methods and Mistral-7B (Jiang et al., 2023) as the backbone for GraphLLMs, as these configurations yield optimal performance. All experiments are conducted across three independent runs with random data splits, except for ArXiv, which uses a single official split.

## 3 EVALUATION RESULTS

In this section, we present evaluation results against structural and textual attacks. Due to space limitations, we report the average rank, which is derived for each method by averaging its ranks across

---

[1]We move the evaluation of specific variants like GOOD-AT, GPR-GAE, and GPRGNN-AT to Appendix K. These supplementary results serve as an extended verification of our core findings.

Table 2: Categorization of selected defense models.

| Taxonomy | Subcategory | Selected Defenses / Models |
|---|---|---|
| Basic models | Spatial / Message passing Spectral | GCN (Kipf & Welling, 2017), GAT (Velickovic et al., 2018) APPNP (Klicpera et al., 2019), GPRGNN (Chien et al., 2021) |
| Improving training | Robust training | GRAND (Feng et al., 2020), NoisyGCN (Ennadir et al., 2024), Adversarial Training Gosch et al. (2023) |
| Improving architecture | Probabilistic Similarity-based Robust aggregation | RobustGCN (Zhu et al., 2019) GNNGuard (Zhang & Zitnik, 2020) ElasticGNN (Liu et al., 2021), SoftmedianGDC (Geisler et al., 2021), RUNG (Hou et al., 2024) |
| | Others | EvenNet (Lei et al., 2022) (Spectral), GCORN (Abbahaddou et al., 2024) (Weight Regularization), |
| Improving structure | Unsupervised | Jaccard-GCN (Wu et al., 2019), Cosine-GCN (Mujkanovic et al., 2022) |
| | Supervised | ProGNN (Jin et al., 2020), Stable (Li et al., 2022b), GOOD-AT Li et al. (2024), GPR-GAE Lee & Park (2025) |
| GraphLLMs | Instruction Tuning & Align | GraphGPT (Tang et al., 2024), SFT (w/ nei.) (Wang et al., 2025b), LLaGA (Chen et al., 2024) |

all datasets where it has valid results, with full numerical results available in Appendix J. We adopt rank-based evaluation because averaging raw accuracies across datasets can be misleading due to scale differences, missing results on large-scale graphs, and strong dependence on clean accuracy. Ranking provides a normalized, dataset-agnostic comparison that offers a clearer comparison. Similarly, the results about hybrid and adaptive attacks are deferred to Appendix F and H.

For structural attacks, in the inductive and evasion settings, we employ PGD (Xu et al., 2019) for small-scale datasets and GRBCD (Geisler et al., 2021) for larger ones with a perturbation ratio of 0.20. We use GCN (Kipf & Welling, 2017) as the surrogate model with BoW embeddings to generate victim graphs, and use the generated graphs as the test victim graphs. For *transductive* structural attacks, we adopt HeuristicAttack (Li et al., 2023) with a perturbation ratio of 0.30, and exclude Computer and ArXiv due to scalability issues.

We evaluate performance against textual attacks using a novel LLM-based attack. For evasion attacks, we substitute 40% of the test set nodes with LLM-generated text that differs from the original content. For poisoning attacks, we replace 80% of the training set nodes. Attack algorithm details are provided in Section D. Due to scalability concerns, we exclude the Computer and ArXiv datasets for evasion attacks, and additionally exclude the Photo dataset for poisoning attacks. We do not use gradient-based or unnoticeable word-level textual attacks, as these methods have been shown to have poor transferability across different models, as discussed in Appendix D.4.

## 3.1 AGAINST STRUCTURAL ATTACKS

The results against structural attacks are shown in Figure 2.

**GraphLLMs show Inherent Robustness against Non-adaptive Inductive/Evasion Attacks.** Even without defense mechanisms, SFT-neighbor outperforms most RGNNs. GraphGPT, which also employs instruction tuning, exhibits comparable strong performance, reflecting GraphLLM's superior robustness against structural attacks. Notably, LLaGA shows relatively weaker robustness despite being a GraphLLM. Although LLaGA surpasses GraphGPT in clean perfor-

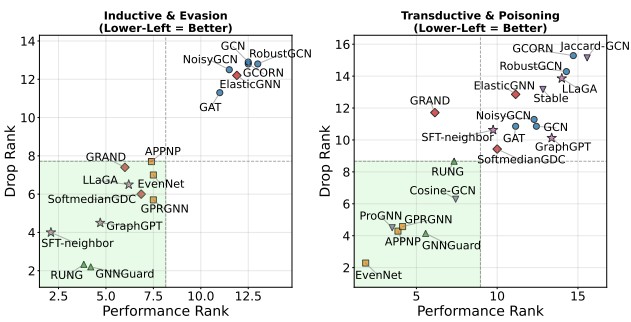

Figure 2: Comparison of robustness against structural attacks. Ranks are averaged across datasets (excluding failures) based on: 1) absolute accuracy under attack (lower rank is better), and 2) relative accuracy drop from the clean baseline (lower drop rank is better).

mance, it proves more vulnerable to structural attacks. This suggests that alignment-based methods, which explicitly utilize graph embeddings to align graph and text spaces, are more susceptible to structural attacks (similar to GCNs), while instruction-tuning models adopt more conservative neighbor utilization strategies.

**Simple Methods Can Shine in TAGs through Advanced Text Encoders.** Despite their simplicity, RGNNs from earlier research can deliver surprisingly strong performance when re-evaluated in TAGs. For instance, GNNGuard, as an early and straightforward RGNN that leverages threshold-based filtering to defend against adversarial attacks, achieves top-tier performance against the inductive/evasion attacks. This contrast stems from prior works that have overlooked the importance of text embeddings and have only evaluated RGNNs on shallow embeddings, such as BoW or TF-IDF (Lei et al., 2022; Hou et al., 2024). In the context of TAGs, methods like GNNGuard can be revitalized to achieve near-SOTA robustness. A detailed analysis is provided in Appendix G.

In fact, by fully harnessing textual features through dataset- and embedding-specific filtering, we can do even better. In Appendix G.3, we propose **Guardual**, a novel extension that eliminates the reliance of GNNGuard on threshold hyperparameters. The results show that Guardual's adaptive filtering mechanism makes it the leading RGNN in the structural evasion setting. These findings underscore that strategic text processing in TAGs fundamentally drives RGNN performance.

**Spectral GNNs Show Superior Robustness against Poisoning Structural Attacks.** As shown in Figure 2 (right), spectral methods, such as EvenNet, APPNP, and GPRGNN, demonstrate superior performance against poisoning attacks in the transductive setting. This phenomenon complements the findings of Gosch et al. (2023) in the evasion/transductive setting, where the robust diffusion process in spectral methods enables flexible use of whole higher-order neighborhoods, thereby enhancing robustness. Structure learning methods, such as ProGNN, also exhibit promising results, though their computational overhead remains a significant practical limitation. In contrast, the performance of GraphLLMs starts to decline. While SFT-neighbor is robust against evasion attacks, the perturbations introduced during training result in a notable performance drop. Among GraphLLMs, LLaGA remains the most vulnerable due to its greater reliance on structure.

## 3.2 AGAINST TEXTUAL ATTACKS

The results against textual attacks are shown in Figure 3.

**Simple GNNs Excel Advanced Baselines.** When confronting textual attacks in the evasion setting, the relative robustness rankings undergo a notable shift. While methods like GNNGuard and RUNG demonstrate superior performance against structural attacks, as shown in Figure 2, they exhibit pronounced vulnerabilities to textual perturbations. SFT-neighbor and GraphGPT also suffer from significant performance degradation. In contrast, even naive models, such as GCN and GAT, exhibit the desired robustness against these textual attacks.

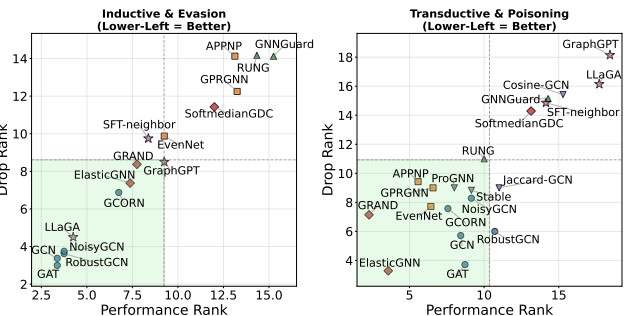

Figure 3: Comparison of robustness against textual attacks. Ranks are averaged across datasets (excluding failures) based on: 1) absolute accuracy under attack (lower rank is better), and 2) relative accuracy drop from the clean baseline (lower drop rank is better).

**GraphLLMs Struggle Against Textual Poisoning.** For poisoning attacks, both GNNs and RGNNs exhibit remarkable robustness. Even when 80% of the training nodes' text is replaced, GNNs can still benefit from the transductive learning paradigm and achieve accurate predictions by aggregating information from nodes' neighbors. Under text poisoning attacks, LLM-based methods experience a significant decline in performance. For example, on the `CiteSeer` dataset, SFT-neighbor's accuracy drops by 25%, while most GNNs suffer only a 5%-10% decrease (see Tables 39 and 45 for detailed accuracy). This suggests that GraphLLMs are more vulnerable to perturbations in the training set, relying heavily on high-quality training text to maintain strong performance.

## 4 THE TEXT-STRUCTURE TRADE-OFF

In Section 3, we found that in the inductive/evasion setting, the structurally robust models fail against textual attacks (e.g., GNNGuard, SFT-neighbor); and vice versa (e.g., NoisyGCN). Building on this, we now explore how vulnerabilities in one dimension relate to the other.

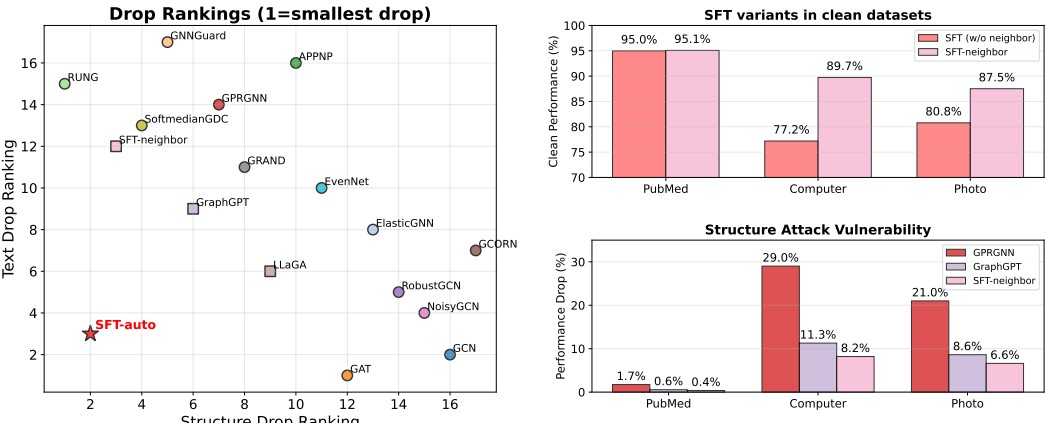

Figure 4: The text-structure robustness trade-off. **(Left)** SFT-auto uniquely balances this trade-off, while baseline models are polarized towards either text or structure robustness. **(Right)** On **text-friendly** datasets (e.g., PubMed), models are less reliant on structure and thus less vulnerable to its perturbation. The opposite holds for **structure-critical** datasets (e.g., Computer, Photo).

**Architecture Effect.** As shown in Figure 4 **(Left)**, models with different architectures exhibit a clear text-structure robustness trade-off. Structure-oriented architectures such as LLaGA and vanilla GNNs are highly vulnerable to structural perturbations yet remain comparatively stable under textual attacks. By contrast, text-oriented models like SFT-neighbor and GraphGPT, as well as RGNNs designed to enhance structural robustness, show strong resistance to structural attacks but collapse under textual ones. These comparisons highlight the inherent bias of different backbones: classifiers are generally robust to either structure or text perturbations, but rarely to both.

**Dataset Effect.** Dataset characteristics critically shape the text-structure trade-off. As shown in Figure 4 **(Right)**, the benefit of neighbor information differs markedly across datasets, directly impacting vulnerability. In the **top-right** panel, SFT variants gain substantial accuracy from neighbors on structure-critical datasets such as `Computer`/`Photo`, but little on text-friendly datasets like `PubMed`. This reliance pattern explains the robustness outcomes in the **bottom-right** panel: models relatively resistant to structural perturbations (e.g., GPRGNN, GraphGPT, SFT-neighbor) experience only minor drops on text-friendly datasets, yet still suffer significantly on structure-critical ones. Thus, while the trade-off is influenced by model design, its manifestation heavily depends on the nature of the datasets.

## 5 ADDRESSING THE TEXT-STRUCTURE ROBUSTNESS TRADE-OFF

The text-structure trade-off remains a key challenge, with no current model effectively balancing both aspects. In this section, we explore methods to overcome this limitation.

### 5.1 ATTEMPTS BY BUILDING ROBUST GRAPHLLMS

To address the text-structure trade-off, we explore noise-injection and similarity-filtering strategies inspired by existing RGNNs, adapting them for GraphLLMs.

**Noise-Injection Methods.** Drawing inspiration from GRAND (Feng et al., 2020) and NoisyGCN (Ennadir et al., 2024), we inject targeted perturbations during training to reduce the distribution gap between training and adversarial test conditions. We implement three variants: `-noise` (structural noise injection), `-noisetxt` (textual noise injection), and `-noisefull`

(combined noise types). All methods use a 10% noise ratio. GraphGPT-noisetxt and GraphGPT-noisefull are neglected due to poor clean performance.

Figure 5 demonstrates that targeted noise injection provides defense against corresponding attack types. The `-noise` variant helps improve structure robustness, while `-noisetxt` helps improve textual robustness. However, when the attack type is unknown, significant trade-offs emerge. Notably, `-noisefull` fails to achieve simultaneous defense against both attack types. None of the variants shows better results against both attacks. This limitation restricts noise-injection methods to specialized defenses for specific types of attacks.

**Similarity-Filtering Methods.** Inspired by the similarity-filtering strategy of GNNGuard, we propose two variants: `-simf`, which employs edge filtering similar to GNNGuard, and `-simp`, which modifies the SFT prompt to guide adaptive reliance—leveraging neighborhood information when text is unreliable and depending on textual content when structure is compromised. Figure 5 reveals that `-simf` exhibits behavior analogous to GNNGuard, providing effective defense against structural attacks but remaining vulnerable to text-based perturbations. In contrast, `-simp` yields modest robustness improvements without significant performance gains. This suggests that simple instruction modifications are insufficient for LLMs to learn effective defenses that break the identified trade-off and learn effective defenses.

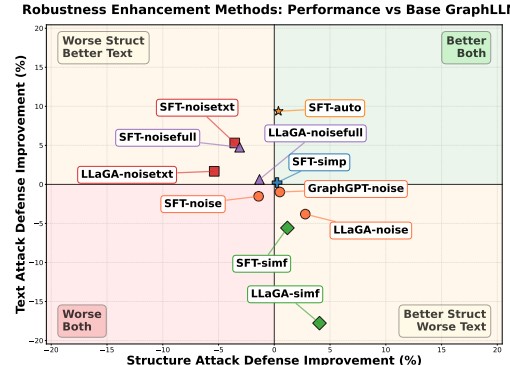

Figure 5: Performance of robust variants against base GraphLLMs. Points represent improvements over base models against structural (x-axis) and textual (y-axis) evasion attacks.

## 5.2 AUTO FRAMEWORK FOR ADVERSARIAL ATTACK DETECTION AND RECOVERY

The preliminary attempts highlight the fundamental challenge of achieving balanced robustness against both types of attacks. As a solution, we propose `SFT-auto`, a novel model that leverages the reasoning abilities of LLMs to defend against both textual and structural adversarial attacks.

**Training.** The training phase employs a principled data augmentation strategy to endow the model with attack recognition and recovery capabilities. We use an adaptive attack ratio $r = \min(1/|\mathcal{C}|, 0.15)$ to ensure balanced detection across datasets with varying class distributions. Training data comprises three distinct types: **Normal samples** ($\mathcal{S}_{\text{normal}}$) preserve original node-neighbor pairs to maintain standard classification ability; **Attack samples** ($\mathcal{S}_{\text{attack}}$) contain nodes with text deliberately replaced by content from different-class nodes, labeled as "text_attacked" to teach attack recognition; and **Recovery samples** ($\mathcal{S}_{\text{recovery}}$) remove center text entirely, compelling the model to leverage neighbor information for robust prediction. This training paradigm enables the LLM to handle $(|\mathcal{C}| + 1)$-class attack detection and $|\mathcal{C}|$-class recovery tasks through specialized prompts.

**Inference.** The inference phase implements a three-stage adaptive pipeline that dynamically responds to detected attack patterns. **Stage 1: Attack Detection:** The LLM identifies text-attacked nodes through an extended $(|\mathcal{C}| + 1)$-dimensional classification space, while structure attacks are detected via embedding-based similarity analysis. Nodes exhibiting low cosine similarity ($< 0.5$) to over half their neighbors are flagged as structure-attacked. Text attack detection takes precedence to prevent redundant dual flagging. **Stage 2: Adaptive Recovery:** Text-attacked nodes bypass their corrupted center text entirely, relying solely on original neighbor information for classification. Structure-attacked nodes leverage their preserved own text combined with filtered neighbors. Connections to text-attacked nodes or those with low similarity are removed. Normal nodes employ standard classification using their original text and neighbors, with only text-attacked neighbors filtered. The detailed pseudo-code of SFT-auto is given in Algorithm 1.

**Complexity Analysis.** The computational cost of SFT-auto is comparable to SFT-neighbor, as both methods are bottlenecked by the per-sample forward pass through the LLM. Training requires at most $1.3\times$ more samples due to data augmentation (with ratio $r \leq 0.15$). For inference, let $\mathcal{T}_{\text{LLM}}$ be the time for a single forward pass. The average per-sample inference time for SFT-auto is $\mathcal{T}_{\text{avg}} \approx$

$(1 + p_{\text{attack}}) \cdot \mathcal{T}_{\text{LLM}}$, where $p_{\text{attack}}$ is the small fraction of detected nodes requiring recovery. The value of $p_{\text{attack}}$ is bounded above by 2, which incurs an acceptable worst-case overhead, and is typically very small in practice. Given the equivalent per-sample cost, SFT-auto's overall runtime is comparable to its baseline, making it an efficient framework for achieving balanced robustness.

**Results.** SFT-auto demonstrates superior performance, as visualized in Figure 4. Compared to baselines, SFT-auto has more consistent performance against both structural and textual attacks.

## 5.3 GNN with Auto Design

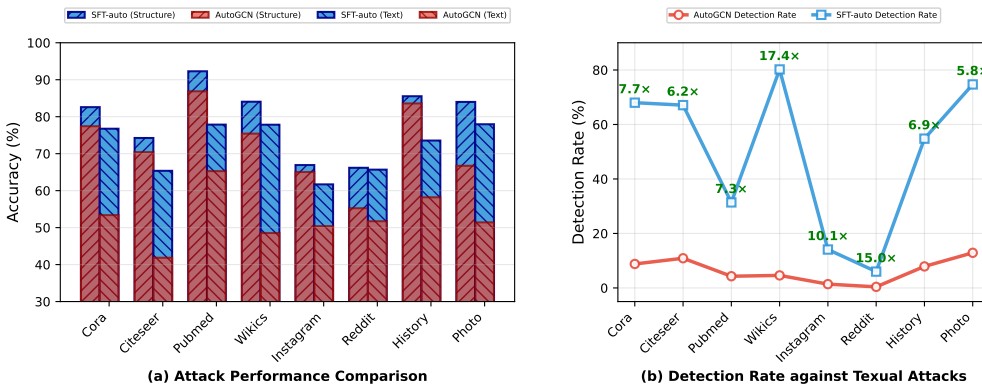

(a) Attack Performance Comparison    (b) Detection Rate against Texual Attacks

Figure 6: Performance comparison between SFT-auto and AutoGCN. **(a)** Accuracy under structural (left bars) and textual (right bars) attacks. **(b)** Detection efficacy against textual attacks.

Having established SFT-Auto's effectiveness, a natural question arises: *Can similar auto-detection principles enhance the robustness of GNNs?* To explore this, we implement AutoGCN by replacing the LLM predictor with a GCN, while maintaining the same detection pipeline architecture.

Figure 6 reveals that AutoGCN exhibits substantial degradation across both attack modalities, with particularly pronounced deficiencies in textual anomaly detection (Figure 6b), where SFT-Auto achieves 6.2–17.4× improvements over AutoGCN. This performance gap illuminates fundamental architectural differences: LLMs possess inherent multi-modal reasoning capabilities, enabling seamless integration of detection and classification within unified frameworks. Conversely, GNNs lack the linguistic sophistication required for text anomaly detection. This suggests promising research directions toward hybrid architectures that combine GNNs' structural robustness with LLMs' semantic understanding through multi-stage prediction pipelines built on verified, clean data.

## 6 Related Work

**Robust GNNs and GraphLLMs.** The vulnerability of GNNs to adversarial attacks has been extensively studied (Jin et al., 2021; Xue et al., 2025), motivating a series of RGNNs. Robust training methods modify the learning process to enhance resilience, exemplified by GRAND (Feng et al., 2020) and NoisyGCN (Ennadir et al., 2024). Robust architectural designs introduce inherently stable mechanisms, such as GNNGuard (Zhang & Zitnik, 2020), which filters edges based on node similarity, SoftMedian-GDC (Geisler et al., 2021), which applies median-based aggregation, and RUNG (Hou et al., 2024), which adopts an unbiased aggregator for improved soft filtering. Graph structure learning methods, including ProGNN (Jin et al., 2020) and Stable (Li et al., 2022b), refine the graph topology to denoise adversarial input. However, these RGNNs rely solely on shallow embeddings, neglecting the influence of raw textual information in TAGs on robustness, and focus primarily on defending against structural perturbations.

Recently, LLMs have been introduced for robust TAG learning. Representative methods include GraphEdit (Guo et al., 2024b), RLLMGNN (Zhang et al., 2025b), and LangGSL (Su et al., 2024), which leverage LLMs to adjust or reconstruct graph structure under adversarial settings. However, these approaches use LLMs solely as structure refiners and remain tightly coupled to GNN backbones, limiting their capacity to capture deeper interactions between text and structure.

**Graph Robustness Evaluation.** For GNNs and RGNNs, Zheng et al. (2021) propose GRB, which evaluates model robustness under GIAs. More recently, Guo et al. (2024a) initiate the study of robustness for LLM-based predictors in TAGs. TrustGLM (Zhang et al., 2025a) extends GraphLLM as the evaluation target and explores defense strategies such as noise injection. Olatunji et al. (2025) propose a deep evaluation into GraphLLMs against text and structural attacks. However, existing evaluation frameworks lack uniformity and fairness across different model categories and attack settings. This limitation obscures the key findings presented in our study.

## 7 CONCLUSION

This paper presents a comprehensive evaluation of graph learning methods on TAGs against both textual and structural attacks, evaluating GNNs, RGNNs, GraphLLMs, and LLMs across ten datasets from four domains under transductive poisoning and inductive evasion settings. The experiments reveal key insights: different classifier types exhibit distinct text-structure trade-offs; simple RGNN can shine again with a proper text encoder; and GNNs and LLMs demonstrate vulnerabilities to different attack types. The paper presents a novel method, SFT-auto, to address the identified trade-off, introducing a unified LLM-based framework that is robust against both textual and structural perturbations. The paper also includes comprehensive ablation studies and evaluations against adaptive and hybrid attacks in the appendix, establishing a foundation for future research in TAG security.

## REPRODUCIBILITY STATEMENT

We provide implementation details, configurations, and model cards in Appendix B. We provide detailed dataset descriptions in Appendix C. We provide complete specifications and hyperparameters of the attacks in Appendix D, and descriptions of all defense models, hyperparameters, and variants in Appendix E.

We report the full numerical results underlying the main text, including per-dataset, per-model, and per-attack breakdowns in Appendix J, enabling exact comparison and re-analysis.

## ACKNOWLEDGEMENTS

The work was partially done at Gaoling School of Artificial Intelligence, Beijing Key Laboratory of Research on Large Models and Intelligent Governance, Engineering Research Center of Next-Generation Intelligent Search and Recommendation, MOE, and Pazhou Laboratory (Huangpu), Guangzhou, Guangdong 510555, China. This research was supported in part by National Natural Science Foundation of China (No. 92470128, No. U2241212), by National Science and Technology Major Project (2022ZD0114802), by Beijing Outstanding Young Scientist Program No.BJJWZYJH012019100020098, National Key Research and Development Plan of China (2023YFB4502305), and Ant Group through CCF-Ant Research Fund. We also wish to acknowledge the support provided by the fund for building world-class universities (disciplines) of Renmin University of China, by Engineering Research Center of Next-Generation Intelligent Search and Recommendation, Ministry of Education, by Intelligent Social Governance Interdisciplinary Platform, Major Innovation & Planning Interdisciplinary Platform for the "Double-First Class" Initiative, Public Policy and Decision-making Research Lab, and Public Computing Cloud, Renmin University of China.

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

# Contents

## A USAGE OF LLMS

In this study, Large Language Models (LLMs) were employed exclusively for the purpose of linguistic enhancement, including grammatical corrections and stylistic improvements. The conception, development, and interpretation of all technical material, experimental designs, analytical processes, and findings were conducted without any contribution from LLMs. The authors affirm their complete accountability for the entire content of this work.

## B IMPLEMENTATION AND CONFIGURATIONS

### B.1 IMPLEMENTATION

We rely on GreatX Li et al. (2022a) for GNN and RGNN implementations, and on NodeBed Wu et al. (2025) for GraphLLM and dataset loading.

### B.2 CONFIGURATION

Experiments were run on a machine with an NVIDIA A100-SXM4 GPU (80 GB), an Intel Xeon CPU (2.30 GHz), and 512 GB of RAM.

### B.3 MODEL CARDS

We used the following public models (links to their official model cards):

- Mistral-7B (Jiang et al., 2023)
- Llama 3.1–8B (Dubey et al., 2024)
- Ministral-8B (Mistral AI Team, 2024)
- Qwen3–8B (Yang et al., 2025)
- GPT-4o mini (Hurst et al., 2024)
- RoBERTa (Liu et al., 2019)
- MiniLM (Wang et al., 2020)

## C DETAILS OF DATASETS

We evaluate our methods on 10 text-attributed graph datasets spanning four domains, selected from the LLMNodeBed benchmark (Wu et al., 2025) to ensure comprehensive coverage of real-world scenarios.

**Academic Networks**

- **Cora** (Sen et al., 2008): Computer science papers organized into seven research areas.
- **CiteSeer** (Giles et al., 1998): CS publications spanning six categories including AI and databases.
- **Pubmed** (Yang et al., 2016): Biomedical literature focused on diabetes research with three classification types.
- **ArXiv** (Hu et al., 2020): Large-scale CS paper collection covering 40 specialized subcategories from the arXiv repository.

**Web Link Network**

- **WikiCS** (Mernyei & Cangea, 2020): Computer science Wikipedia articles categorized into ten technical domains, interconnected through hyperlink references.

**Social Networks**

- **Instagram** (Huang et al., 2024): User profiles differentiated between personal and business accounts based on profile characteristics.
- **Reddit** (Huang et al., 2024): Community users classified by engagement levels using posting history and interaction patterns.

**E-Commerce Networks**

- **History** (Yan et al., 2023): Historical literature products with detailed categorical organization.
- **Photo** (Yan et al., 2023): Photography equipment spanning professional and consumer categories.
- **Computer** (Yan et al., 2023): Technology products including hardware components and accessories.

Table 3: Dataset statistics for the 10 evaluated datasets from LLMNodeBed (Wu et al., 2025).

| Domain | Dataset | Classes | Nodes | Edges |
|---|---|---|---|---|
| Academic | Cora | 7 | 2,708 | 5,429 |
| | CiteSeer | 6 | 3,186 | 4,277 |
| | Pubmed | 3 | 19,717 | 44,338 |
| | arXiv | 40 | 169,343 | 1,166,243 |
| Web Link | WikiCS | 10 | 11,701 | 215,863 |
| Social | Instagram | 2 | 11,339 | 144,010 |
| | Reddit | 2 | 33,434 | 198,448 |
| E-Commerce | History | 12 | 41,551 | 358,574 |
| | Photo | 12 | 48,362 | 500,928 |
| | Computer | 10 | 87,229 | 721,081 |

## D   DETAILS OF ATTACK METHODS

### D.1   STRUCTURAL ATTACKS

**PGD Attack (Xu et al., 2019).**   A white box gradient-based discrete GMA that iteratively perturbs the graph structure via projected gradient ascent over continuous relaxation variables, followed by stochastic binarization to apply edge additions/removals under a budget constraint.

*Hyperparameters:*

- Learning rate: $\eta_0 = 0.1$.
- Optimization: 200 optimization epochs + 20 sampling epochs.
- Budget: Default 20% of total edges.
- Target Embedding: BoW.

**PGD-Guard (Threshold-based PGD) (Mujkanovic et al., 2022).**   An adaptive variant that constrains perturbations to pairs of nodes whose cosine similarity exceeds a threshold, emulating defense-aware strategies intended to bypass similarity filtering (targeting GNNGuard (Zhang & Zitnik, 2020)).

*Hyperparameters:*

- Cosine similarity thresholds: $[0.0, 0.3, 0.5, 0.7]$.
- Base settings: same as standard PGD.

- Budget: 20% of edges.
- Target Embedding: RoBERTa.

**GRBCD Attack (Geisler et al., 2021).** A white-box GMA based on greedy randomized block coordinate descent over the discrete edge space. At each step, random edge blocks are scored by gradients and greedily flipped within the budget.

*Hyperparameters:*

- Block size: 1,000,000.
- Sampling: 50 trials per iteration; 20 final samples for selection.
- Early stopping: patience-based with tolerance $\epsilon = 10^{-7}$.
- Target Embedding: BoW.

**HeuristicAttack (Li et al., 2023).** A scalable DICE-style heuristic ("Disconnect Internally, Connect Externally") with training-aware constraints that prioritizes edges involving training nodes and degree-based node selection, approaching gray-box poisoning MetaAttack (Zügner & Günnemann, 2019) performance via distribution shifts maximization.

*Hyperparameters:*

- Add vs. remove probability: 0.5. We tried from $[0.3, 0.5, 0.7, 0.9, 1.0]$, and found threshold= 0.5 yields most stable performance.
- Node sampling: inverse-degree probability (lower degree $\Rightarrow$ higher probability).

### D.2 Text-based Attacks

**LLM Text Attack.** We generate neighborhood-aware prompts to induce an LLM to rewrite node texts so that the predicted label is driven away from (i) the node's current class and (ii) the dominant classes in its immediate neighborhood, while preserving length and fluency.

*Prompt Template (instantiated per target node).*

---

**Graph node classification task**
Available classes: {classes_str}

Target node {node_id}:
Original text: "{text}"
Original label: "{current_label}"
{neighbor_info}   *(e.g., "Neighbor labels: [. . . ] (counts: {. . . })" or "No neighbors found")*

**Task.** Rewrite the text to be as different as possible from the original while keeping a similar length.
**Requirements:**

- Must *not* belong to the original class: "{current_label}".
- Should *not* belong to neighbor classes: {unique_neighbor_labels}  or  None.
- {target_instruction} *(e.g., prefer a class from the allowed set, or the least frequent neighbor class if all are forbidden).*
- Make the content maximally dissimilar from the original semantics.
- Keep the word count roughly similar.
- Produce content that is most unlikely under the node/neighbor context for the target class.

Goal: create text that is jointly inconsistent with the original content and its local graph context.
Return *only* the modified text, with no explanations or notes.
**Modified text:**

---

*Algorithm.*

- **Target selection:** sample nodes with degree-weighted probabilities. Lower degrees, higher probabilities.

- **Context extraction:** for each node, gather its original text, current label, neighbors' labels, and counts.

- **Constraint synthesis:** define the forbidden set as {current label} ∪ {neighbor labels}; compute the allowed label set over all classes. If empty, choose the least frequent neighbor label; else prefer a label maximally different from the forbidden set.

- **Prompting:** instantiate the above template with {`classes_str`}, {`node_id`}, {`text`}, {`current_label`}, {`neighbor_info`}, and {`target_instruction`}.

- **Generation:** query the LLM to generate a response.

- **Post-processing:** clean text, validate constraints (length and class-avoidance heuristics), and embed to update node features.

*Hyperparameters.*

- Backbone LLM: GPT-4o-mini (Hurst et al., 2024).

- Temperature: 0.7.

- Target nodes: Training if under the poison setting, test nodes if under the evasion setting.

- Budget: Poison: 80% of the training nodes; Evasion: 40% of the test nodes.

### D.3 HYBRID TEXT-LEVEL GRAPH INJECTION ATTACKS

**WTGIA: Word-level Text GIA (Lei et al., 2024).** We follow the original WTGIA pipeline with dataset-specific adaptations during inductive learning.

*Configuration.*

- Text generator: Llama-3.1-8B (Dubey et al., 2024) with no-topic prompts and vocabulary masking.

- Edge connectivity: We use $n_{\text{inject\_edges}} = \text{num\_edges} \times \text{ptb\_rate}$, so the number of total inject edges aligned to GMA rates if the budget is set the same. Specifically, we yield 17 (Cora), 9 (CiteSeer), 22 (PubMed) at ptb_rate $= 0.2$.

- Node injections: 60 (Cora), 90 (CiteSeer), 400 (PubMed) at ptb_rate $= 0.2$; scale proportionally for ptb_rate $= 0.4$.

- BoW sparsity: 0.15 (Best according to the original paper).

- FGSM optimization: step size $\epsilon = 0.01$ for 100 epochs; sequential injection steps of 0.2 with ATDGIA strategy.

- Batching: 50 for PubMed; 1 for other datasets.

### D.4 ATTACKS EXCLUDED IN THE MAIN PAPER

In this paper, we focus on untargeted attacks and text-based attacks. Therefore, methods such as Nettack (Zügner et al., 2018) and feature attacks in GRB (Zheng et al., 2021) are not employed. Additionally, while some attacks conform to our experimental setting, we choose not to adopt them. In this subsection, we provide detailed justification for these exclusions.

#### D.4.1 WHY NOT METTACK?

**Mettack (Zügner & Günnemann, 2019).** Mettack (Zügner & Günnemann, 2019) is a gray-box structural poisoning attack that employs a surrogate GCN and bi-level optimization with meta-gradients to identify vulnerable edges.

*Hyperparameters.*

- Surrogate learning rate: 0.1; momentum: 0.9.

- Meta learning rate: adaptive.

- Meta epochs: 100.

- Regularization modes: $\lambda \in \{0.0 \text{ (meta-self)}, 0.5 \text{ (meta-both)}, 1.0 \text{ (meta-train)}\}$. We use $\lambda \in \{0.0 \text{ (meta-self)}\}$ as it yields the strongest attacks.

In the main paper, we opted for HeuristicAttack instead of Mettack for the following reasons:

- The attack performance of Mettack is not significant when transferring to validation-based defenses, as demonstrated in Table 4. This phenomenon is also evidenced by the GreatX repository (Li et al., 2022a).

- Mettack suffers from scalability limitations, making it applicable only to datasets of a size comparable to the PubMed level.

Table 4: Attack performance comparison on Cora and CiteSeer datasets using a surrogate GCN with 64 hidden units. Results show mean accuracy $\pm$ standard deviation across three runs.

| Dataset | Attack Method | Clean | No Validation | With Validation |
|---|---|---|---|---|
| Cora | HeuristicAttack | 82.91±0.83 | 61.30±1.00 | 70.33±2.89 |
| | Mettack | 82.91±0.83 | 75.13±1.28 | 77.86±0.08 |
| CiteSeer | HeuristicAttack | 71.65±0.61 | 65.90±1.18 | 70.64±1.11 |
| | Mettack | 71.65±0.61 | 62.00±0.69 | 68.82±0.15 |

In contrast, HeuristicAttack, with its superior scalability and more consistent performance across validation conditions, is a more flexible and reliable choice for evaluation.

### D.4.2 WHY NOT TEXTATTACK?

**Introduction of Text Adversarial Attack.** While our main approach employs LLM-based text generation for adversarial attacks, one may be concerned about the unnoticeability of such substitutions. To address this limitation, we conduct a comprehensive analysis using established NLP adversarial attack methods that explicitly optimize for imperceptibility. Specifically, we employ the TextAttack library and select TextFooler as our primary attack method, including BAE, PWWS, and HotFlip, in preliminary evaluations.

TextFooler operates by strategically perturbing individual words within input sentences to generate semantically equivalent yet syntactically modified text. The method prioritizes semantic preservation while introducing subtle lexical modifications, rendering the perturbations challenging to detect for both human evaluators and automated systems. This characteristic makes TextFooler particularly suitable when imperceptibility constitutes a critical requirement.

*Experimental Configuration.* We configure TextFooler with MiniLM embeddings and utilize default parameters from the TextAttack library (Morris et al., 2020). The victim model is set as GCN. Our evaluation employs a perturbation rate of 0.4 across all experiments. The experiment environment is set as the inductive evasion setting.

*Key Findings.* Our empirical analysis reveals a critical dependency between attack effectiveness and the alignment of embedding representations used in both the attack generation and target model defense mechanisms. As shown in Table 5, when the attack embeddings (MiniLM) match those employed by the target model (MiniLM), TextFooler demonstrates substantially degraded performance across all evaluated GNN architectures, with a notable performance drop. However, as shown in Tables 6 and 7, when embedding misalignment occurs—specifically when target models utilize different embedding schemes such as BoW or RoBERTa, the attack effectiveness diminishes considerably.

These results provide compelling evidence that the text adversarial attack still overfits the surrogate model and the embedding type. To ensure an effective attack strength consistently, we use the LLM-based text attack that generally degrades the performance of all backbones with all encoders.

Table 5: TextFooler attack results, MiniLM embedding for the defender. Bold indicates best performance, underline indicates second best.

| Method | Cora | CiteSeer | PubMed | WikiCS | Instagram | Reddit | History |
|--------|------|----------|--------|--------|-----------|--------|---------|
| GCN | $57.32 \pm 2.84$ | $50.78 \pm 2.14$ | $63.32 \pm 1.19$ | $\mathbf{74.26 \pm 13.51}$ | $52.85 \pm 1.89$ | $53.80 \pm 0.49$ | $84.47 \pm 0.40$ |
| GAT | $\underline{64.88 \pm 7.50}$ | $41.43 \pm 2.29$ | $\underline{66.59 \pm 5.63}$ | $51.86 \pm 22.41$ | $52.38 \pm 4.75$ | $53.07 \pm 1.44$ | $83.82 \pm 0.60$ |
| APPNP | $53.75 \pm 1.22$ | $52.40 \pm 1.67$ | $61.94 \pm 0.88$ | $60.99 \pm 15.99$ | $55.67 \pm 0.83$ | $46.40 \pm 0.44$ | $\underline{84.80 \pm 0.48}$ |
| GPRGNN | $50.62 \pm 0.43$ | $44.98 \pm 0.68$ | $65.68 \pm 1.10$ | $59.02 \pm 16.72$ | $49.54 \pm 5.39$ | $45.54 \pm 0.79$ | $82.96 \pm 0.66$ |
| RobustGCN | $\mathbf{76.32 \pm 3.28}$ | $\mathbf{68.34 \pm 0.80}$ | $\mathbf{72.91 \pm 1.10}$ | $64.67 \pm 0.61$ | $62.13 \pm 0.47$ | $\mathbf{57.69 \pm 0.83}$ | $\mathbf{84.89 \pm 0.44}$ |
| NoisyGCN | $59.35 \pm 1.61$ | $53.24 \pm 1.09$ | $63.78 \pm 0.76$ | $64.15 \pm 13.95$ | $50.24 \pm 1.82$ | $\underline{54.03 \pm 0.79}$ | $84.55 \pm 0.34$ |
| GRAND | $60.02 \pm 1.94$ | $\underline{66.93 \pm 1.12}$ | $65.35 \pm 0.56$ | $49.55 \pm 0.92$ | $\mathbf{63.76 \pm 0.35}$ | $51.08 \pm 0.69$ | $83.03 \pm 0.52$ |
| EvenNet | $63.22 \pm 2.30$ | $\underline{59.77 \pm 0.27}$ | $65.91 \pm 0.97$ | $62.95 \pm 15.51$ | $54.29 \pm 2.65$ | $47.83 \pm 1.58$ | $84.73 \pm 0.56$ |
| GNNGuard | $49.57 \pm 0.98$ | $46.13 \pm 1.40$ | $66.45 \pm 0.32$ | $59.48 \pm 15.24$ | $42.12 \pm 1.44$ | $44.98 \pm 3.69$ | $82.91 \pm 0.35$ |

Table 6: TextFooler attack results, BoW embedding for the defender. Bold indicates best performance, underline indicates second best.

| Method | Cora | CiteSeer | PubMed | WikiCS | Instagram | Reddit | History |
|--------|------|----------|--------|--------|-----------|--------|---------|
| GCN | $85.30 \pm 1.61$ | $72.99 \pm 1.93$ | $86.13 \pm 0.30$ | $81.52 \pm 0.24$ | $63.11 \pm 0.70$ | $60.54 \pm 0.93$ | $81.79 \pm 0.11$ |
| GAT | $\underline{86.04 \pm 1.91}$ | $\underline{73.30 \pm 0.94}$ | $86.12 \pm 0.20$ | $81.33 \pm 0.32$ | $64.43 \pm 1.05$ | $61.51 \pm 1.21$ | $\underline{82.13 \pm 0.60}$ |
| APPNP | $\underline{85.73 \pm 2.16}$ | $\underline{70.85 \pm 0.38}$ | $85.04 \pm 0.11$ | $78.46 \pm 1.62$ | $63.30 \pm 1.48$ | $56.88 \pm 0.95$ | $\underline{81.35 \pm 0.26}$ |
| GPRGNN | $81.49 \pm 1.80$ | $69.80 \pm 0.63$ | $84.21 \pm 0.43$ | $79.18 \pm 1.61$ | $63.96 \pm 0.46$ | $59.26 \pm 0.97$ | $78.28 \pm 0.57$ |
| RobustGCN | $\mathbf{86.59 \pm 1.52}$ | $72.73 \pm 0.13$ | $86.39 \pm 0.57$ | $\underline{82.41 \pm 0.37}$ | $\mathbf{65.86 \pm 0.40}$ | $59.81 \pm 0.39$ | $81.48 \pm 0.67$ |
| NoisyGCN | $85.55 \pm 1.28$ | $72.36 \pm 1.41$ | $86.01 \pm 0.30$ | $81.65 \pm 0.35$ | $62.82 \pm 1.00$ | $\underline{61.63 \pm 0.11}$ | $\mathbf{82.19 \pm 0.14}$ |
| GRAND | $83.83 \pm 2.18$ | $\mathbf{73.72 \pm 1.61}$ | $\mathbf{87.47 \pm 0.47}$ | $80.44 \pm 0.37$ | $64.45 \pm 0.25$ | $\mathbf{62.77 \pm 1.99}$ | $79.42 \pm 0.68$ |
| EvenNet | $83.70 \pm 2.03$ | $71.21 \pm 1.04$ | $\underline{87.43 \pm 0.29}$ | $\mathbf{82.53 \pm 0.35}$ | $\underline{64.93 \pm 1.39}$ | $60.06 \pm 0.73$ | $81.07 \pm 0.51$ |
| GNNGuard | $80.38 \pm 1.22$ | $66.14 \pm 1.23$ | $82.94 \pm 0.49$ | $69.09 \pm 3.86$ | $62.48 \pm 0.46$ | $54.75 \pm 0.52$ | $78.04 \pm 0.32$ |

Table 7: TextFooler attack results, RoBerta embedding for the defender. Bold indicates best performance, underline indicates second best.

| Method | Cora | CiteSeer | PubMed | WikiCS | Instagram | Reddit | History |
|--------|------|----------|--------|--------|-----------|--------|---------|
| GCN | $\mathbf{87.39 \pm 1.00}$ | $75.18 \pm 0.87$ | $86.35 \pm 0.22$ | $\underline{84.39 \pm 0.46}$ | $66.06 \pm 0.68$ | $\underline{66.41 \pm 0.38}$ | $85.01 \pm 0.30$ |
| GAT | $86.84 \pm 1.28$ | $\underline{75.60 \pm 0.70}$ | $87.15 \pm 0.23$ | $83.80 \pm 0.82$ | $\underline{67.09 \pm 0.75}$ | $63.92 \pm 0.31$ | $84.59 \pm 0.54$ |
| APPNP | $79.40 \pm 1.39$ | $73.35 \pm 0.92$ | $88.21 \pm 0.30$ | $80.56 \pm 4.05$ | $64.27 \pm 1.16$ | $57.39 \pm 0.16$ | $\underline{85.74 \pm 0.38}$ |
| GPRGNN | $82.41 \pm 0.68$ | $70.53 \pm 1.63$ | $85.21 \pm 0.29$ | $79.51 \pm 4.44$ | $62.24 \pm 1.49$ | $54.89 \pm 1.44$ | $84.65 \pm 0.62$ |
| RobustGCN | $\underline{86.96 \pm 1.99}$ | $74.03 \pm 0.32$ | $86.52 \pm 0.09$ | $83.28 \pm 0.59$ | $\mathbf{67.39 \pm 0.17}$ | $59.42 \pm 0.79$ | $84.57 \pm 0.44$ |
| NoisyGCN | $86.90 \pm 0.90$ | $75.39 \pm 0.80$ | $86.35 \pm 0.28$ | $83.75 \pm 1.14$ | $66.48 \pm 0.75$ | $\mathbf{66.52 \pm 0.42}$ | $85.04 \pm 0.19$ |
| GRAND | $86.10 \pm 1.21$ | $\mathbf{77.27 \pm 0.34}$ | $\mathbf{89.78 \pm 0.35}$ | $83.13 \pm 0.85$ | $66.49 \pm 0.41$ | $63.53 \pm 1.27$ | $\mathbf{85.92 \pm 0.41}$ |
| EvenNet | $83.09 \pm 1.55$ | $74.19 \pm 0.64$ | $\underline{89.45 \pm 0.57}$ | $\mathbf{84.91 \pm 1.63}$ | $65.42 \pm 0.79$ | $59.59 \pm 0.36$ | $85.62 \pm 0.57$ |
| GNNGuard | $71.40 \pm 1.74$ | $68.65 \pm 0.46$ | $82.27 \pm 0.82$ | $75.48 \pm 5.68$ | $60.58 \pm 1.71$ | $54.14 \pm 1.26$ | $84.64 \pm 0.52$ |

# E DETAILS OF DEFENSE METHODS

## E.1 INTRODUCTION OF DEFENSE MODEL

For GNNs and RGNNs, we have the following methods as baselines.

**Spatial/Message Passing Models**

1. **GCN** (Kipf & Welling, 2017): Graph Convolutional Network using localized spectral convolution with Chebyshev polynomials.

2. **GAT** (Velickovic et al., 2018): Graph Attention Network employing multi-head attention mechanisms for adaptive neighborhood aggregation.

**Spectral Models**

1. **APPNP** (Klicpera et al., 2019): Approximate Personalized Propagation of Neural Predictions, combining neural predictions with personalized PageRank.

2. **GPRGNN** (Chien et al., 2021): Generalized PageRank Graph Neural Network with learnable graph filter coefficients.

3. **EvenNet** (Lei et al., 2022): Spectral-based defense using even convolution networks with teleportation mechanisms.

**Robust Training Methods**

1. **GRAND** (Feng et al., 2020): Graph Random Neural Networks with consistency regularization using random propagation and DropNode.

2. **NoisyGCN** (Ennadir et al., 2024): GCN with feature noise injection during training to improve robustness.

3. **GPRGNN-AT** (Gosch et al., 2023): GPRGNN with adversarial training. Results are left to Appendix K.

**Probabilistic Methods**

1. **RobustGCN** (Zhu et al., 2019): Robust Graph Convolutional Network with Gaussian-based attention and variance-based message passing.

**Similarity-based Methods**

1. **GNNGuard** (Zhang & Zitnik, 2020): Attention-based neighborhood filtering using cosine similarity thresholds to detect and mitigate adversarial edges.

1. **ElasticGNN** (Liu et al., 2021): Elastic message passing with $L_1/L_2$ regularization for handling graph heterophily.

2. **SoftMedianGDC** (Geisler et al., 2021): Soft median aggregation with Gaussian Diffusion Convolution and temperature control.

3. **RUNG** (Hou et al., 2024): Robust Graph Neural Networks with uncertainty quantification and Laplacian smoothing.

**Other Architectural Improvements**

1. **GCORN** (Abbahaddou et al., 2024): Higher-order Graph Convolutional Networks with polynomial filters and weight regularization.

**Unsupervised Structure Cleaning**

1. **Jaccard-GCN** (Wu et al., 2019): Preprocesses graphs by removing edges with low Jaccard similarity between node features.
2. **Cosine-GCN** (Mujkanovic et al., 2022): Edge filtering based on cosine similarity thresholds between node feature vectors.

**Supervised Structure Learning**

1. **ProGNN** (Jin et al., 2020): Joint optimization of graph structure and GNN parameters with sparsity and smoothness constraints.
2. **Stable** (Li et al., 2022b): An unsupervised pipeline that optimizes graph structure by learning edge weights using a metric function combining node feature and structure information. It employs Cosine and Jaccard similarity with learnable thresholds to filter adversarial edges.
3. **GOOD-AT**: It trains an ensemble of OOD detectors using adversarial edges as supervision signals to learn which edges are anomalous, then removes detected perturbations to recover a cleaner graph structure. Results are left to Appendix K.
4. **GPR-GAE**: It learns to reconstruct clean graph structure via a self-supervised graph autoencoder with multiple GPR filters, using the original graph's reconstruction loss as implicit supervision for structure learning. Results are left to Appendix K.

**GraphLLM Defenses**

1. **GraphGPT** (Tang et al., 2024): Graph-text alignment model using contrastive learning between graph embeddings and text representations.
2. **LLaGA** (Chen et al., 2024): Large Language and Graph Assistant that aligns graph structural information with language model representations through multi-modal learning.
3. **SFT with Neighbors** (Wang et al., 2025b): Supervised Fine-Tuning approach that incorporates neighborhood information into LLM prompts for enhanced graph understanding.

### E.2 CONFIGURATION AND HYPERPARAMETERS

This sub-section details configurations and hyperparameters for both GNN-based defenses and GraphLLM methods used in our evaluation framework.

#### E.2.1 GNN AND RGNNs

**General Settings** All GNN-based defense models share the following general hyperparameters:

- **Learning rate**: 0.01 (consistent across all methods)
- **Weight decay**: Grid search over [0.0, 0.0005]
- **Dropout**: Grid search over [0.5, 0.7] (except model-specific variations)
- **Hidden dimensions**: 128 for small datasets (Cora, CiteSeer, PubMed, Instagram, WikiCS); 256 for large datasets
- **GAT adjustment**: Hidden dimension reduced by factor of 8 due to multi-head attention
- **Training epochs**: Dataset-dependent with early stopping
  - Small datasets (Cora, CiteSeer, Instagram, PubMed, WikiCS): 400 epochs
  - Medium datasets (Computer, Photo, History, Reddit): 600 epochs
  - Large datasets (ArXiv): 1000 epochs
- **Patience**: Dataset-dependent early stopping
  - Small datasets: 100 epochs patience
  - Medium datasets: 200 epochs patience
  - Large datasets: 400 epochs patience

**Specific Hyperparameters.**   The following table summarizes the model-specific hyperparameter ranges for GNN-based defense methods. For the additional baselines in Appendix K, we use the default hyperparameters following official implementations.

| Model | Parameter | Values |
|---|---|---|
| GCN | None | None |
| GAT | Num_Heads | 8 |
| APPNP | Alpha | [0.1, 0.3, 0.7, 0.9] |
| GPRGNN | Alpha | [0.1, 0.3, 0.7, 0.9] |
| RobustGCN | None | None |
| ElasticGNN | Lambda1
Lambda2 | [3, 6]
[3, 6] |
| GNNGuard | Threshold | [0.3, 0.4, 0.5, 0.6, 0.7] |
| NoisyGCN | Beta | [0.1, 0.3, 0.5] |
| GCORN | None | None |
| ProGNN | Alpha
Beta | [0.0005, 0.3]
[1.5, 2.5] |
| Stable | Cosine Threshold
Jaccard Threshold
Alpha | [0.3, 0.5, 0.7]
[0.02, 0.03]
[0.1, 0.03, 0.6] |
| GCN-Jaccard | Jaccard Threshold | [0.03, 0.05, 0.1] |
| GCN-Cosine | Cosine Threshold | [0.3, 0.4, 0.5, 0.6, 0.7] |
| GRAND | Dropnode
Order
MLP Input Dropout
N Samples
Reg Consistency
Sharpening Temperature | [0.5]
[2, 4]
[0.5]
[4]
[0.7, 1.0]
[0.5] |
| SoftMedianGDC | Temperature
Teleport Probability
Neighbors | [0.5, 1.0]
[0.15, 0.25]
[64] |
| RUNG | Lambda
Gamma | [0.7, 0.9]
[1, 3] |
| EvenNet | K
Alpha
DP Rate | [10]
[0.1, 0.3, 0.7, 0.9]
[0.5] |

### E.2.2   GRAPHLLMS

**General Configurations**   GraphLLM methods employ distinct training configurations optimized for large-scale language model integration:

- **Base LLM**: Mistral-7B (4096-dimensional output) as primary backbone
- **Text encoding**: RoBERTa for feature extraction and alignment
- **Batch processing**: Varies by model complexity (8-64 samples per batch)
- **Learning rates**: Lower rates (1e-4) for stable LLM fine-tuning
- **Weight decay**: 0.05 for regularization
- **Gradient clipping**: Applied to prevent exploding gradients
- **Mixed precision**: Enabled for memory efficiency

**GraphGPT.** For GraphGPT, we adhere to the best hyperparameters in clean datasets provided by (Wu et al., 2025).

**LLaGA.** For LLaGA, we adhere to the best hyperparameters in clean datasets provided by (Wu et al., 2025).

- **Language model embedding**: RoBERTa
- **Neighborhood template**: HO (Hopfield) encoding

**SFT with Neighbors.** Supervised Fine-Tuning incorporates neighborhood-aware prompting with LoRA optimization:

- **Maximum neighbors**: maximum_neighbor=6 for context window management
- **Neighbor filtering strategy**: Degree-based selection - neighbors are ranked by node degree in descending order, and top-6 highest-degree neighbors are selected for prompt inclusion
- **LoRA configuration**: r=8, alpha=16, dropout=0.1, target_modules=[q_proj, v_proj]
- **Sequence lengths**: max_txt_length=128, max_origin_txt_length=128, max_ans_length=16
- **Optimization**: AdamW optimizer with gradient accumulation
- **Prompt engineering**: Integrates 1-hop neighbor information with degree-based prioritization for enhanced context understanding

*Neighbor-aware Prompt Template:*

---

**Node Classification Task**
Question: You are doing node classification task in a citation graph. Given the content of the center node: {origin_text} and its neighbor information: {neighbor_text}, each node represents a paper and the relationship represents the citation relationship between papers, we need to classify the center node into 7 classes: {classes}. Please tell me which class the center node belongs to? Answer only the class name without any other words.
Answer:

---

### E.2.3 GraphLLMs with Noisy Training

This subsection covers noise injection strategies across different GraphLLM architectures in Section

**GraphGPT with Noise.** GraphGPT incorporates noise injection through graph-level modifications using the noise_utils framework. Supports only the structural "noise" variant. Noise is applied globally to the entire graph structure before training. Configurations follow the clean dataset parameters from (Wu et al., 2025) with additional noise strategies applied during contrastive learning. The perturb ratio is set to 10%.

GraphGPT-noisetxt and GraphGPT-noisefull are excluded because the noise injected hurts clean performance too much, as shown in Table 9.

Table 9: Performance comparison of GraphGPT variants across under the inductive setting on clean datasets.

| Method | Cora | CiteSeer | PubMed | WikiCS |
|---|---|---|---|---|
| GraphGPT | $81.06 \pm 2.33$ | $74.35 \pm 2.51$ | $94.14 \pm 0.23$ | $82.31 \pm 1.31$ |
| GraphGPT-noise | $80.63 \pm 2.24$ | $74.56 \pm 3.07$ | $94.09 \pm 0.33$ | $82.32 \pm 1.97$ |
| GraphGPT-noisetxt | $66.79 \pm 3.84$ | $57.37 \pm 5.20$ | $86.15 \pm 1.88$ | $64.56 \pm 2.94$ |

**LLaGA with Noise.** LLaGA applies noise injection through graph-level modifications using the noise_utils framework. The noise strategies modify the global graph structure or replace text content across the entire graph before multi-modal learning. Noise integration occurs with the RoBERTa language model embedding and HO (Hopfield) neighborhood template encoding. Base configurations identical to clean LLaGA training. The perturb ratio is set to 10%.

**SFT with Noise Variants.** SFT implements three distinct noise injection strategies during training:

*Noise (Structure):* Injection strategy targets high-degree nodes with 10% probability, adding random unconnected nodes as fake neighbors.

*NoiseTxt (Text):* Injection strategy replaces text content of 10% high-degree nodes with text from different-class nodes.

*Noisefull (Both):* Injection strategy combines both structural and text noise - applies both neighbor injection (10% probability) and text replacement (10% of high-degree nodes) simultaneously.

- **Injection ratio**: 10% probability for structural noise, 10% of high-degree nodes for text replacement
- **Target selection**: High-degree nodes (degree > average degree)
- **Replacement strategy**: Random selection from unconnected nodes (structural) or different-class nodes (text)
- **Training augmentation**: Applied only during the training phase, inference uses clean data
- **Base configurations**: Identical to SFT with Neighbors for all other parameters

*SFT Noise Training Strategy (Sample-Level):* Unlike GraphGPT/LLaGA which apply graph-level noise modifications, SFT applies noise injection at the sample level during data preparation. For structural noise, 10% probability of adding random unconnected nodes as fake neighbors to high-degree nodes during neighbor selection for each training sample. For text noise, 10% of high-degree nodes have their text content replaced with text from nodes of different classes during individual sample creation. NoiseFull combines both strategies with independent application - both structural neighbor injection and text replacement occur simultaneously for each sample.

### E.2.4 SFT WITH SIMILARITY CONSTRAINTS

**SFT-simp.** The similarity variant teaches the model to selectively use neighbor information based on similarity constraints through prompt engineering:

- **Training**: Standard neighbor-based training (identical to SFT with Neighbors)
- **Inference strategy**: Prompt-based similarity awareness
- **Similarity criteria**: Text content and label consistency

*Similarity-Aware Prompt Template:*

---

**Node Classification with Similarity Constraints**
Question: You are doing node classification task in a citation graph. Given the content of the center node: {origin_text} and its neighbor information: {neighbor_text}, each node represents a paper and the relationship represents the citation relationship between papers, we need to classify the center node into {num_classes} classes: {classes}. Consider neighbor information for classification ONLY when: 1) neighbors are similar to the center node, or 2) neighbors are similar to each other. Similarity can be based on text content or label consistency. Otherwise, ignore neighbor information. Please tell me which class the center node belongs to? Answer only the class name without any other words.
Answer:

---

### E.2.5 SFT-AUTO

**Pipeline.** The auto variant implements comprehensive attack detection and recovery through multi-stage inference with specialized prompt templates. The pseudo-code of the algorithm is provided in Algorithm 1, and the related prompts are listed below.

*Attack Detection Prompt Template:*

---

**Node Classification with Attack Detection**

Question: You are doing node classification in a citation graph. Given the content of the center node: {origin_text} and its neighbor information: {neighbor_text}, classify the center node into {num_classes} classes: {classes} or "text_attacked". The center node may be attacked. If its class is unclear or differs from most of its neighbors, classify it as "text_attacked" instead. Please tell me which class the center node belongs to? Answer only the class name without any other words.
Answer:

---

*Recovery Prompt Template (Neighbor-Only):*

---

**Recovery Using Neighbor Information Only**

Question: You are doing node classification task in a citation graph. Based only on the neighbor information: {neighbor_text}, each node represents a paper and the relationship represents the citation relationship between papers, we need to classify the center node into {num_classes} classes: {classes}. Please predict the class of the center node using only neighbor information. Answer only the class name without any other words.
Answer:

---

**Algorithm 1:** Auto Variant Framework for Adversarial Attack Detection and Recovery

---

**Input:** Graph data $G = (V, E, X)$, training nodes $V_{\text{train}}$, test nodes $V_{\text{test}}$, similarity threshold $\tau$
**Output:** Trained model $\mathcal{M}$, final predictions $\mathbf{y}_{\text{pred}}$

1 **Training Phase:**;
2 $r \leftarrow \min\left(\frac{1}{|C|}, 0.15\right)$ // Adaptive attack ratio
3 $\mathcal{S}_{\text{normal}} \leftarrow \text{PREPARENORMALSAMPLES}(V_{\text{train}})$;
4 $\mathcal{S}_{\text{attack}} \leftarrow \text{GENERATETEXTATTACKSAMPLES}(V_{\text{train}}, r)$;
5 $\mathcal{S}_{\text{recovery}} \leftarrow \text{GENERATERECOVERYSAMPLES}(\mathcal{S}_{\text{attack}})$;
6 $\mathcal{S}_{\text{all}} \leftarrow \mathcal{S}_{\text{normal}} \cup \mathcal{S}_{\text{attack}} \cup \mathcal{S}_{\text{recovery}}$;
7 $\mathcal{M} \leftarrow \text{TRAINMODEL}(\mathcal{S}_{\text{all}})$;

8 **Inference Phase:**;
9 $\mathcal{I}_{\text{text}} \leftarrow \emptyset, \mathcal{I}_{\text{struct}} \leftarrow \emptyset$;
   // Stage 1: Attack Detection
10 **foreach** $v_i \in V_{\text{test}}$ **do**
11    **if** $\text{LOWSIMILARITYNEIGHBORS}(v_i, \tau) \geq |\mathcal{N}_i|/2$ **then**
12       $\mathcal{I}_{\text{struct}} \leftarrow \mathcal{I}_{\text{struct}} \cup \{i\}$;
13    **if** $\mathcal{M}(v_i) = $ *"text_attacked"* **then**
14       $\mathcal{I}_{\text{text}} \leftarrow \mathcal{I}_{\text{text}} \cup \{i\}$;

   // Stage 2: Recovery and Final Prediction
15 **foreach** $v_i \in V_{\text{test}}$ **do**
16    **if** $i \in \mathcal{I}_{\text{text}}$ **then**
17       $\mathbf{y}_{\text{pred}}[i] \leftarrow \mathcal{M}(\text{neighbors\_only}(v_i))$;
18    **else if** $i \in \mathcal{I}_{\text{struct}}$ **then**
19       $\mathbf{y}_{\text{pred}}[i] \leftarrow \mathcal{M}(v_i, \text{filtered\_neighbors}(v_i, \tau))$;
20    **else**
21       $\mathbf{y}_{\text{pred}}[i] \leftarrow \mathcal{M}(v_i, \mathcal{N}_i)$;
22 **return** $\mathcal{M}$, $\mathbf{y}_{\text{pred}}$

---

## F  RESULTS AGAINST WTGIA

In this section, we present results against Text-level GIA and WTGIA attacks.

We introduced the WTGIA methodology and basic settings in Section D.3. Specifically, we evaluate on three datasets: Cora, CiteSeer, and PubMed, following the original paper. We present results for two perturbation rates: ptb_rate $= 0.2$ and ptb_rate $= 0.4$.

At ptb_rate = 0.2, WTGIA introduces the same number of edges as structural evasion attacks. Considering that in GMA, each edge modification affects two nodes in the original graph, we also conduct experiments at ptb_rate = 0.4, where the number of affected edges approximately equals that of GMA. By aligning the attack budgets between GMA and Text-level GIA, we can study the attack strength of Text-level GIA compared to GMA.

Following our other experiments, we evaluate both GNN and RGNN models using three embedding types: BoW, MiniLM, and RoBERTa. Note that WTGIA inherently uses BoW as the victim text encoder. In this experiment, we investigate how different text encoders affect defense performance against WTGIA.

Table 10: WTGIA Attacked Test Accuracy (Perturb ratio being 20%)

| Method | Cora | | | CiteSeer | | | PubMed | | |
|---|---|---|---|---|---|---|---|---|---|
| | BoW | MiniLM | RoBERTa | BoW | MiniLM | RoBERTa | BoW | MiniLM | RoBERTa |
| GCN | 59.29 | 83.46 | 85.36 | 49.22 | 71.73 | 74.03 | 77.49 | 84.65 | 83.95 |
| GAT | 57.56 | 80.87 | 82.60 | 45.61 | 66.30 | 67.29 | 74.02 | 74.99 | 82.21 |
| APPNP | 60.95 | 84.13 | 86.41 | 65.62 | 76.02 | 74.66 | 85.07 | 89.03 | 91.22 |
| GCORN | 53.38 | 83.83 | 81.67 | 47.49 | 73.20 | 72.47 | 61.15 | 79.74 | 80.92 |
| GPRGNN | 69.43 | 83.76 | 84.93 | 64.63 | 73.56 | 74.56 | 83.99 | 89.44 | 90.78 |
| GRAND | 59.78 | 84.62 | **86.78** | 56.22 | 75.76 | **76.38** | 80.03 | 86.49 | 89.47 |
| EvenNet | 78.78 | 82.53 | 82.60 | 68.86 | 73.77 | 73.41 | 83.64 | 86.66 | 89.19 |
| ElasticGNN | 60.02 | 84.50 | 85.42 | 54.49 | 73.88 | 73.30 | 78.25 | 79.68 | 85.86 |
| RobustGCN | 78.91 | 83.64 | 85.61 | 68.34 | 73.77 | 73.98 | 81.63 | 79.44 | 82.06 |
| GNNGuard | 67.59 | 82.90 | 83.15 | 65.46 | 75.24 | 73.72 | 82.40 | 88.95 | 90.29 |
| SoftmedianGDC | 80.87 | 84.87 | 83.70 | 71.79 | 75.24 | 74.71 | 86.32 | 89.54 | 91.20 |
| NoisyGCN | 63.78 | 83.15 | 85.06 | 49.63 | 72.10 | 73.30 | 78.26 | 85.24 | 86.03 |
| RUNG | 79.52 | 83.27 | 83.95 | 70.27 | 74.03 | 73.56 | 85.68 | 89.64 | 90.99 |
| GraphGPT | | 74.97 | | | 72.83 | | | 92.84 | |
| LLaGA | | 83.21 | | | 72.36 | | | 89.76 | |
| SFT-neighbor | | 79.89 | | | 73.51 | | | **94.74** | |

Table 11: WTGIA Attacked Test Accuracy (Perturb ratio being 40%)

| Method | Cora | | | CiteSeer | | | PubMed | | |
|---|---|---|---|---|---|---|---|---|---|
| | BoW | MiniLM | RoBERTa | BoW | MiniLM | RoBERTa | BoW | MiniLM | RoBERTa |
| GCN | 47.91 | 82.16 | 84.99 | 33.28 | 70.01 | 72.73 | 75.77 | 79.54 | 79.09 |
| GAT | 47.23 | 76.32 | 80.44 | 27.90 | 60.92 | 62.38 | 66.59 | 63.28 | 72.74 |
| APPNP | 50.55 | 84.01 | 85.67 | 60.29 | **75.39** | 74.61 | 86.07 | 88.88 | 91.18 |
| GCORN | 32.10 | 82.23 | 81.00 | 26.12 | 70.32 | 67.08 | 53.20 | 77.50 | 77.49 |
| GPRGNN | 64.45 | 82.78 | 83.70 | 52.87 | 73.20 | 72.94 | 84.04 | 89.01 | 90.64 |
| GRAND | 48.03 | 84.44 | **86.10** | 44.36 | 74.97 | 74.29 | 76.36 | 86.16 | 88.90 |
| EvenNet | 73.37 | 81.06 | 81.49 | 66.25 | 73.51 | 73.30 | 77.50 | 81.52 | 86.51 |
| ElasticGNN | 46.19 | 83.27 | 83.64 | 37.57 | 72.52 | 71.89 | 75.57 | 70.76 | 80.49 |
| RobustGCN | 75.03 | 81.80 | 85.85 | 62.12 | 73.35 | 72.99 | 75.97 | 70.62 | 75.52 |
| GNNGuard | 61.32 | 82.78 | 83.76 | 61.86 | 75.34 | 73.41 | 81.11 | 89.05 | 90.24 |
| SoftmedianGDC | 78.97 | 84.38 | 83.95 | 70.79 | 74.92 | 74.61 | 86.17 | 89.77 | 90.99 |
| NoisyGCN | 52.46 | 82.16 | 83.76 | 34.12 | 71.26 | 71.89 | 75.22 | 81.03 | 81.67 |
| RUNG | 79.83 | 82.66 | 84.19 | 68.13 | 73.46 | 74.76 | 85.81 | 89.77 | 91.11 |
| GraphGPT | | 71.16 | | | 70.22 | | | 92.29 | |
| LLaGA | | 81.86 | | | 68.23 | | | 89.32 | |
| SFT-neighbor | | 77.80 | | | 73.20 | | | **94.62** | |

The experimental results are presented in Table 10 and Table 11. We have the following discoveries: **Cross-Modal Attack Transferability**: WTGIA exhibits significantly stronger attack performance against BoW-based models compared to advanced text encoders like MiniLM and RoBERTa, even under a strong budget (40%). This pronounced performance gap highlights the limited transferability of text-level GIAs across different embeddings, suggesting that WTGIA overfits to the victim's specific encoder (BoW). In contrast, structural attacks display slightly more consistent degradation across text representations, underscoring the modality-specific nature of text-level perturbations.

**Vulnerability of Text-Aware Models**: Text-level perturbations disproportionately impact models reliant on textual features, particularly LLM-based approaches like SFT-neighbor. On the Cora

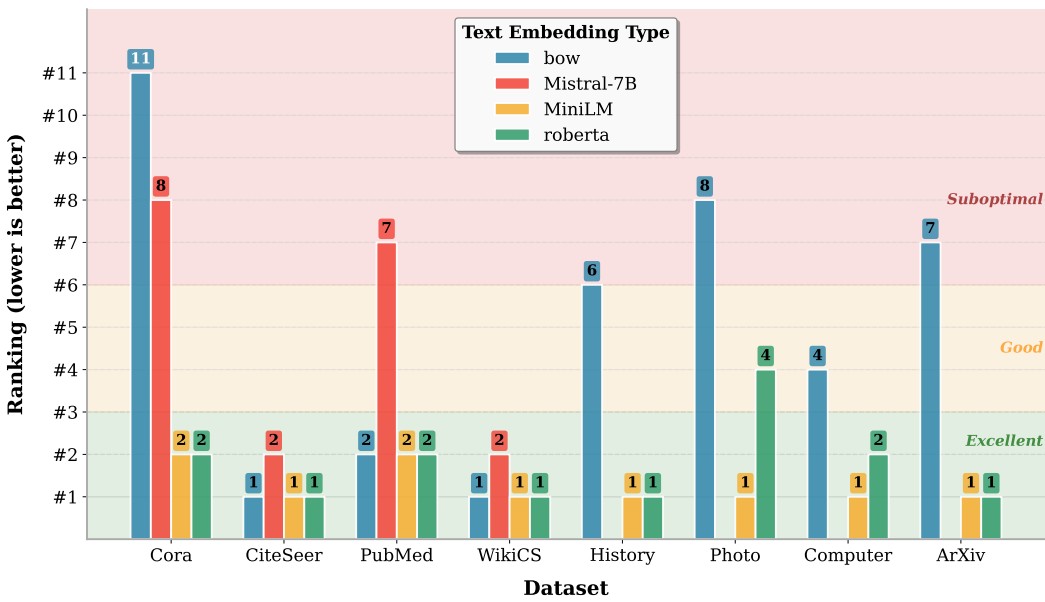

Figure 7: GNNGuard performance ranking across different text embeddings under structure attacks. The bar chart displays ranking positions (lower is better) for each embedding type across eight benchmark datasets in the inductive setting with a perturbation rate of 0.2.

dataset, SFT-neighbor maintains competitive accuracy ($\tilde{8}2\%$) under structural attacks but suffers greater degradation under WTGIA.

## G WHY GNNGUARD EXCELS: UNDERSTANDING AND IMPROVING

In the main text, we observe that GNNGuard achieves remarkable performance improvements. This raises a fundamental question: what drives GNNGuard's exceptional effectiveness? In this section, we provide a comprehensive analysis to dissect the underlying mechanisms.

### G.1 RESULTS OF GNNGUARD UNDER DIFFERENT TEXT ENCODERS

In Figure 7, we plot the performance of GNNGuard against structure evasion attacks using different text encoders. We exclude the results on Instagram and Reddit because the performance differences among all methods on these datasets are not significant. In the ranking, we remove the GraphLLM methods because they do not necessarily depend on embeddings. The results reveal substantial variations in defensive effectiveness depending on the embedding choice. In previous works (Zhang & Zitnik, 2020; Lei et al., 2022; Mujkanovic et al., 2022; Hou et al., 2024), GNNGuard is evaluated using embeddings like BoW and TF-IDF, where its ranking is indeed low as shown in the table. However, when switching to advanced language model embeddings like MiniLM and RoBERTa, its ranking improves significantly. For instance, on Cora, when using embeddings like BoW and Mistral-7B, its ranking falls within the suboptimal region, but when employing MiniLM and RoBERTa, it demonstrates improved ranking relative to other RGNNs and GNNs, indicating that text encoders significantly influence GNNGuard's ranking.

### G.2 DIFFERENCE BETWEEN TEXT-ENCODERS

To understand why the text encoders significantly affect the performance of GNNGuard, we systematically evaluate the effectiveness of different text embedding methods for similarity-based edge filtering, which is essential for its defense mechanism. Our analysis compares BoW, TF-IDF, Mistral-7B, RoBERTa, and MiniLM embeddings across multiple graph datasets to identify which representations best distinguish intra-class from inter-class edges.

Figure 8: Comprehensive embedding comparison for the Photo dataset. (a) Preserve-filter trade-off curves showing the Pareto frontier between intra-class preservation and inter-class filtering rates; (b) KDE-smoothed similarity distributions revealing the separation between intra-class (solid) and inter-class (dashed) edges; (c) Quantitative quality metrics including separation score, AUC, discriminability, threshold gap, and non-overlap score; (d) Threshold effectiveness curves illustrating how preservation and filtering rates vary with similarity thresholds; (e) Summary statistics table presenting mean similarities and standard deviations for each embedding type.

Figure 8 presents our comprehensive analysis framework using the Photo dataset as an example. The multi-panel visualization reveals critical insights about embedding effectiveness:

- **Preserve-Filter Trade-off (Panel a):** Each curve represents an embedding's ability to simultaneously preserve intra-class edges while filtering inter-class edges across 101 similarity thresholds. The curves are generated by computing, for each threshold $\tau \in [0, 1]$, the fraction of intra-class edges with similarity $\geq \tau$ (x-axis) and inter-class edges with similarity $< \tau$ (y-axis). Embeddings with curves closer to the upper-left corner exhibit superior discriminative capacity. In the Photo dataset, RoBERTa's curve dominates, achieving 80% inter-class filtering while maintaining 40% intra-class preservation.

- **Similarity Distributions (Panel b):** Kernel density estimation with Gaussian kernels ($\sigma$ selected via Scott's rule) visualizes the probability density functions of cosine similarities. The aggregation across three random seeds ensures robustness. For Photo, MiniLM, and RoBERTa exhibit clear bimodal separation with intra-class similarities and inter-class differences, while others show substantial overlap with both distributions.

- **Quality Metrics (Panel c):** Five metrics quantify embedding effectiveness:
  - Separation Score: Cohen's d $= (\mu_{\text{intra}} - \mu_{\text{inter}})/\sqrt{(\sigma_{\text{intra}}^2 + \sigma_{\text{inter}}^2)/2}$
  - AUC Score: Area under the ROC curve, treating edge classification as a binary prediction task
  - Discriminability: $\max_{\tau}[(P(\text{sim}_{\text{intra}} \geq \tau) + P(\text{sim}_{\text{inter}} < \tau))/2]$
  - Threshold Gap: $Q_{20}(\text{sim}_{\text{intra}}) - Q_{80}(\text{sim}_{\text{inter}})$ where $Q_p$ denotes percentile
  - Non-overlap: $1 - \sum_i \min(h_{\text{intra}}(i), h_{\text{inter}}(i))\Delta x$ using 50-bin histograms

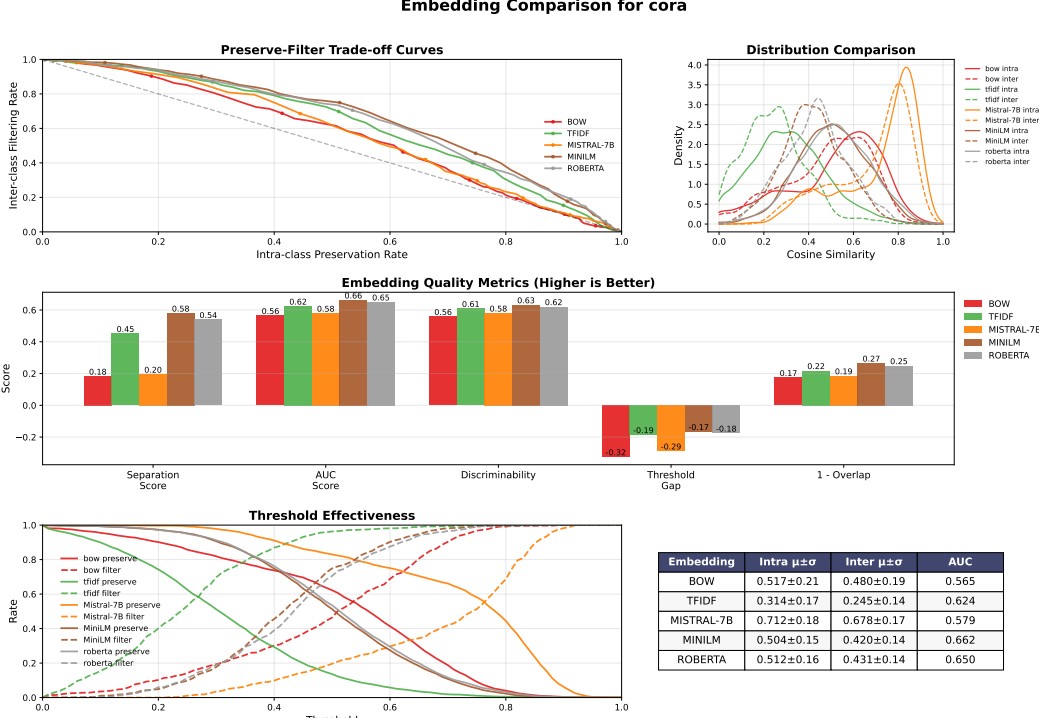

Figure 9: Comprehensive embedding comparison for the Cora dataset. (a) Preserve-filter trade-off curves showing the Pareto frontier between intra-class preservation and inter-class filtering rates; (b) KDE-smoothed similarity distributions revealing the separation between intra-class (solid) and inter-class (dashed) edges; (c) Quantitative quality metrics including separation score, AUC, discriminability, threshold gap, and non-overlap score; (d) Threshold effectiveness curves illustrating how preservation and filtering rates vary with similarity thresholds; (e) Summary statistics table presenting mean similarities and standard deviations for each embedding type.

- **Threshold Effectiveness (Panel d):** This analysis reveals the operational characteristics of each embedding. The solid lines show $P(\text{sim}_{\text{intra}} \geq \tau)$ while dashed lines show $P(\text{sim}_{\text{inter}} < \tau)$ as functions of threshold $\tau$. The vertical separation between paired curves indicates discriminative power at each threshold.

- **Statistical Summary (Panel e):**

This analysis framework demonstrates that contextual embeddings (RoBERTa, MiniLM) provide superior edge discrimination compared to sparse representations. While all GNNs and RGNNs benefit from better embeddings in the classification task, GNNGuard enjoys more protection against structural attacks with better representations due to the better distinguishability of a better embedding. Another example on the Cora dataset is provided in Figure 9.

### G.3 GUARDUAL: ROBUSTNESS AGAINST TEST-TIME STRUCTURAL ATTACKS

Despite achieving promising results with advanced embeddings, GNNGuard exhibits several limitations in practice. First, the current threshold selection relies on validation set performance, which leads to overly conservative thresholds due to the inherent performance-robustness trade-off on clean validation data. Second, in real-world scenarios where training data integrity can be more readily ensured, applying aggressive thresholds during training risks excessive edge filtering, thereby degrading model performance on benign graphs.

To address these limitations, we introduce Guardual, an adaptive defense mechanism that employs dual similarity thresholds to balance training stability with robust defense capabilities. This method

uses the optimal threshold for preserving benign graph structure during training, which differs from the threshold needed for effective adversarial filtering during inference.

### G.3.1 DUAL-THRESHOLD DESIGN

Guardual employs two complementary thresholds optimized for different phases of model deployment:

- **Conservative Threshold (Training):** This threshold prioritizes preserving intra-class edges to maintain graph connectivity and training stability. Computed using a weighted objective function: $\text{score}_{\text{conservative}} = 0.7 \cdot P_{\text{preserve}} + 0.3 \cdot P_{\text{filter}}$, where $P_{\text{preserve}}$ represents the fraction of intra-class edges retained and $P_{\text{filter}}$ denotes the fraction of inter-class edges removed. The 70-30 weighting ensures that sufficient graph structure remains intact during the learning phase, preventing performance degradation from excessive edge pruning.

- **Balanced Threshold (Testing):** During inference, the model switches to a balanced threshold that equally weights preservation and filtering: $\text{score}_{\text{balanced}} = 0.5 \cdot P_{\text{preserve}} + 0.5 \cdot P_{\text{filter}}$. This equal weighting provides stronger defense against adversarial edges while accepting slightly reduced intra-class preservation, as gradient flow is no longer a concern during evaluation.

Given specific embeddings, we can pre-compute all scores and obtain training/test thresholds. The thresholds for Guardual using RoBERTa embeddings are listed in Table 12.

Table 12: Dual thresholds employed by Guardual across different datasets. Conservative thresholds prioritize edge preservation during training, while balanced thresholds enhance adversarial filtering during testing.

| Dataset | Training Threshold (Conservative) | Testing Threshold (Balanced) |
|---|---|---|
| Cora | 0.250 | 0.503 |
| CiteSeer | 0.320 | 0.580 |
| PubMed | 0.360 | 0.633 |
| WikiCS | 0.283 | 0.433 |
| Instagram | 0.000 | 0.471 |
| Reddit | 0.000 | 0.177 |
| History | 0.260 | 0.457 |
| Photo | 0.000 | 0.476 |
| Computer | 0.000 | 0.457 |
| ArXiv | 0.320 | 0.540 |

### G.3.2 RESULTS OF GUARDUAL

The results are presented in Figure 10. We can see that, although it removes a hyperparameter compared to GNNGuard, Guardual achieves general improvement. In Computer and Photo datasets (highlighted in red), Guardual demonstrates the most significant enhancements, with improvements of +8.23% and +10.74% respectively. As shown in the result performance, it becomse the most robust RGNN against structural evasion attacks, despite its simple and effective design.

## H ADAPTIVE ATTACKS

### H.1 ADAPTIVE ATTACKS AGAINST GNNGUARD

Following Mujkanovic et al. (2022), we investigate adaptive attacks targeting GNNGuard, leveraging its robust performance against structural attacks. We employ a modified PGD attack Xu et al. (2019), termed PGDGuard, which restricts the attack search space to edges with similarity exceeding a threshold $\epsilon$. All edge additions or removals satisfy this similarity constraint. We fix the attack embedding to RoBERTa and evaluate PGDGuard across $\epsilon$ values of 0.0, 0.3, 0.5, and 0.7, assessing the performance of GNN and RGNN models under these conditions. Results are presented in Figure 11.

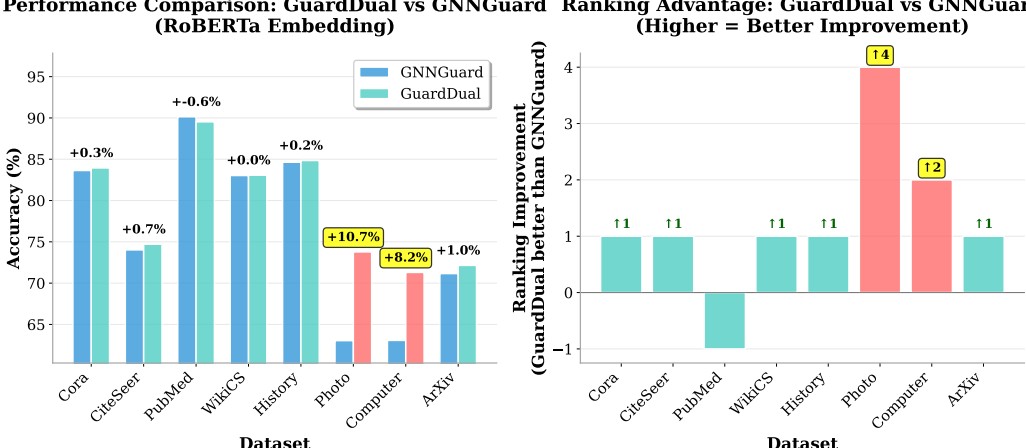

Figure 10: Comprehensive comparison between GuardDual and GNNGuard under RoBERTa embedding for structure attack defense. Left subplot shows absolute accuracy performance with Guard-Dual consistently outperforming GNNGuard across most datasets. Right subplot displays ranking improvements, where positive values indicate GuardDual's superior competitive position.

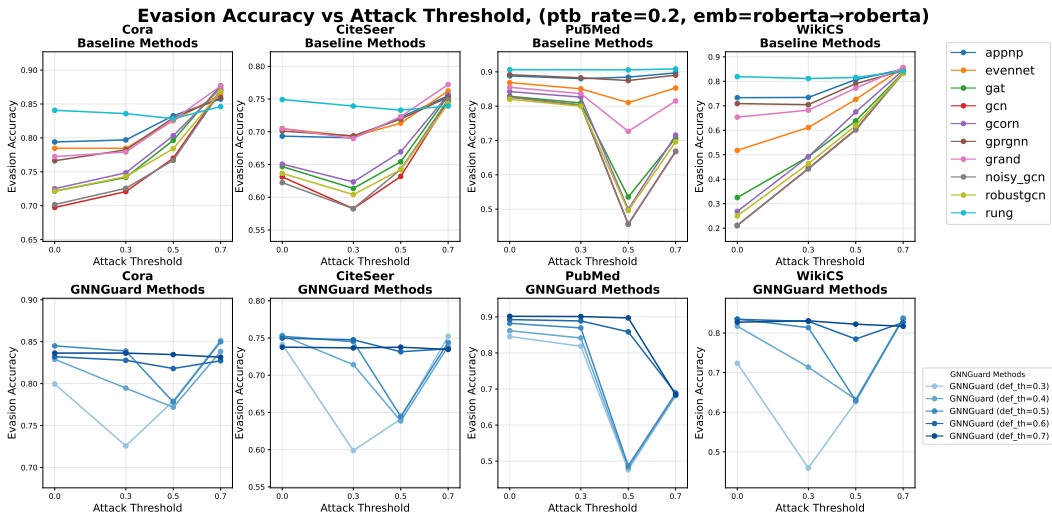

Figure 11: Evasion accuracy vs attack threshold for different defense methods with perturbation rate 0.2 and embedding types roberta.

Key observations are summarized as follows:

- **U-Shaped Performance Curve for GNNGuard**: GNNGuard exhibits a U-shaped trend in evasion accuracy with respect to PGDGuard's attack threshold $\epsilon$. When $\epsilon$ significantly deviates from GNNGuard's filtering threshold (either lower or higher), the attack's effectiveness diminishes. However, when $\epsilon$ closely aligns with GNNGuard's threshold, the attack achieves optimal evasion, indicating a trade-off: at lower $\epsilon$, PGDGuard's perturbations lack sufficient potency, while at higher $\epsilon$, GNNGuard's filtering mechanism effectively mitigates the attack.

- **Trade-Offs for Other Defense Methods**: While increasing the attack threshold $\epsilon$ may amplify PGDGuard's impact on GNNGuard, it generally enhances the performance of other defense methods. On datasets like Cora and WikiCS, attacks tailored to GNNGuard's threshold tend to weaken against alternative defenses, posing significant challenges for attackers aiming to generalize across diverse defense strategies.

Table 13: Performance (%) for PGD-structure on **GNNs** (GCN, GRAND, RUNG). Rows = attacker embeddings; columns = defender embeddings.

| Model / Dataset | Cora | | | CiteSeer | | | PubMed | | | WikiCS | | |
|---|---|---|---|---|---|---|---|---|---|---|---|---|
| | BoW | MiniLM | RoBERTa | BoW | MiniLM | RoBERTa | BoW | MiniLM | RoBERTa | BoW | MiniLM | RoBERTa |
| *GCN* | | | | | | | | | | | | |
| Attacker = BoW | 66.11 | 72.39 | 73.31 | 54.86 | 61.34 | 62.70 | 80.12 | 83.87 | 82.92 | 20.39 | 30.44 | 31.54 |
| Attacker = MiniLM | 70.11 | 62.98 | 68.39 | 54.60 | 47.13 | 54.18 | 82.60 | 81.57 | 83.55 | 33.25 | 25.37 | 28.25 |
| Attacker = RoBERTa | 71.34 | 70.85 | 69.43 | 63.95 | 63.74 | 63.58 | 82.21 | 82.29 | 82.73 | 25.12 | 21.50 | 20.86 |
| *GRAND* | | | | | | | | | | | | |
| Attacker = BoW | 71.89 | 78.60 | 79.89 | 64.00 | 70.22 | 70.74 | 82.45 | 83.88 | 86.41 | 48.57 | 66.75 | 71.86 |
| Attacker = MiniLM | 75.28 | 70.91 | 77.06 | 64.79 | 64.99 | 66.88 | 84.02 | 83.43 | 86.03 | 64.96 | 53.57 | 67.01 |
| Attacker = RoBERTa | 75.40 | 76.45 | 76.51 | 68.03 | 70.43 | 70.06 | 84.54 | 84.25 | 85.77 | 57.21 | 61.13 | 65.36 |
| *RUNG* | | | | | | | | | | | | |
| Attacker = BoW | 78.04 | 83.64 | 84.56 | 67.97 | 73.46 | 73.30 | 85.70 | 89.78 | 90.72 | 75.00 | 79.95 | 82.73 |
| Attacker = MiniLM | 78.04 | 83.70 | 83.83 | 67.92 | 73.35 | 73.93 | 85.87 | 89.53 | 90.61 | 76.18 | 80.26 | 82.53 |
| Attacker = RoBERTa | 79.09 | 83.58 | 84.32 | 69.04 | 74.14 | 73.93 | 86.06 | 89.56 | 90.72 | 75.94 | 80.14 | 82.39 |

Table 14: Perfomrance (%) for PGD-structure on **LLaGA** and **SFT-Neighbor**. Rows = attacker embeddings; columns = datasets.

| Model / Attacker | Cora | CiteSeer | PubMed | WikiCS |
|---|---|---|---|---|
| *LLaGA* | | | | |
| BoW | 75.21 | 66.35 | 87.38 | 66.99 |
| RoBERTa | 72.88 | 67.24 | 86.70 | 60.47 |
| *SFT-Neighbor* | | | | |
| BoW | 82.59 | 74.24 | 92.29 | 84.05 |
| RoBERTa | 81.24 | 71.84 | **94.72** | 86.19 |

- **Dataset-Specific Exceptions**: Despite the general trade-off, notable exceptions arise, particularly on PubMed at $\epsilon = 0.5$. Here, PGDGuard achieves superior attack performance across all methods. This is attributed to PubMed's fine-grained classification, where embedding similarities (metric in subplot b of Figure 8) for both intra-class and inter-class nodes cluster around 0.6. Selecting $\epsilon = 0.5$ effectively targets potentially harmful edges, highlighting the efficacy of dataset-specific, embedding-aware attack strategies.

These findings underscore the nuanced interplay between attack thresholds and defense mechanisms, emphasizing the importance of aligning attack strategies with dataset-specific embedding characteristics to maximize evasion effectiveness.

## H.2 ADAPTIVE ATTACKS ON EMBEDDING TRANSFERABILITY

**Setup.** In the main experiments, we initially used BoW as the victim's text embedding for structural evasion attacks. To model adaptive attackers and embedding-aware defenders, we vary the text encoder used the victim model in PGD among {BoW, MiniLM, RoBERTa} while the defender may defend with {BoW, MiniLM, RoBERTa}. We evaluate GCN, GRAND, RUNG, LLaGA, and SFT-Neighbor on datasets Cora, CiteSeer, PubMed, and WikiCS.

The results are shown in Table 13 and Table 14. We have the following findings:

- **Embedding Match Helps the Attacker**. Across models, the attack performance is highest when the attacker and defender use the same embedding. If embedding is mismatched, even from advanced to BoW, the attack performance significantly degrades. In fact, when the surrogate text encoder is LM, BoW as the defender's embedding can be strong.

- **Transfer is Stronger within LM-family.** MiniLM ↔ RoBERTa transfers better than BoW ↔ LM.

- **Transferability to GraphLLMs Varies among Datasets.** LLaGA, which takes RoBERTa as the text encoder, also suffers more if the attacker uses RoBERTa as the surrogate. However, for SFT, the results vary among datasets. This observation means that the effective text encoder for attackers could depend on the dataset's characteristics and the victim model.

# I   ABLATION STUDY FOR SFT VARIANTS

In this subsection, we conduct an ablation study to analyze the impact of different neighbor selection strategies and prompt templates in our SFT variants. We examine the effectiveness of degree-based neighbor selection versus random selection, and investigate the influence of incorporating label information in the prompting strategy.

## I.1   IMPACT OF DIFFERENT PROMPT TEMPLATES

Due to context length limitations, currently, we feed nodes' neighbors with the top degree as representatives in the prompt. In this subsection, we conduct an ablation study to examine the impact of randomly selecting neighbors.

### I.1.1   DEGREE-BASED VS RANDOM NEIGHBOR SELECTION

Table 15: Comparison between degree-based (SFT-neighbor) and random (SFT-rand) neighbor selection strategies across different attack scenarios. **Bolded values** indicate the better performance between the two methods for each dataset and scenario.

| Method | Cora | CiteSeer | PubMed | WikiCS | Instagram | Reddit | History |
|---|---|---|---|---|---|---|---|
| | | | *Clean Results on Inductive Split* | | | | |
| SFT-neighbor | **88.25 ± 1.29** | 76.17 ± 1.81 | **95.08 ± 0.51** | **87.75 ± 0.46** | 69.24 ± 0.54 | 66.74 ± 1.06 | **86.81 ± 0.77** |
| SFT-rand | 87.27 ± 2.30 | **77.27 ± 0.47** | 95.05 ± 0.20 | 87.72 ± 0.58 | **69.96 ± 0.72** | **68.46 ± 0.09** | 86.64 ± 0.45 |
| | | | *Structure Attack Results on Inductive Split* | | | | |
| SFT-neighbor | **82.35 ± 1.94** | **71.53 ± 1.86** | 94.73 ± 0.41 | **86.00 ± 1.40** | 68.25 ± 0.20 | 62.29 ± 3.31 | **85.98 ± 0.51** |
| SFT-rand | 79.09 ± 2.82 | 71.42 ± 0.48 | **94.79 ± 0.17** | 85.19 ± 0.68 | **68.50 ± 0.71** | **65.23 ± 1.16** | 84.77 ± 0.83 |
| | | | *Text Attack Results on Inductive Split* | | | | |
| SFT-neighbor | 75.65 ± 1.33 | 43.84 ± 2.14 | 72.27 ± 3.53 | **51.53 ± 1.24** | **64.33 ± 1.46** | 65.52 ± 1.24 | 64.84 ± 0.98 |
| SFT-rand | **78.60 ± 1.78** | **47.08 ± 2.76** | **73.22 ± 1.74** | 50.93 ± 0.82 | 61.71 ± 1.71 | **66.58 ± 0.54** | **68.83 ± 6.35** |

As shown in Table 15, the comparison between degree-based and random neighbor selection reveals distinct performance patterns across evaluation scenarios. Under clean conditions, SFT-neighbor demonstrates marginal advantages on most datasets. Interestingly, under text attack scenarios, SFT-rand exhibits superior robustness on several datasets, including Cora (78.60% vs 75.65%) and Cite-Seer (47.08% vs 43.84%). However, under structural attacks, SFT-rand is more vulnerable, exhibiting larger performance drops. In conclusion, trade-offs also exist between the two variants.

### I.1.2   LABEL INFORMATION IN PROMPTING

In (Wang et al., 2025b), incorporating label information in the prompt can yield better results for specific scenarios. We test the performance of such a prompt on clean datasets and against adversarial attacks. As shown in Table 16, incorporating explicit label information (SFT-neighbor-label) often leads to unstable performance, particularly on certain datasets like Instagram and Reddit, where severe degradation is observed across all scenarios. This lack of robustness and dataset-specific instability indicates potential overfitting or incompatibility with certain graph structures or data distributions. Consequently, label-enhanced prompting is not adopted due to its inconsistent performance.

## I.2   IMPACT OF DIFFERENT LLM BACKBONES

In the main paper, we use Mistral-7B as our LLM backbone as it shows the best performance in paper (Wu et al., 2025). In this subsection, we present the results of SFT with neighbor-aware prompts using different LLM backbones.

We use LLMs Mistral-7B (Jiang et al., 2023), Ministral-8B (Mistral AI Team, 2024), LLama3.1-8B (Dubey et al., 2024) and Qwen3-8B (Yang et al., 2025). The results are presented in Tables 17, 18, and 19.

The results demonstrate that:

Table 16: Comparison between standard degree-based selection (SFT-neighbor) and label-enhanced prompting (SFT-neighbor-label) across different attack scenarios. **Bolded values** indicate the better performance between the two methods for each dataset and scenario. Red values indicate significant performance degradation.

| Method | Cora | CiteSeer | PubMed | WikiCS | Instagram | Reddit | History |
|---|---|---|---|---|---|---|---|
| *Clean Results on Inductive Split* | | | | | | | |
| SFT-neighbor | **88.25 ± 1.29** | 76.17 ± 1.81 | **95.08 ± 0.51** | 87.75 ± 0.46 | **69.24 ± 0.54** | **66.74 ± 1.06** | **86.81 ± 0.77** |
| SFT-neighbor-label | 87.33 ± 2.50 | **77.07 ± 0.72** | 94.80 ± 0.44 | 87.75 ± 0.66 | 35.54 ± 3.54 | 14.08 ± 0.59 | 86.81 ± 0.69 |
| *Structure Attack Results on Inductive Split* | | | | | | | |
| SFT-neighbor | 82.35 ± 1.94 | 71.53 ± 1.86 | **94.73 ± 0.41** | 86.00 ± 1.40 | **68.25 ± 0.20** | **62.29 ± 3.31** | 85.98 ± 0.51 |
| SFT-neighbor-label | **82.54 ± 1.67** | **73.35 ± 1.03** | 94.41 ± 0.38 | **86.28 ± 0.15** | 45.84 ± 2.23 | 22.42 ± 2.01 | **86.00 ± 0.78** |
| *Text Attack Results on Inductive Split* | | | | | | | |
| SFT-neighbor | **75.65 ± 1.33** | **43.84 ± 2.14** | 72.27 ± 3.53 | **51.53 ± 1.24** | **64.33 ± 1.46** | **65.52 ± 1.24** | **64.84 ± 0.98** |
| SFT-neighbor-label | 64.51 ± 7.10 | 41.80 ± 0.77 | **74.97 ± 4.68** | 50.17 ± 0.58 | 34.64 ± 2.11 | 13.91 ± 0.59 | 64.58 ± 3.41 |

Table 17: Clean performance of SFT with neighbor-aware prompt in the inductive setting .

| LLM | cora | CiteSeer | pubmed | wikics | history |
|---|---|---|---|---|---|
| Mistral-7B | **88.25 ± 1.29** | 76.17 ± 1.81 | 95.08 ± 0.51 | **87.75 ± 0.46** | **86.81 ± 0.77** |
| Ministral-8B | 86.29 ± 0.91 | 76.70 ± 0.55 | **95.50 ± 0.08** | 87.28 ± 0.34 | 86.57 ± 0.85 |
| Llama3-8B | 86.10 ± 0.93 | **77.43 ± 1.18** | 95.38 ± 0.15 | 86.33 ± 0.15 | 86.68 ± 0.78 |
| Qwen3-8B | 85.55 ± 0.95 | 77.07 ± 1.49 | 95.18 ± 0.37 | 86.17 ± 0.91 | 86.08 ± 0.82 |

Table 18: SFT with neighbor prompt against structure attacks in the inductive setting.

| LLM | cora | CiteSeer | pubmed | wikics | history |
|---|---|---|---|---|---|
| Mistral-7B | **82.35 ± 1.94** | 71.53 ± 1.86 | 94.73 ± 0.41 | **86.00 ± 1.40** | 85.98 ± 0.51 |
| Ministral-8B | 79.64 ± 1.87 | 74.30 ± 0.98 | **95.23 ± 0.07** | 85.97 ± 0.91 | 86.06 ± 0.72 |
| Llama3-8B | 79.15 ± 1.12 | **74.34 ± 0.48** | 95.00 ± 0.14 | 84.98 ± 1.01 | **86.08 ± 0.69** |
| Qwen3-8B | 77.92 ± 1.68 | 72.94 ± 0.79 | 94.77 ± 0.49 | 84.73 ± 1.34 | 85.26 ± 0.70 |

Table 19: SFT with neighbor prompt against text attacks in the inductive setting.

| LLM | cora | CiteSeer | pubmed | wikics | history |
|---|---|---|---|---|---|
| Mistral-7B | **75.65 ± 1.33** | 43.84 ± 2.14 | **72.27 ± 3.53** | **51.53 ± 1.24** | **64.84 ± 0.98** |
| Ministral-8B | 68.76 ± 1.77 | 42.11 ± 1.02 | 71.02 ± 1.32 | 49.95 ± 0.44 | 55.15 ± 3.43 |
| Llama3-8B | 69.74 ± 3.81 | 43.37 ± 2.81 | 72.19 ± 1.67 | 49.58 ± 0.42 | 57.82 ± 3.37 |
| Qwen3-8B | 74.85 ± 2.33 | **44.56 ± 2.69** | 71.85 ± 0.56 | 49.44 ± 0.41 | 63.13 ± 3.19 |

- Mistral-7B consistently achieves the best performance across most datasets and settings, validating our choice of backbone in the main experiments

- Post-attack performance generally follows the same trends as clean performance, with models maintaining their relative advantages—for instance, Mistral-7B preserves its superiority on Cora across all attack scenarios

- Minor variations exist in specific contexts, such as on Cora and History datasets under textual attacks, where both Mistral-7B and Qwen3-8B demonstrate stronger robustness, though overall differences remain modest across backbones.

## J   FULL EXPERIMENT RESULTS

### J.1   OVERVIEW AND CLARIFICATION OF RANK-BASED EVALUATION

The full results are provided in Section J.2 and Section J.3. In Section J.4, we include results against a smaller perturb ratio. We can see that compared to the main experiments, the attack effectiveness degrades.

In the main paper, we rely primarily on rank-based evaluation for the following reasons:

- Per-dataset plots are dense and difficult to read within the limited space of the main paper. Lines with similar performance tend to mix up.

- Averaging raw accuracies across datasets can be misleading due to scale differences and missing results on large-scale datasets, whereas rank provides a normalized, dataset-agnostic comparison.

- Raw accuracies are heavily influenced by clean accuracy, while rank mitigates such biases. Rank-based evaluation thus offers a clearer and more balanced summary of robustness across datasets.

To recover the rank-related results in the main paper, the procedure is as follows:

1. For each dataset, all methods with valid performance values are collected. A dash in the tables ($-$) indicates a method was not applicable due to OOM or scalability issues.

2. Methods are sorted within each dataset to determine their rank.

3. The final rank is the average of a method's ranks across all datasets where it has values.

For example, a method with ranks $(1, 2, 3)$ has a final rank of 2, while a method with ranks $(1, 2, -)$ has a final rank of 1.5. Note that within each figure, only the methods included are considered during ranking.

### J.2   GNN AND RGNN RESULTS

#### J.2.1   CLEAN, INDUCTIVE

Results are in Tables 20, 22, 21, 23.

Table 20: Clean test accuracy under the inductive setting. (emb=BoW)

| Method | Cora | CiteSeer | PubMed | WikiCS | Instagram | Reddit | History | Photo | Computer | ArXiv | Avg Rank |
|---|---|---|---|---|---|---|---|---|---|---|---|
| GCN | $85.98 \pm 1.36$ | $\underline{73.93 \pm 1.94}$ | $86.84 \pm 0.25$ | $81.92 \pm 0.29$ | $63.05 \pm 0.49$ | $60.22 \pm 0.82$ | $81.97 \pm 0.13$ | $84.23 \pm 0.17$ | $87.75 \pm 0.23$ | $71.35 \pm 0.00$ | 5.40 |
| GAT | $86.22 \pm 1.86$ | $73.88 \pm 0.85$ | $86.72 \pm 0.21$ | $82.40 \pm 0.62$ | $64.67 \pm 1.37$ | $61.81 \pm 1.15$ | $\underline{82.07 \pm 0.63}$ | $\mathbf{84.75 \pm 0.05}$ | $88.09 \pm 0.22$ | $\underline{71.79 \pm 0.00}$ | 3.80 |
| APPNP | $86.22 \pm 1.40$ | $71.94 \pm 0.97$ | $87.01 \pm 0.21$ | $80.52 \pm 0.62$ | $63.10 \pm 1.59$ | $57.27 \pm 1.38$ | $81.19 \pm 0.43$ | $82.99 \pm 0.28$ | $\underline{88.11 \pm 0.16}$ | $71.27 \pm 0.00$ | 7.00 |
| GPRGNN | $83.95 \pm 1.57$ | $71.32 \pm 1.63$ | $\mathbf{87.80 \pm 0.02}$ | $81.23 \pm 0.32$ | $64.34 \pm 1.16$ | $60.03 \pm 0.72$ | $78.59 \pm 0.42$ | $81.65 \pm 0.66$ | $85.02 \pm 0.05$ | $65.67 \pm 0.00$ | 8.60 |
| RobustGCN | $86.22 \pm 2.01$ | $73.77 \pm 0.63$ | $86.86 \pm 0.36$ | $82.88 \pm 0.04$ | $\mathbf{66.74 \pm 0.34}$ | $58.06 \pm 1.18$ | $81.53 \pm 0.57$ | $84.03 \pm 0.32$ | $87.67 \pm 0.17$ | $68.20 \pm 0.00$ | 5.50 |
| GCORN | $83.33 \pm 1.48$ | $73.35 \pm 0.59$ | $86.73 \pm 0.35$ | $81.25 \pm 0.22$ | $64.98 \pm 0.45$ | $\mathbf{65.01 \pm 0.58}$ | $80.92 \pm 0.24$ | $82.04 \pm 0.08$ | $84.53 \pm 0.09$ | $69.86 \pm 0.00$ | 7.10 |
| NoisyGCN | $86.35 \pm 1.21$ | $73.25 \pm 1.36$ | $86.73 \pm 0.20$ | $81.85 \pm 0.31$ | $63.12 \pm 0.43$ | $61.84 \pm 0.31$ | $\mathbf{82.20 \pm 0.25}$ | $84.38 \pm 0.16$ | $87.75 \pm 0.11$ | $71.25 \pm 0.00$ | 5.00 |
| GRAND | $85.12 \pm 1.76$ | $\mathbf{74.19 \pm 1.41}$ | $\underline{87.75 \pm 0.45}$ | $81.49 \pm 0.19$ | $65.02 \pm 0.48$ | $\underline{62.50 \pm 1.78}$ | $79.65 \pm 0.62$ | $\underline{81.71 \pm 0.42}$ | $83.78 \pm 0.14$ | $68.23 \pm 0.00$ | 6.00 |
| SoftmedianGDC | $80.57 \pm 1.90$ | $73.35 \pm 0.38$ | $85.58 \pm 0.08$ | $78.48 \pm 0.32$ | $62.74 \pm 1.15$ | $61.33 \pm 0.50$ | $76.78 \pm 0.46$ | $-$ | $-$ | $-$ | 10.57 |
| EvenNet | $84.81 \pm 1.51$ | $72.52 \pm 0.70$ | $87.63 \pm 0.40$ | $\underline{83.01 \pm 0.36}$ | $65.24 \pm 0.79$ | $60.24 \pm 0.11$ | $81.01 \pm 0.59$ | $84.07 \pm 0.03$ | $87.86 \pm 0.18$ | $70.14 \pm 0.00$ | 5.40 |
| ElasticGNN | $\mathbf{86.59 \pm 1.97}$ | $73.77 \pm 0.07$ | $87.60 \pm 0.11$ | $\mathbf{83.11 \pm 0.49}$ | $\underline{65.58 \pm 1.04}$ | $61.62 \pm 0.43$ | $81.71 \pm 0.40$ | $84.05 \pm 0.08$ | $\mathbf{88.60 \pm 0.12}$ | $\mathbf{72.24 \pm 0.00}$ | 2.80 |
| GNNGuard | $82.29 \pm 1.23$ | $69.85 \pm 0.98$ | $86.24 \pm 0.32$ | $75.05 \pm 0.72$ | $63.95 \pm 0.64$ | $54.37 \pm 0.35$ | $78.05 \pm 0.38$ | $77.81 \pm 0.22$ | $75.87 \pm 0.35$ | $70.49 \pm 0.00$ | 10.80 |
| RUNG | $81.73 \pm 2.35$ | $71.06 \pm 0.27$ | $85.72 \pm 0.24$ | $75.00 \pm 0.25$ | $64.40 \pm 0.16$ | $57.63 \pm 0.60$ | $-$ | $-$ | $-$ | $-$ | 11.17 |

Table 21: Clean test accuracy under the inductive setting. (emb=Mistral-7B)

| Method | Cora | CiteSeer | PubMed | WikiCS | Instagram | Reddit | History | Photo | Computer | ArXiv | Avg Rank |
|---|---|---|---|---|---|---|---|---|---|---|---|
| GCN | $86.84 \pm 1.99$ | $75.44 \pm 0.63$ | $88.24 \pm 0.63$ | $83.43 \pm 0.36$ | $68.49 \pm 0.30$ | $65.39 \pm 5.04$ | $84.74 \pm 0.44$ | $85.64 \pm 0.34$ | $88.33 \pm 0.16$ | $74.18 \pm 0.00$ | 6.00 |
| GAT | $80.14 \pm 1.03$ | $71.58 \pm 0.77$ | $86.63 \pm 0.62$ | $66.28 \pm 4.25$ | $64.39 \pm 0.27$ | $61.14 \pm 0.58$ | $83.36 \pm 0.61$ | $82.83 \pm 1.88$ | $87.56 \pm 0.48$ | $72.91 \pm 0.00$ | 10.20 |
| APPNP | $84.69 \pm 1.20$ | $75.91 \pm 0.82$ | $92.02 \pm 0.22$ | $85.79 \pm 0.46$ | $69.02 \pm 0.18$ | $63.02 \pm 0.51$ | $84.88 \pm 0.56$ | $85.53 \pm 0.49$ | $\underline{88.51 \pm 0.40}$ | $\mathbf{76.22 \pm 0.00}$ | 4.40 |
| GPRGNN | $85.85 \pm 0.98$ | $\mathbf{77.12 \pm 0.38}$ | $\mathbf{92.44 \pm 0.22}$ | $86.50 \pm 0.65$ | $67.75 \pm 0.95$ | $\underline{65.56 \pm 0.51}$ | $84.82 \pm 0.38$ | $84.75 \pm 0.62$ | $87.84 \pm 0.16$ | $\underline{76.11 \pm 0.00}$ | 4.20 |
| RobustGCN | $58.61 \pm 8.78$ | $46.03 \pm 3.30$ | $45.01 \pm 7.14$ | $27.01 \pm 5.97$ | $64.26 \pm 0.53$ | $52.75 \pm 0.27$ | $56.16 \pm 0.43$ | $42.14 \pm 0.11$ | $25.89 \pm 0.18$ | $5.86 \pm 0.00$ | 11.60 |
| GCORN | $86.16 \pm 1.98$ | $76.33 \pm 0.78$ | $90.50 \pm 0.39$ | $85.69 \pm 0.39$ | $66.18 \pm 0.35$ | $\mathbf{69.28 \pm 0.88}$ | $\mathbf{85.26 \pm 0.31}$ | $\mathbf{86.36 \pm 0.27}$ | $87.67 \pm 0.25$ | $75.52 \pm 0.00$ | 4.50 |
| NoisyGCN | $\underline{87.95 \pm 1.30}$ | $75.18 \pm 0.27$ | $88.23 \pm 0.03$ | $84.00 \pm 0.29$ | $\underline{68.83 \pm 0.66}$ | $65.38 \pm 3.55$ | $84.74 \pm 0.43$ | $\underline{85.95 \pm 0.35}$ | $88.44 \pm 0.39$ | $74.29 \pm 0.00$ | 5.30 |
| GRAND | $86.29 \pm 1.45$ | $76.49 \pm 0.64$ | $90.80 \pm 0.33$ | $85.32 \pm 1.05$ | $68.25 \pm 0.10$ | $63.55 \pm 0.88$ | $84.57 \pm 0.57$ | $77.99 \pm 3.35$ | $79.59 \pm 1.60$ | $-$ | 6.56 |
| SoftmedianGDC | $82.35 \pm 1.60$ | $73.04 \pm 0.26$ | $91.70 \pm 0.18$ | $81.48 \pm 3.09$ | $-$ | $60.79 \pm 0.42$ | $84.50 \pm 0.61$ | $-$ | $-$ | $-$ | 9.33 |
| EvenNet | $85.79 \pm 1.04$ | $75.60 \pm 0.70$ | $90.77 \pm 0.07$ | $\mathbf{87.13 \pm 0.53}$ | $\mathbf{69.34 \pm 0.24}$ | $64.91 \pm 0.30$ | $\underline{84.90 \pm 0.53}$ | $85.00 \pm 0.29$ | $88.34 \pm 0.62$ | $74.12 \pm 0.00$ | 4.50 |
| ElasticGNN | $\mathbf{88.25 \pm 1.30}$ | $76.70 \pm 0.91$ | $89.33 \pm 0.46$ | $\underline{86.62 \pm 0.34}$ | $68.74 \pm 0.47$ | $49.95 \pm 0.31$ | $\underline{84.90 \pm 0.49}$ | $85.60 \pm 0.46$ | $\mathbf{88.96 \pm 0.44}$ | $76.09 \pm 0.00$ | 4.20 |
| GNNGuard | $85.49 \pm 0.74$ | $74.35 \pm 0.52$ | $90.20 \pm 0.23$ | $84.25 \pm 1.71$ | $\underline{69.12 \pm 0.21}$ | $59.48 \pm 7.12$ | $-$ | $-$ | $-$ | $-$ | 7.50 |
| RUNG | $82.90 \pm 1.81$ | $73.67 \pm 0.38$ | $\underline{92.34 \pm 0.06}$ | $83.53 \pm 0.65$ | $68.50 \pm 0.08$ | $63.60 \pm 0.49$ | $-$ | $-$ | $-$ | $-$ | 7.17 |

Table 22: Clean test accuracy under the inductive setting. (emb=MiniLM)

| Method | Cora | CiteSeer | PubMed | WikiCS | Instagram | Reddit | History | Photo | Computer | ArXiv | Avg Rank |
|---|---|---|---|---|---|---|---|---|---|---|---|
| GCN | 86.35 ± 2.07 | 75.76 ± 0.98 | 89.28 ± 0.51 | 83.81 ± 0.13 | 65.06 ± 0.68 | 66.98 ± 0.59 | 84.57 ± 0.36 | 86.04 ± 0.25 | 88.93 ± 0.11 | 73.86 ± 0.00 | 5.60 |
| GAT | 87.33 ± 1.58 | 74.87 ± 0.85 | 87.53 ± 0.08 | 83.47 ± 0.21 | 67.11 ± 0.28 | 65.00 ± 0.85 | 83.84 ± 0.53 | 86.14 ± 0.17 | 89.10 ± 0.17 | 73.31 ± 0.00 | 6.20 |
| APPNP | 84.93 ± 1.92 | 75.76 ± 0.87 | 89.22 ± 0.32 | 83.44 ± 0.56 | 66.05 ± 0.06 | 58.83 ± 0.60 | 84.73 ± 0.59 | 85.28 ± 0.16 | 89.29 ± 0.10 | 75.71 ± 0.00 | 5.90 |
| GPRGNN | 86.41 ± 1.67 | 75.60 ± 0.53 | 89.96 ± 0.26 | 82.41 ± 0.64 | 63.10 ± 0.88 | 59.08 ± 0.08 | 83.12 ± 0.50 | 83.77 ± 0.25 | 87.75 ± 0.15 | 71.97 ± 0.00 | 8.60 |
| RobustGCN | **87.95 ± 1.03** | 76.18 ± 1.05 | 88.07 ± 0.19 | 84.12 ± 0.05 | **67.28 ± 0.16** | 58.65 ± 0.21 | 84.86 ± 0.38 | 85.97 ± 0.21 | 88.83 ± 0.15 | 74.09 ± 0.00 | 5.00 |
| GCORN | 86.72 ± 2.04 | 76.07 ± 1.15 | 88.80 ± 0.03 | 83.51 ± 0.30 | 64.62 ± 0.61 | 67.08 ± 0.51 | 84.72 ± 0.47 | 84.37 ± 0.21 | 85.84 ± 0.06 | 74.35 ± 0.00 | 5.90 |
| NoisyGCN | 85.92 ± 1.77 | 76.07 ± 0.48 | 89.22 ± 0.18 | 84.07 ± 0.09 | 65.14 ± 0.54 | **67.19 ± 0.52** | 84.54 ± 0.38 | **86.15 ± 0.32** | 89.08 ± 0.20 | 73.57 ± 0.00 | 5.20 |
| GRAND | 86.41 ± 1.28 | **77.48 ± 1.74** | 88.15 ± 0.14 | 83.06 ± 0.04 | 65.46 ± 0.31 | 64.60 ± 0.49 | 83.03 ± 0.52 | 82.40 ± 0.46 | 83.49 ± 0.08 | 72.22 ± 0.00 | 7.90 |
| SoftmedianGDC | 86.47 ± 1.61 | 76.65 ± 0.34 | **90.31 ± 0.27** | 82.22 ± 0.49 | 61.60 ± 0.57 | 63.43 ± 0.13 | 83.64 ± 0.53 | – | – | – | 7.00 |
| EvenNet | 86.78 ± 1.26 | 75.76 ± 2.25 | 88.97 ± 0.07 | **84.76 ± 0.22** | 65.95 ± 0.81 | 61.46 ± 0.20 | 84.63 ± 0.46 | 85.56 ± 0.22 | 89.15 ± 0.14 | 73.78 ± 0.00 | 5.10 |
| ElasticGNN | 86.53 ± 1.98 | 77.06 ± 1.26 | 88.67 ± 0.20 | 84.54 ± 0.27 | 66.68 ± 0.34 | 63.53 ± 0.41 | **84.98 ± 0.54** | 85.70 ± 0.25 | **89.66 ± 0.09** | **75.82 ± 0.00** | 3.60 |
| GNNGuard | 82.84 ± 1.14 | 75.08 ± 0.46 | 88.88 ± 0.06 | 80.72 ± 0.61 | 60.95 ± 1.04 | 56.83 ± 0.60 | 82.92 ± 0.15 | 74.70 ± 0.25 | 79.01 ± 0.08 | 71.25 ± 0.00 | 11.60 |
| RUNG | 84.75 ± 1.65 | 73.46 ± 0.87 | 89.78 ± 0.14 | 81.48 ± 0.38 | 63.51 ± 0.38 | 58.72 ± 0.51 | – | – | – | – | 10.17 |

Table 23: Clean test accuracy under the inductive setting. (emb=RoBERTa)

| Method | Cora | CiteSeer | PubMed | WikiCS | Instagram | Reddit | History | Photo | Computer | ArXiv | Avg Rank |
|---|---|---|---|---|---|---|---|---|---|---|---|
| GCN | 87.76 ± 1.51 | 75.60 ± 1.09 | 88.76 ± 0.25 | 84.86 ± 0.61 | 66.75 ± 0.29 | 67.76 ± 0.74 | 85.00 ± 0.31 | 86.46 ± 0.26 | 89.24 ± 0.15 | 73.29 ± 0.00 | 6.20 |
| GAT | 87.70 ± 1.30 | 76.59 ± 0.45 | 88.14 ± 0.48 | 84.42 ± 0.12 | 67.80 ± 0.23 | 64.64 ± 0.22 | 84.66 ± 0.67 | 86.30 ± 0.17 | 89.70 ± 0.28 | 74.09 ± 0.00 | 5.70 |
| APPNP | 84.75 ± 1.61 | 75.76 ± 0.63 | **91.26 ± 0.22** | 85.03 ± 0.63 | 67.21 ± 0.13 | 59.99 ± 0.45 | 85.79 ± 0.57 | 85.70 ± 0.20 | 88.99 ± 0.14 | 76.00 ± 0.00 | 5.80 |
| GPRGNN | 86.29 ± 1.61 | 74.19 ± 0.45 | 91.05 ± 0.16 | 84.38 ± 1.00 | 64.26 ± 0.58 | 58.35 ± 0.67 | 84.68 ± 0.45 | 84.76 ± 0.23 | 88.23 ± 0.17 | 71.83 ± 0.00 | 9.70 |
| RobustGCN | 87.70 ± 1.28 | 74.82 ± 0.59 | 87.92 ± 0.12 | 84.37 ± 0.18 | **67.97 ± 0.08** | 63.09 ± 0.87 | 84.43 ± 0.43 | 84.79 ± 0.39 | 86.78 ± 0.14 | 73.73 ± 0.00 | 8.50 |
| GCORN | 87.52 ± 1.15 | 76.28 ± 0.15 | 89.83 ± 0.16 | 85.48 ± 0.27 | 65.17 ± 0.52 | **67.85 ± 0.48** | 85.28 ± 0.49 | 85.64 ± 0.05 | 87.51 ± 0.06 | 73.69 ± 0.00 | 6.30 |
| NoisyGCN | 88.01 ± 0.69 | 75.86 ± 0.51 | 88.75 ± 0.32 | 84.35 ± 0.61 | 67.12 ± 0.32 | 67.75 ± 0.78 | 84.98 ± 0.08 | 86.37 ± 0.22 | 89.23 ± 0.25 | 73.35 ± 0.00 | 6.50 |
| GRAND | **88.75 ± 0.94** | **77.85 ± 1.29** | 90.47 ± 0.24 | **85.90 ± 0.31** | 67.64 ± 0.26 | 65.83 ± 0.49 | **86.00 ± 0.36** | 85.54 ± 0.22 | 88.13 ± 0.02 | 75.85 ± 0.00 | 3.80 |
| SoftmedianGDC | 85.36 ± 2.13 | 75.71 ± 0.90 | 90.76 ± 0.22 | 83.53 ± 0.94 | 65.53 ± 0.50 | 62.91 ± 0.19 | 84.78 ± 0.47 | – | – | – | 8.71 |
| EvenNet | 86.35 ± 1.18 | 75.86 ± 0.71 | 90.74 ± 0.15 | **85.95 ± 0.63** | 67.68 ± 0.65 | 61.65 ± 0.79 | 85.63 ± 0.40 | **86.55 ± 0.16** | 89.51 ± 0.21 | 74.75 ± 0.00 | 4.20 |
| ElasticGNN | **88.31 ± 1.51** | 75.76 ± 0.52 | 89.89 ± 0.20 | 85.48 ± 0.25 | 67.64 ± 0.80 | 64.06 ± 0.87 | 85.81 ± 0.57 | **90.07 ± 0.08** | **76.22 ± 0.00** | | 3.80 |
| GNNGuard | 83.64 ± 1.94 | 74.03 ± 0.53 | 90.21 ± 0.41 | 83.03 ± 0.53 | 64.09 ± 1.23 | 58.58 ± 0.33 | 84.65 ± 0.49 | 83.07 ± 0.08 | 82.38 ± 0.06 | 71.16 ± 0.00 | 12.10 |
| RUNG | 84.93 ± 1.46 | 73.30 ± 1.19 | 90.72 ± 0.25 | 84.37 ± 0.44 | 66.43 ± 0.27 | 58.80 ± 0.97 | – | – | – | – | 10.17 |
| Guardual | 84.26 ± 1.71 | 74.71 ± 1.22 | 89.83 ± 0.29 | 84.54 ± 0.34 | 65.53 ± 0.32 | 62.24 ± 0.22 | 85.01 ± 0.28 | 79.78 ± 0.17 | 77.39 ± 0.31 | 72.17 ± 0.00 | 9.90 |

### J.2.2 CLEAN, TRANSDUCTIVE

Results are in Tables 24, 25, 26.

Table 24: Clean test accuracy under the transductive setting. (ptb_rate=0.3, atk_emb=BoW, def_emb=BoW)

| Method | Cora | CiteSeer | PubMed | WikiCS | Instagram | Reddit | History | Avg Rank |
|---|---|---|---|---|---|---|---|---|
| GCN | 83.19 ± 0.75 | 71.32 ± 0.83 | 85.45 ± 0.24 | 80.01 ± 0.29 | 65.97 ± 0.09 | 61.84 ± 0.49 | 80.14 ± 0.25 | 6.86 |
| GAT | 82.99 ± 1.03 | 72.30 ± 0.56 | 85.38 ± 0.21 | 79.76 ± 0.35 | 65.41 ± 0.06 | 60.73 ± 0.30 | 80.67 ± 0.12 | 7.86 |
| APPNP | 83.69 ± 0.30 | 72.25 ± 1.10 | **87.34 ± 0.08** | 80.49 ± 0.33 | 65.78 ± 0.10 | 60.78 ± 0.66 | 80.84 ± 0.06 | 4.14 |
| GPRGNN | 83.06 ± 0.73 | 72.56 ± 0.65 | 86.48 ± 0.24 | 79.80 ± 0.14 | **66.12 ± 0.23** | 60.37 ± 1.08 | 80.49 ± 0.05 | 5.43 |
| RobustGCN | 83.77 ± 0.80 | 71.64 ± 0.66 | 86.45 ± 0.30 | **81.15 ± 0.14** | 65.94 ± 0.31 | 59.29 ± 0.82 | **80.95 ± 0.15** | 5.43 |
| GCORN | 77.40 ± 0.60 | 69.77 ± 0.58 | 84.71 ± 0.32 | 78.87 ± 0.33 | 65.43 ± 0.45 | **66.34 ± 0.08** | 78.36 ± 0.02 | 11.57 |
| NoisyGCN | 83.19 ± 0.12 | 72.36 ± 0.94 | 85.75 ± 0.42 | 79.74 ± 0.35 | 65.77 ± 0.15 | 63.83 ± 0.13 | 80.34 ± 0.32 | 6.00 |
| GRAND | **83.97 ± 0.92** | 71.71 ± 0.14 | 87.16 ± 0.30 | 80.71 ± 0.38 | 65.94 ± 0.27 | 61.58 ± 1.33 | 80.35 ± 0.27 | 4.43 |
| SoftmedianGDC | 82.77 ± 1.35 | 71.99 ± 0.28 | 85.15 ± 0.07 | 74.33 ± 0.49 | 65.98 ± 0.40 | 59.67 ± 0.52 | 78.04 ± 0.08 | 10.57 |
| EvenNet | 82.88 ± 0.56 | 71.70 ± 0.70 | 86.50 ± 0.24 | 80.71 ± 0.31 | 65.77 ± 0.72 | 64.61 ± 0.68 | 80.48 ± 0.07 | 5.57 |
| ElasticGNN | 83.19 ± 0.17 | **73.14 ± 0.40** | 87.00 ± 0.17 | 80.18 ± 0.27 | 65.41 ± 0.37 | 60.58 ± 0.47 | 80.67 ± 0.07 | 5.43 |
| GNNGuard | 80.31 ± 0.37 | 71.20 ± 0.28 | 86.13 ± 0.26 | 77.08 ± 0.28 | 64.26 ± 0.79 | 58.79 ± 0.43 | 79.54 ± 0.18 | 12.71 |
| RUNG | 81.91 ± 0.59 | 72.20 ± 0.80 | 85.57 ± 0.30 | 78.75 ± 0.47 | 65.55 ± 0.23 | 60.24 ± 0.11 | – | 10.17 |
| Cosine-GCN | 79.48 ± 1.38 | 69.70 ± 0.92 | 84.03 ± 0.26 | 77.61 ± 0.47 | 64.78 ± 0.50 | 59.07 ± 0.36 | 78.26 ± 0.21 | 14.57 |
| Jaccard-GCN | 80.60 ± 1.25 | 69.98 ± 0.82 | 84.12 ± 0.34 | 79.11 ± 0.29 | 65.66 ± 0.13 | 61.77 ± 0.26 | 78.44 ± 0.19 | 10.86 |
| Stable | 79.97 ± 0.45 | 68.29 ± 0.67 | 84.00 ± 0.43 | 78.94 ± 0.55 | 64.66 ± 0.51 | 63.32 ± 0.31 | – | 12.83 |
| ProGNN | 77.79 ± 0.19 | 70.13 ± 0.93 | – | – | – | – | – | 14.50 |

### J.2.3 STRUCTURE ATTACK, INDUCTIVE

Results are in Tables 27, 29, 28, 30.

### J.2.4 STRUCTURE ATTACK, TRANSDUCTIVE

Results are in Tables 31, 32, 33.

### J.2.5 TEXT ATTACK, INDUCTIVE

Results are in Tables 34, 35, 36.

### J.2.6 TEXT ATTACK, TRANSDUCTIVE

Results are in Tables 37, 38, 39.

Table 25: Clean test accuracy under the transductive setting. (ptb_rate=0.3, atk_emb=BoW, def_emb=MiniLM)

| Method | Cora | CiteSeer | PubMed | WikiCS | Instagram | Reddit | History | Avg Rank |
|---|---|---|---|---|---|---|---|---|
| GCN | 84.34 ± 0.34 | 74.32 ± 0.88 | 86.78 ± 0.48 | 81.31 ± 0.05 | 65.67 ± 0.33 | 65.81 ± 0.63 | 82.94 ± 0.11 | 8.43 |
| GAT | 83.97 ± 1.30 | 73.86 ± 0.61 | 85.94 ± 0.35 | 81.85 ± 0.31 | 65.56 ± 0.07 | 63.31 ± 0.86 | 82.65 ± 0.01 | 10.29 |
| APPNP | 84.33 ± 0.79 | 74.43 ± 0.36 | **88.36 ± 0.26** | 81.59 ± 0.28 | 65.68 ± 0.44 | 62.54 ± 0.09 | 83.44 ± 0.10 | 5.14 |
| GPRGNN | 84.31 ± 0.17 | 73.98 ± 0.91 | 88.04 ± 0.21 | 81.45 ± 0.23 | 65.21 ± 0.30 | 62.34 ± 0.25 | 82.61 ± 0.09 | 10.00 |
| RobustGCN | 84.05 ± 0.51 | 75.10 ± 0.65 | 86.92 ± 0.12 | 81.82 ± 0.30 | **66.14 ± 0.41** | 64.05 ± 0.29 | 83.18 ± 0.06 | 5.86 |
| GCORN | 82.77 ± 0.80 | 74.94 ± 0.65 | 86.99 ± 0.14 | 81.07 ± 0.35 | 65.75 ± 0.15 | 67.37 ± 0.29 | 83.12 ± 0.02 | 7.86 |
| NoisyGCN | 83.73 ± 0.44 | 74.11 ± 0.59 | 87.15 ± 0.23 | 81.42 ± 0.07 | 65.46 ± 0.01 | **67.52 ± 0.25** | 82.84 ± 0.12 | 9.00 |
| GRAND | 84.33 ± 0.53 | **75.82 ± 0.49** | 87.38 ± 0.16 | 81.69 ± 0.22 | 65.51 ± 0.24 | 65.02 ± 0.83 | 83.10 ± 0.18 | 5.86 |
| SoftmedianGDC | 84.42 ± 0.25 | 75.47 ± 0.45 | 88.16 ± 0.30 | 81.04 ± 0.38 | 65.88 ± 0.28 | 63.72 ± 0.29 | **83.54 ± 0.10** | 4.57 |
| EvenNet | 84.39 ± 0.60 | 74.75 ± 0.99 | 87.45 ± 0.23 | 82.11 ± 0.59 | 65.60 ± 0.23 | 62.01 ± 0.42 | 83.28 ± 0.07 | 5.86 |
| ElasticGNN | **84.63 ± 0.15** | 74.43 ± 0.36 | 87.69 ± 0.05 | 81.47 ± 0.23 | 65.77 ± 0.31 | 62.54 ± 0.14 | 83.24 ± 0.05 | 5.43 |
| GNNGuard | 82.71 ± 0.29 | 74.36 ± 0.58 | 87.37 ± 0.10 | 81.55 ± 0.01 | 64.79 ± 0.67 | 59.29 ± 0.57 | 82.75 ± 0.26 | 11.57 |
| RUNG | 82.19 ± 0.26 | 73.68 ± 0.20 | 87.76 ± 0.26 | 81.37 ± 0.01 | 65.02 ± 0.53 | 61.51 ± 0.11 | – | 12.67 |
| Cosine-GCN | 82.02 ± 0.81 | 74.19 ± 0.34 | 86.30 ± 0.22 | 81.36 ± 0.09 | 65.45 ± 0.38 | 59.37 ± 0.08 | 82.50 ± 0.08 | 13.43 |
| Jaccard-GCN | 83.42 ± 0.19 | 73.85 ± 0.73 | 86.30 ± 0.26 | 80.94 ± 0.16 | 65.67 ± 0.41 | 66.10 ± 0.33 | 82.30 ± 0.06 | 11.57 |
| Stable | 84.09 ± 0.19 | 73.68 ± 1.06 | 86.84 ± 0.23 | **82.53 ± 0.28** | 65.33 ± 0.54 | 66.21 ± 0.24 | – | 8.83 |
| ProGNN | 83.20 ± 0.51 | 75.34 ± 0.72 | – | – | – | – | – | 8.00 |

Table 26: Clean test accuracy under the transductive setting. (ptb_rate=0.3, atk_emb=BoW, def_emb=RoBERTa)

| Method | Cora | CiteSeer | PubMed | WikiCS | Instagram | Reddit | History | Avg Rank |
|---|---|---|---|---|---|---|---|---|
| GCN | 84.03 ± 0.51 | 73.98 ± 0.70 | 87.39 ± 0.11 | 81.85 ± 0.46 | 66.68 ± 0.20 | 63.41 ± 0.12 | 83.55 ± 0.20 | 10.57 |
| GAT | 84.25 ± 0.48 | 74.19 ± 0.40 | 86.20 ± 0.20 | 82.05 ± 0.25 | 66.45 ± 0.24 | 62.91 ± 0.19 | 83.33 ± 0.21 | 12.29 |
| APPNP | 85.03 ± 0.24 | 74.74 ± 0.22 | **89.13 ± 0.04** | 82.51 ± 0.55 | **67.48 ± 0.11** | 63.80 ± 0.15 | 84.43 ± 0.20 | 3.43 |
| GPRGNN | 85.05 ± 0.36 | 74.42 ± 0.50 | 89.06 ± 0.28 | 82.17 ± 0.44 | 67.26 ± 0.13 | 63.74 ± 0.12 | 84.05 ± 0.21 | 4.86 |
| RobustGCN | 83.54 ± 0.91 | 74.55 ± 1.12 | 87.27 ± 0.07 | 81.83 ± 0.19 | 66.62 ± 0.25 | 62.73 ± 0.74 | 83.20 ± 0.09 | 12.43 |
| GCORN | 84.34 ± 0.61 | 74.67 ± 1.32 | 87.44 ± 0.05 | 82.17 ± 0.15 | 66.46 ± 0.30 | **67.60 ± 0.18** | 83.81 ± 0.10 | 7.86 |
| NoisyGCN | 84.59 ± 0.75 | 73.69 ± 0.77 | 87.29 ± 0.23 | 82.12 ± 0.18 | 66.66 ± 0.61 | 63.87 ± 0.14 | 83.55 ± 0.16 | 9.71 |
| GRAND | **85.16 ± 0.61** | **76.06 ± 0.62** | 89.11 ± 0.08 | **84.16 ± 0.31** | 66.94 ± 0.47 | 66.34 ± 0.61 | **84.93 ± 0.09** | 2.00 |
| SoftmedianGDC | 83.56 ± 0.61 | 74.11 ± 0.43 | 88.48 ± 0.21 | 80.22 ± 0.67 | 66.93 ± 0.21 | 63.10 ± 0.40 | 84.30 ± 0.13 | 9.43 |
| EvenNet | 84.06 ± 1.02 | 74.94 ± 0.60 | 88.99 ± 0.14 | 83.15 ± 0.53 | 67.05 ± 0.16 | 63.23 ± 0.51 | 83.90 ± 0.14 | 5.29 |
| ElasticGNN | 84.88 ± 0.19 | 74.41 ± 1.05 | 88.28 ± 0.17 | 82.65 ± 0.23 | 66.87 ± 0.20 | 63.17 ± 0.13 | 83.99 ± 0.18 | 6.57 |
| GNNGuard | 83.22 ± 0.55 | 73.81 ± 0.50 | 87.98 ± 0.29 | 82.47 ± 0.19 | 67.23 ± 0.31 | 59.39 ± 0.39 | 83.87 ± 0.15 | 10.14 |
| RUNG | 83.42 ± 0.64 | 75.35 ± 0.88 | 88.86 ± 0.26 | 82.64 ± 0.33 | 66.82 ± 0.27 | 62.85 ± 0.17 | – | 8.17 |
| Cosine-GCN | 83.25 ± 1.08 | 73.43 ± 0.61 | 87.05 ± 0.16 | 82.60 ± 0.63 | 66.96 ± 0.21 | 59.37 ± 0.49 | 83.75 ± 0.20 | 11.43 |
| Jaccard-GCN | 83.54 ± 0.77 | 72.98 ± 0.42 | 87.04 ± 0.15 | 81.47 ± 0.42 | 66.70 ± 0.50 | 63.41 ± 0.12 | 83.18 ± 0.05 | 12.86 |
| Stable | 84.40 ± 0.80 | 72.70 ± 1.30 | 86.66 ± 0.27 | 82.65 ± 0.36 | 66.75 ± 0.26 | 65.33 ± 0.58 | – | 9.00 |
| ProGNN | 83.57 ± 0.79 | 74.91 ± 0.55 | – | – | – | – | – | 7.50 |

Table 27: Accuracy under the inductive/evasion setting against structural attack. (ptb_rate=0.2, atk_emb=BoW, def_emb=BoW)

| Method | Cora | CiteSeer | PubMed | WikiCS | Instagram | Reddit | History | Photo | Computer | ArXiv | Avg Rank |
|---|---|---|---|---|---|---|---|---|---|---|---|
| GCN | 66.11 ± 2.48 | 54.86 ± 1.00 | 80.12 ± 0.71 | 20.39 ± 0.56 | 7.11 ± 0.81 | 50.47 ± 3.28 | 41.70 ± 0.39 | 38.88 ± 2.42 | 35.95 ± 0.33 | 32.86 ± 0.00 | 11.60 |
| GAT | 67.53 ± 2.03 | 57.31 ± 1.07 | 80.18 ± 0.58 | 39.47 ± 1.08 | 45.03 ± 5.36 | 53.83 ± 3.18 | 49.44 ± 1.63 | 51.95 ± 1.37 | 44.08 ± 0.03 | 39.62 ± 0.00 | 8.10 |
| APPNP | 69.13 ± 2.82 | 64.37 ± 0.99 | 83.75 ± 0.14 | 57.37 ± 1.22 | 55.61 ± 2.66 | 52.40 ± 2.04 | 59.33 ± 0.76 | 55.21 ± 0.26 | 41.06 ± 0.19 | 43.12 ± 0.00 | 5.40 |
| GPRGNN | 71.83 ± 1.22 | 62.49 ± 0.60 | 82.53 ± 0.25 | 57.64 ± 0.86 | 60.96 ± 1.02 | **64.21 ± 1.65** | 64.79 ± 0.97 | **59.31 ± 0.10** | 50.72 ± 0.45 | **50.18 ± 0.00** | 2.90 |
| RobustGCN | 69.99 ± 2.80 | 58.25 ± 1.48 | 80.33 ± 0.61 | 31.14 ± 0.82 | 60.89 ± 0.47 | 52.74 ± 1.37 | 51.27 ± 0.75 | 48.72 ± 0.90 | 40.05 ± 0.94 | 41.67 ± 0.00 | 7.90 |
| GCORN | 68.57 ± 2.52 | 58.52 ± 0.97 | 81.69 ± 0.50 | 34.16 ± 0.21 | 32.63 ± 2.84 | 59.81 ± 1.73 | 41.51 ± 0.77 | 39.00 ± 0.67 | 37.12 ± 0.15 | 31.39 ± 0.00 | 8.90 |
| NoisyGCN | 66.24 ± 2.87 | 54.49 ± 0.73 | 80.27 ± 0.37 | 20.38 ± 0.57 | 60.86 ± 0.55 | 55.26 ± 1.17 | 41.41 ± 0.76 | 39.05 ± 1.28 | 36.61 ± 0.33 | 34.48 ± 0.00 | 10.90 |
| GRAND | 71.89 ± 3.25 | 64.00 ± 0.93 | 82.45 ± 0.08 | 48.57 ± 1.07 | **64.15 ± 0.37** | 58.74 ± 3.02 | 47.45 ± 0.66 | 49.11 ± 0.74 | 45.02 ± 0.16 | 49.90 ± 0.00 | 4.60 |
| SoftmedianGDC | 74.11 ± 2.93 | 64.32 ± 1.07 | 82.35 ± 0.25 | 72.66 ± 0.24 | 55.28 ± 1.11 | 61.88 ± 0.92 | **69.80 ± 0.49** | – | – | – | 3.86 |
| EvenNet | 70.42 ± 2.27 | 61.86 ± 1.03 | 82.47 ± 0.33 | 32.88 ± 1.56 | 62.38 ± 1.38 | 60.10 ± 1.03 | 61.94 ± 0.41 | 54.04 ± 0.65 | 46.07 ± 0.72 | 38.01 ± 0.00 | 5.10 |
| ElasticGNN | 68.51 ± 2.73 | 56.43 ± 0.00 | 81.07 ± 0.64 | 37.33 ± 0.24 | 39.99 ± 6.61 | 45.74 ± 0.54 | 60.02 ± 0.80 | 55.96 ± 0.40 | 40.43 ± 0.38 | 43.43 ± 0.00 | 7.60 |
| GNNGuard | 67.16 ± 2.91 | **68.03 ± 0.90** | 84.56 ± 0.14 | **75.02 ± 0.71** | 63.57 ± 0.58 | 53.91 ± 0.30 | 53.70 ± 0.28 | 46.16 ± 0.33 | 44.98 ± 0.28 | 39.25 ± 0.00 | 5.00 |
| RUNG | **78.04 ± 2.22** | 67.97 ± 1.37 | **85.70 ± 0.25** | 75.00 ± 0.25 | 62.42 ± 0.08 | 55.41 ± 0.58 | – | – | – | – | 2.50 |

Table 28: Accuracy under the inductive/evasion setting against structural attack. (ptb_rate=0.2, atk_emb=BoW, def_emb=Mistral-7B)

| Method | Cora | CiteSeer | PubMed | WikiCS | Instagram | Reddit | History | Photo | Computer | ArXiv | Avg Rank |
|---|---|---|---|---|---|---|---|---|---|---|---|
| GCN | 73.06 ± 2.89 | 63.06 ± 1.54 | 82.33 ± 1.07 | 34.86 ± 1.97 | 62.23 ± 1.33 | 62.90 ± 9.60 | 64.32 ± 1.25 | 55.53 ± 0.58 | 51.20 ± 0.30 | 47.70 ± 0.00 | 8.10 |
| GAT | 67.59 ± 1.23 | 61.39 ± 1.56 | 81.65 ± 1.16 | 38.33 ± 3.51 | 64.23 ± 0.56 | 57.27 ± 0.44 | 65.34 ± 1.52 | 54.54 ± 0.27 | 51.71 ± 1.72 | 49.98 ± 0.00 | 8.60 |
| APPNP | 78.54 ± 0.71 | 69.70 ± 1.09 | 90.35 ± 0.26 | 79.27 ± 1.00 | 68.52 ± 0.35 | 60.18 ± 0.63 | 76.83 ± 0.83 | 64.66 ± 1.12 | 50.41 ± 1.33 | **59.74 ± 0.00** | 3.50 |
| GPRGNN | 80.57 ± 1.81 | 67.40 ± 1.05 | 88.59 ± 0.74 | 75.64 ± 2.00 | 65.78 ± 1.72 | 69.00 ± 0.22 | 75.27 ± 1.04 | **67.23 ± 1.51** | 47.75 ± 0.83 | **63.55 ± 0.00** | 3.80 |
| RobustGCN | 52.21 ± 7.89 | 41.17 ± 2.86 | 44.91 ± 6.81 | 25.12 ± 3.15 | 64.17 ± 0.42 | 53.50 ± 0.29 | 56.18 ± 0.42 | 42.14 ± 0.11 | 25.89 ± 0.18 | 5.86 ± 0.00 | 11.20 |
| GCORN | 72.69 ± 2.23 | 63.06 ± 0.41 | 84.22 ± 0.78 | 33.45 ± 0.25 | 58.89 ± 1.76 | **71.78 ± 3.75** | 57.14 ± 1.16 | 53.32 ± 0.54 | 44.48 ± 0.14 | 43.08 ± 0.00 | 8.90 |
| NoisyGCN | 73.19 ± 1.55 | 62.64 ± 0.60 | 82.20 ± 0.65 | 34.96 ± 0.61 | 61.71 ± 0.18 | 60.23 ± 9.70 | 64.24 ± 1.06 | 56.87 ± 1.04 | 51.36 ± 0.99 | 48.39 ± 0.00 | 8.20 |
| GRAND | 78.72 ± 1.40 | 68.97 ± 0.38 | 85.75 ± 0.26 | 69.40 ± 1.64 | 65.89 ± 0.26 | 57.97 ± 1.01 | 73.79 ± 0.70 | 64.17 ± 2.55 | **61.53 ± 1.13** | – | 4.67 |
| SoftmedianGDC | 77.12 ± 1.63 | 67.92 ± 0.85 | 88.31 ± 0.25 | 77.08 ± 2.93 | – | 59.37 ± 0.30 | **76.97 ± 0.47** | – | – | – | 4.83 |
| EvenNet | 79.21 ± 1.88 | 68.60 ± 1.31 | 87.26 ± 0.25 | 63.04 ± 1.29 | **69.24 ± 0.27** | 65.92 ± 0.10 | 69.05 ± 0.50 | 56.15 ± 1.44 | 52.42 ± 1.95 | 51.32 ± 0.00 | 4.10 |
| ElasticGNN | 76.01 ± 2.11 | 63.17 ± 0.44 | 83.50 ± 0.75 | 57.28 ± 3.23 | 62.01 ± 3.46 | 49.95 ± 0.31 | 68.02 ± 0.49 | 62.22 ± 0.21 | 51.74 ± 1.08 | 54.03 ± 0.00 | 7.10 |
| GNNGuard | 73.37 ± 1.06 | 71.89 ± 0.30 | 84.59 ± 0.73 | 80.83 ± 1.86 | 67.56 ± 0.30 | 55.14 ± 5.73 | – | – | – | – | 5.67 |
| RUNG | **82.41 ± 1.66** | **73.56 ± 0.45** | **92.33 ± 0.08** | 83.55 ± 0.56 | 68.69 ± 0.13 | 60.14 ± 1.15 | – | – | – | – | 2.17 |

Table 29: Accuracy under the inductive/evasion setting against structural attack. (ptb_rate=0.2, atk_emb=BoW, def_emb=MiniLM)

| Method | Cora | CiteSeer | PubMed | WikiCS | Instagram | Reddit | History | Photo | Computer | ArXiv | Avg Rank |
|---|---|---|---|---|---|---|---|---|---|---|---|
| GCN | 72.39 ± 2.45 | 61.34 ± 1.31 | 83.87 ± 0.87 | 30.44 ± 1.40 | 57.58 ± 0.72 | 65.79 ± 1.80 | 64.83 ± 0.83 | 58.03 ± 0.53 | 55.90 ± 0.31 | 41.35 ± 0.00 | 9.60 |
| GAT | 73.19 ± 1.28 | 61.60 ± 0.64 | 82.34 ± 0.70 | 47.83 ± 0.63 | 62.71 ± 0.53 | 64.50 ± 2.28 | 66.16 ± 1.46 | 58.86 ± 0.77 | 54.95 ± 0.74 | 48.70 ± 0.00 | 8.60 |
| APPNP | 80.87 ± 2.20 | 74.66 ± 0.66 | 87.32 ± 0.39 | 76.79 ± 0.71 | 63.30 ± 0.34 | 57.28 ± 0.80 | 78.18 ± 0.31 | 65.51 ± 0.25 | 54.42 ± 0.59 | 57.86 ± 0.00 | 4.50 |
| GPRGNN | 77.86 ± 2.42 | 70.69 ± 0.22 | 87.17 ± 0.12 | 71.37 ± 0.74 | 61.41 ± 0.79 | 58.05 ± 0.17 | 77.44 ± 0.39 | 66.16 ± 0.35 | 60.50 ± 0.48 | 64.18 ± 0.00 | 4.50 |
| RobustGCN | 73.31 ± 2.70 | 62.70 ± 1.63 | 82.55 ± 0.52 | 34.20 ± 0.96 | 63.86 ± 0.50 | 52.26 ± 0.38 | 65.76 ± 0.76 | 58.02 ± 0.68 | 55.38 ± 0.64 | 50.41 ± 0.00 | 9.10 |
| GCORN | 74.29 ± 2.02 | 62.23 ± 0.89 | 83.01 ± 0.54 | 36.32 ± 1.47 | 55.29 ± 2.25 | 67.74 ± 2.32 | 59.00 ± 0.87 | 53.73 ± 0.63 | 46.66 ± 0.40 | 38.91 ± 0.00 | 9.70 |
| NoisyGCN | 72.82 ± 2.34 | 61.55 ± 0.85 | 83.77 ± 0.61 | 30.49 ± 1.29 | 55.28 ± 2.89 | 68.01 ± 2.17 | 65.27 ± 0.68 | 58.19 ± 0.68 | 56.79 ± 0.15 | 41.32 ± 0.00 | 9.20 |
| GRAND | 78.60 ± 1.34 | 70.22 ± 1.22 | 83.88 ± 0.36 | 66.75 ± 0.42 | 64.32 ± 0.38 | 65.06 ± 1.18 | 72.45 ± 0.62 | 65.91 ± 0.14 | 61.17 ± 0.27 | 57.13 ± 0.00 | 4.40 |
| SoftmedianGDC | 78.66 ± 2.83 | 70.48 ± 0.53 | 86.61 ± 0.26 | 77.43 ± 0.34 | 59.22 ± 0.67 | 66.16 ± 0.80 | 77.78 ± 0.43 | – | – | – | 4.57 |
| EvenNet | 78.41 ± 2.03 | 68.86 ± 1.29 | 84.90 ± 0.17 | 55.20 ± 1.07 | 64.18 ± 0.86 | 57.46 ± 1.19 | 69.13 ± 0.28 | 60.29 ± 0.29 | 58.03 ± 0.59 | 48.64 ± 0.00 | 6.10 |
| ElasticGNN | 73.74 ± 1.80 | 63.01 ± 1.67 | 82.94 ± 0.51 | 55.26 ± 0.68 | 56.32 ± 2.31 | 53.90 ± 3.37 | 68.84 ± 0.67 | 63.10 ± 0.35 | 54.48 ± 0.63 | 55.27 ± 0.00 | 8.40 |
| GNNGuard | 82.84 ± 1.14 | 75.08 ± 0.46 | 88.73 ± 0.13 | 80.66 ± 0.59 | 60.95 ± 1.04 | 56.83 ± 0.61 | 82.93 ± 0.16 | 74.64 ± 0.23 | 73.27 ± 0.15 | 71.25 ± 0.00 | 2.80 |
| RUNG | 83.64 ± 1.65 | 73.46 ± 0.87 | 89.78 ± 0.14 | 79.95 ± 0.23 | 63.93 ± 0.74 | 55.34 ± 0.80 | – | – | – | – | 3.50 |

Table 30: Accuracy under the inductive/evasion setting against structural attack. (ptb_rate=0.2, atk_emb=BoW, def_emb=RoBERTa)

| Method | Cora | CiteSeer | PubMed | WikiCS | Instagram | Reddit | History | Photo | Computer | ArXiv | Avg Rank |
|---|---|---|---|---|---|---|---|---|---|---|---|
| GCN | 73.31 ± 2.11 | 62.70 ± 0.59 | 82.92 ± 0.52 | 31.54 ± 0.71 | 60.60 ± 0.89 | 65.16 ± 2.29 | 65.81 ± 0.48 | 57.37 ± 0.97 | 55.09 ± 0.23 | 45.28 ± 0.00 | 10.50 |
| GAT | 73.43 ± 1.59 | 63.64 ± 0.90 | 83.60 ± 0.60 | 49.65 ± 1.11 | 62.86 ± 1.37 | 61.50 ± 1.18 | 69.08 ± 1.89 | 58.25 ± 0.89 | 54.77 ± 1.09 | 54.77 ± 0.00 | 9.10 |
| APPNP | 81.06 ± 2.31 | 70.43 ± 0.70 | 89.30 ± 0.18 | 76.53 ± 0.48 | 63.11 ± 1.23 | 55.37 ± 0.72 | 79.28 ± 0.40 | 65.14 ± 0.64 | 58.17 ± 0.46 | 58.06 ± 0.00 | 6.10 |
| GPRGNN | 79.27 ± 2.79 | 69.96 ± 1.82 | 89.47 ± 0.45 | 74.38 ± 0.63 | 60.85 ± 0.58 | 56.34 ± 0.35 | 81.30 ± 0.50 | 66.96 ± 0.62 | 62.63 ± 0.94 | 68.73 ± 0.00 | 6.20 |
| RobustGCN | 73.49 ± 1.80 | 60.97 ± 1.02 | 82.15 ± 0.24 | 36.85 ± 1.66 | 64.68 ± 0.32 | 57.59 ± 1.97 | 64.25 ± 0.40 | 56.74 ± 0.78 | 51.85 ± 2.42 | 51.90 ± 0.00 | 10.80 |
| GCORN | 73.49 ± 2.22 | 63.48 ± 0.66 | 83.98 ± 0.55 | 34.97 ± 0.79 | 58.11 ± 0.58 | 68.09 ± 1.86 | 59.00 ± 1.36 | 53.80 ± 0.53 | 46.08 ± 0.17 | 41.82 ± 0.00 | 10.60 |
| NoisyGCN | 73.55 ± 1.35 | 62.59 ± 1.28 | 83.02 ± 0.48 | 32.14 ± 0.77 | 60.85 ± 0.70 | 65.77 ± 2.11 | 65.44 ± 0.58 | 57.76 ± 1.11 | 55.09 ± 0.55 | 45.43 ± 0.00 | 9.60 |
| GRAND | 79.89 ± 0.80 | 70.74 ± 1.58 | 86.41 ± 0.52 | 71.86 ± 0.61 | 64.68 ± 0.32 | 63.44 ± 0.81 | 76.87 ± 0.18 | 69.54 ± 0.14 | 64.94 ± 0.23 | 69.80 ± 0.00 | 4.70 |
| SoftmedianGDC | 80.01 ± 2.53 | 70.53 ± 1.02 | 87.36 ± 0.26 | 81.43 ± 0.66 | 62.85 ± 0.61 | 63.10 ± 0.18 | 79.89 ± 0.51 | – | – | – | 5.71 |
| EvenNet | 79.83 ± 1.30 | 71.47 ± 0.64 | 87.53 ± 0.45 | 63.02 ± 0.33 | 66.64 ± 1.09 | 61.98 ± 2.69 | 72.86 ± 0.14 | 61.93 ± 0.38 | 59.79 ± 0.47 | 53.73 ± 0.00 | 6.00 |
| ElasticGNN | 74.91 ± 1.09 | 61.96 ± 1.03 | 84.22 ± 0.74 | 57.77 ± 1.09 | 57.10 ± 4.69 | 54.11 ± 1.06 | 70.63 ± 0.73 | 62.71 ± 0.60 | 54.45 ± 0.70 | 54.50 ± 0.00 | 9.90 |
| GNNGuard | 83.64 ± 1.94 | 74.03 ± 0.53 | 90.14 ± 0.44 | 83.03 ± 0.53 | 64.07 ± 1.19 | 58.58 ± 0.33 | 84.64 ± 0.49 | 63.04 ± 0.71 | 63.07 ± 0.60 | 71.16 ± 0.00 | 3.60 |
| RUNG | 84.56 ± 1.52 | 73.30 ± 1.19 | 90.72 ± 0.25 | 82.73 ± 0.78 | 67.12 ± 0.56 | 58.11 ± 0.75 | – | – | – | – | 3.17 |
| Guardual | 83.95 ± 1.63 | 74.71 ± 1.22 | 89.53 ± 0.39 | 83.07 ± 0.55 | 65.40 ± 0.05 | 61.64 ± 0.46 | 84.85 ± 0.29 | 73.78 ± 0.44 | 71.30 ± 0.39 | 72.15 ± 0.00 | 2.10 |

Table 31: Accuracy under the transductive/poisoning setting against structural attack. (ptb_rate=0.2, atk_emb=BoW, def_emb=BoW)

| Method | Cora | CiteSeer | PubMed | WikiCS | Instagram | Reddit | History | Avg Rank |
|---|---|---|---|---|---|---|---|---|
| GCN | 64.56 ± 1.14 | 71.31 ± 0.44 | 77.59 ± 0.23 | 79.96 ± 0.35 | 63.87 ± 0.17 | 52.28 ± 1.42 | 64.39 ± 0.33 | 8.71 |
| GAT | 60.78 ± 2.21 | 70.55 ± 1.26 | 75.51 ± 0.28 | 78.55 ± 0.83 | 70.54 ± 1.75 | 53.49 ± 2.11 | 71.18 ± 1.21 | 8.57 |
| APPNP | 70.47 ± 1.21 | 72.32 ± 0.77 | 83.61 ± 0.19 | 79.83 ± 0.17 | 64.33 ± 0.25 | 54.73 ± 0.79 | 74.60 ± 0.07 | 3.71 |
| GPRGNN | 71.45 ± 2.32 | 71.61 ± 0.99 | 81.74 ± 0.78 | 79.02 ± 1.30 | 69.35 ± 0.42 | 69.21 ± 1.36 | 73.49 ± 0.69 | 4.14 |
| RobustGCN | 59.94 ± 0.31 | 70.16 ± 1.81 | 69.61 ± 0.35 | 80.84 ± 0.20 | 63.56 ± 0.60 | 52.19 ± 0.21 | 63.72 ± 2.69 | 12.00 |
| GCORN | 47.92 ± 3.14 | 69.75 ± 0.80 | 55.58 ± 0.59 | 77.62 ± 0.41 | 63.77 ± 0.14 | 52.92 ± 0.27 | 58.59 ± 0.09 | 13.71 |
| NoisyGCN | 63.74 ± 0.44 | 71.87 ± 0.15 | 76.88 ± 0.76 | 79.79 ± 0.34 | 63.87 ± 0.17 | 52.21 ± 1.42 | 63.14 ± 1.96 | 9.86 |
| GRAND | 64.05 ± 3.51 | 72.22 ± 0.98 | 77.90 ± 1.65 | 79.98 ± 0.35 | 63.65 ± 0.13 | 55.05 ± 0.92 | 63.92 ± 0.06 | 7.57 |
| SoftmedianGDC | 69.65 ± 0.71 | 71.85 ± 0.73 | 80.65 ± 1.03 | 73.97 ± 0.50 | 63.50 ± 0.14 | 53.19 ± 1.01 | 72.79 ± 0.14 | 8.71 |
| EvenNet | 72.28 ± 1.98 | 69.78 ± 0.66 | 85.63 ± 0.23 | 80.79 ± 0.61 | 79.57 ± 0.84 | 66.64 ± 1.25 | 77.04 ± 0.20 | 3.14 |
| ElasticGNN | 64.37 ± 1.89 | 71.64 ± 0.57 | 78.75 ± 0.68 | 79.90 ± 0.50 | 63.81 ± 0.26 | 51.76 ± 1.63 | 66.72 ± 0.80 | 9.14 |
| GNNGuard | 65.68 ± 0.36 | 70.81 ± 0.51 | 81.63 ± 0.39 | 77.45 ± 0.28 | 64.52 ± 0.62 | 54.60 ± 0.33 | 69.35 ± 0.79 | 7.57 |
| RUNG | 68.60 ± 2.18 | 71.14 ± 0.38 | 82.46 ± 0.27 | 78.20 ± 0.38 | 63.89 ± 0.14 | 52.37 ± 1.16 | | 7.83 |
| Cosine-GCN | 65.13 ± 0.98 | 69.01 ± 1.45 | 82.03 ± 0.45 | 77.07 ± 0.36 | 64.36 ± 0.59 | 54.32 ± 0.99 | 70.11 ± 1.38 | 8.71 |
| Jaccard-GCN | 61.21 ± 0.80 | 69.49 ± 0.61 | 74.76 ± 0.52 | 78.45 ± 0.51 | 64.23 ± 0.64 | 51.89 ± 0.65 | 63.40 ± 2.38 | 12.14 |
| Stable | 60.18 ± 4.17 | 67.02 ± 1.18 | 72.35 ± 2.72 | 77.58 ± 0.58 | 63.89 ± 0.23 | 55.77 ± 0.73 | – | 11.67 |
| ProGNN | 76.57 ± 1.96 | 70.22 ± 0.54 | – | – | – | – | – | 6.00 |

Table 32: Accuracy under the transductive/poisoning setting against structural attack. (ptb_rate=0.2, atk_emb=BoW, def_emb=MiniLM)

| Method | Cora | CiteSeer | PubMed | WikiCS | Instagram | Reddit | History | Avg Rank |
|---|---|---|---|---|---|---|---|---|
| GCN | 71.84 ± 2.32 | 73.66 ± 0.20 | 79.44 ± 0.27 | 81.19 ± 0.28 | 63.81 ± 0.18 | 49.89 ± 0.01 | 66.90 ± 1.80 | 11.00 |
| GAT | 70.64 ± 0.61 | 73.51 ± 0.31 | 79.59 ± 0.65 | 80.98 ± 0.28 | 73.43 ± 0.80 | 58.88 ± 0.84 | 78.92 ± 0.42 | 8.86 |
| APPNP | 78.39 ± 0.48 | 74.79 ± 0.58 | 86.55 ± 0.11 | 81.04 ± 0.15 | 64.02 ± 0.62 | 58.86 ± 0.23 | 82.15 ± 0.11 | 4.43 |
| GPRGNN | 78.10 ± 0.74 | 74.70 ± 0.75 | 85.80 ± 0.27 | 81.32 ± 0.35 | 72.95 ± 2.44 | 67.84 ± 0.36 | 80.39 ± 0.10 | 4.57 |
| RobustGCN | 71.33 ± 1.69 | 74.00 ± 0.32 | 78.17 ± 0.48 | 81.65 ± 0.20 | 63.80 ± 0.19 | 49.99 ± 0.13 | 65.64 ± 1.31 | 10.29 |
| GCORN | 58.96 ± 2.70 | 74.71 ± 0.74 | 57.61 ± 0.29 | 79.47 ± 0.44 | 63.87 ± 0.17 | 49.87 ± 0.02 | 56.22 ± 0.07 | 12.86 |
| NoisyGCN | 72.79 ± 3.09 | 73.99 ± 0.31 | 79.23 ± 0.88 | 79.23 ± 0.38 | 63.81 ± 0.27 | 49.96 ± 0.09 | 68.48 ± 2.00 | 10.14 |
| GRAND | 74.08 ± 1.81 | 75.18 ± 0.96 | 80.62 ± 0.66 | 81.55 ± 0.26 | 63.65 ± 0.04 | 50.79 ± 0.73 | 68.14 ± 0.50 | 8.00 |
| SoftmedianGDC | 78.06 ± 0.49 | 75.19 ± 1.08 | 85.76 ± 0.39 | 80.66 ± 0.60 | 62.83 ± 1.24 | 51.03 ± 0.55 | 81.18 ± 0.15 | 8.29 |
| EvenNet | 78.93 ± 1.18 | 74.74 ± 0.19 | 87.37 ± 0.12 | 82.18 ± 0.26 | 78.67 ± 0.82 | 67.23 ± 1.28 | 82.81 ± 0.08 | 2.00 |
| ElasticGNN | 74.70 ± 1.77 | 74.16 ± 0.68 | 79.21 ± 0.63 | 81.25 ± 0.38 | 63.83 ± 0.15 | 50.02 ± 0.24 | 64.78 ± 0.24 | 9.29 |
| GNNGuard | 80.08 ± 0.54 | 73.69 ± 1.47 | 86.08 ± 0.25 | 81.00 ± 0.13 | 64.34 ± 0.39 | 58.70 ± 0.94 | 82.14 ± 0.06 | 5.86 |
| RUNG | 78.20 ± 0.94 | 73.78 ± 0.16 | 86.13 ± 0.42 | 80.76 ± 0.10 | 64.15 ± 0.36 | 51.70 ± 0.90 | – | 7.50 |
| Cosine-GCN | 80.08 ± 0.83 | 73.90 ± 0.45 | 85.50 ± 0.42 | 80.68 ± 0.26 | 64.53 ± 0.73 | 58.59 ± 0.95 | 81.98 ± 0.09 | 6.57 |
| Jaccard-GCN | 72.07 ± 1.57 | 73.47 ± 0.22 | 77.08 ± 0.35 | 80.45 ± 0.51 | 63.74 ± 0.14 | 40.96 ± 5.10 | 63.14 ± 0.45 | 14.43 |
| Stable | 75.20 ± 0.81 | 72.80 ± 0.69 | 78.89 ± 0.91 | 80.88 ± 0.17 | 63.63 ± 0.38 | 49.02 ± 0.64 | – | 13.33 |
| ProGNN | 83.20 ± 0.51 | 73.78 ± 1.51 | – | – | – | – | – | 6.00 |

Table 33: Accuracy under the transductive/poisoning setting against structural attack. (ptb_rate=0.2, atk_emb=BoW, def_emb=RoBERTa)

| Method | Cora | CiteSeer | PubMed | WikiCS | Instagram | Reddit | History | Avg Rank |
|---|---|---|---|---|---|---|---|---|
| GCN | 69.57 ± 3.44 | 74.68 ± 0.41 | 78.44 ± 1.30 | 81.69 ± 0.37 | 63.86 ± 0.18 | 52.77 ± 0.72 | 68.51 ± 1.53 | 11.00 |
| GAT | 67.64 ± 2.71 | 73.86 ± 0.43 | 78.80 ± 1.09 | 81.44 ± 0.13 | 72.22 ± 1.78 | 56.14 ± 3.40 | 78.09 ± 0.41 | 9.86 |
| APPNP | 80.62 ± 0.26 | 75.01 ± 0.50 | 88.23 ± 0.27 | 82.87 ± 0.50 | 66.24 ± 0.09 | 59.59 ± 0.21 | 83.73 ± 0.12 | 3.57 |
| GPRGNN | 79.93 ± 1.21 | 75.11 ± 0.59 | 87.36 ± 0.55 | 82.32 ± 0.38 | 73.19 ± 6.80 | 68.58 ± 0.59 | 82.70 ± 0.13 | 3.71 |
| RobustGCN | 66.44 ± 3.26 | 74.19 ± 0.28 | 64.78 ± 1.59 | 81.96 ± 0.26 | 63.64 ± 0.20 | 51.28 ± 0.46 | 64.55 ± 2.99 | 12.86 |
| GCORN | 67.65 ± 0.86 | 74.68 ± 0.96 | 61.51 ± 0.48 | 80.77 ± 0.40 | 63.88 ± 0.17 | 49.40 ± 0.33 | 62.32 ± 0.67 | 13.29 |
| NoisyGCN | 71.57 ± 1.42 | 74.57 ± 0.94 | 77.73 ± 0.92 | 81.61 ± 0.47 | 63.75 ± 0.21 | 53.08 ± 3.68 | 70.33 ± 1.22 | 11.29 |
| GRAND | 80.05 ± 0.76 | 75.81 ± 0.74 | 83.73 ± 0.50 | 83.23 ± 0.45 | 63.93 ± 0.20 | 57.16 ± 0.40 | 77.99 ± 0.55 | 5.43 |
| SoftmedianGDC | 78.00 ± 0.44 | 73.14 ± 0.42 | 85.85 ± 0.43 | 79.78 ± 0.56 | 64.38 ± 0.35 | 57.77 ± 0.51 | 82.63 ± 0.26 | 9.43 |
| EvenNet | 80.82 ± 1.22 | 75.02 ± 0.16 | 88.86 ± 0.10 | 83.34 ± 0.81 | 81.09 ± 0.20 | 68.71 ± 0.46 | 84.10 ± 0.05 | 1.57 |
| ElasticGNN | 75.02 ± 0.48 | 73.87 ± 0.19 | 82.05 ± 0.74 | 82.28 ± 0.17 | 63.84 ± 0.14 | 52.56 ± 1.83 | 70.57 ± 1.78 | 10.43 |
| GNNGuard | 79.93 ± 1.52 | 73.99 ± 0.62 | 86.95 ± 0.06 | 82.45 ± 0.75 | 66.80 ± 0.29 | 60.06 ± 0.46 | 83.48 ± 0.13 | 5.14 |
| RUNG | 79.23 ± 1.25 | 75.45 ± 0.88 | 87.55 ± 0.45 | 81.80 ± 0.23 | 63.90 ± 0.23 | 54.61 ± 1.40 | – | 6.83 |
| Cosine-GCN | 79.88 ± 0.66 | 73.10 ± 1.19 | 86.75 ± 0.27 | 81.91 ± 0.71 | 66.65 ± 0.35 | 58.18 ± 0.38 | 83.54 ± 0.23 | 7.00 |
| Jaccard-GCN | 69.82 ± 1.49 | 72.47 ± 0.45 | 75.94 ± 1.43 | 81.05 ± 0.47 | 63.50 ± 0.49 | 49.56 ± 2.21 | 63.00 ± 4.95 | 14.29 |
| Stable | 78.28 ± 1.45 | 72.46 ± 0.66 | 77.66 ± 1.31 | 80.25 ± 0.96 | 63.93 ± 0.20 | 53.32 ± 2.22 | – | 12.00 |
| ProGNN | 83.57 ± 0.79 | 74.98 ± 0.38 | – | – | – | – | – | 3.50 |

Table 34: Accuracy under the inductive/evasion setting against textual attack. (ptb_rate=0.4, def_emb=BoW)

| Method | Cora | CiteSeer | PubMed | WikiCS | Instagram | Reddit | History | Photo | Avg Rank |
|---|---|---|---|---|---|---|---|---|---|
| GCN | 82.72 ± 1.65 | 60.66 ± 0.77 | 82.00 ± 1.30 | 78.50 ± 0.21 | 61.16 ± 0.92 | 59.70 ± 1.08 | 78.01 ± 0.45 | 78.65 ± 0.25 | 4.38 |
| GAT | 82.90 ± 2.20 | 61.76 ± 2.50 | 83.05 ± 0.69 | 71.54 ± 1.13 | 63.21 ± 1.40 | 60.60 ± 1.30 | 79.55 ± 1.09 | 82.05 ± 0.30 | 3.25 |
| APPNP | 80.07 ± 0.94 | 38.35 ± 1.03 | 67.82 ± 0.80 | 50.23 ± 0.54 | 60.42 ± 0.46 | 55.08 ± 0.21 | 74.79 ± 0.41 | 73.52 ± 0.54 | 9.88 |
| GPRGNN | 74.11 ± 2.38 | 40.39 ± 1.58 | 73.55 ± 2.15 | 54.82 ± 0.44 | 59.80 ± 1.14 | 58.65 ± 0.89 | 67.48 ± 0.93 | 71.34 ± 1.77 | 10.12 |
| RobustGCN | 84.01 ± 2.26 | 63.79 ± 2.22 | 82.16 ± 0.80 | 74.60 ± 0.52 | 65.70 ± 0.16 | 56.33 ± 0.87 | 77.49 ± 0.31 | 78.39 ± 0.17 | 3.88 |
| GCORN | 77.31 ± 1.63 | 54.49 ± 0.64 | 77.09 ± 0.83 | 76.35 ± 0.19 | 62.02 ± 0.47 | 65.51 ± 0.65 | 76.46 ± 0.35 | 74.52 ± 0.75 | 5.75 |
| NoisyGCN | 83.27 ± 1.35 | 59.93 ± 0.45 | 81.68 ± 1.24 | 78.39 ± 0.09 | 60.52 ± 0.61 | 61.32 ± 0.64 | 78.13 ± 0.31 | 79.03 ± 0.16 | 3.75 |
| GRAND | 75.77 ± 2.22 | 47.96 ± 1.48 | 78.54 ± 1.72 | 73.77 ± 0.24 | 64.64 ± 0.38 | 62.69 ± 2.23 | 75.60 ± 0.91 | 74.99 ± 0.10 | 6.25 |
| SoftmedianGDC | 70.54 ± 2.01 | 47.23 ± 0.27 | 70.63 ± 1.04 | 47.05 ± 0.39 | 59.16 ± 0.56 | 59.10 ± 0.36 | 61.46 ± 0.46 | – | 10.71 |
| EvenNet | 79.95 ± 0.74 | 52.72 ± 0.90 | 81.17 ± 1.09 | 74.94 ± 0.61 | 62.45 ± 1.25 | 59.61 ± 0.40 | 75.90 ± 0.49 | 82.24 ± 0.26 | 5.50 |
| ElasticGNN | 82.96 ± 2.05 | 58.73 ± 1.53 | 83.07 ± 0.31 | 69.59 ± 0.87 | 64.34 ± 1.24 | 59.96 ± 0.59 | 77.40 ± 0.29 | 78.73 ± 0.37 | 4.25 |
| GNNGuard | 76.81 ± 1.51 | 37.62 ± 0.34 | 66.37 ± 0.47 | 43.70 ± 0.32 | 58.91 ± 0.97 | 51.93 ± 0.65 | 68.78 ± 1.17 | 66.70 ± 0.63 | 11.75 |
| RUNG | 65.50 ± 4.14 | 43.26 ± 0.90 | 64.24 ± 1.08 | 48.90 ± 0.38 | 61.20 ± 0.67 | 55.69 ± 0.63 | – | – | 10.83 |

Table 35: Accuracy under the inductive/evasion setting against textual attack. (ptb_rate=0.4, def_emb=MiniLM)

| Method | Cora | CiteSeer | PubMed | WikiCS | Instagram | Reddit | History | Photo | Avg Rank |
|---|---|---|---|---|---|---|---|---|---|
| GCN | 80.01 ± 2.22 | 58.52 ± 0.91 | 78.25 ± 0.82 | 67.45 ± 0.54 | 59.01 ± 0.93 | 64.33 ± 0.20 | 76.48 ± 0.33 | 73.66 ± 1.11 | 3.38 |
| GAT | 81.98 ± 1.09 | 58.62 ± 1.81 | 78.66 ± 3.94 | 61.65 ± 1.15 | 63.18 ± 1.19 | 62.65 ± 1.46 | 77.01 ± 0.75 | 72.78 ± 3.41 | 3.00 |
| APPNP | 58.73 ± 0.23 | 40.23 ± 1.84 | 66.51 ± 0.64 | 46.68 ± 0.30 | 58.83 ± 1.19 | 53.07 ± 0.32 | 66.07 ± 1.04 | 54.41 ± 0.26 | 10.25 |
| GPRGNN | 71.77 ± 2.13 | 43.16 ± 1.53 | 69.84 ± 0.66 | 46.53 ± 0.28 | 57.88 ± 1.54 | 53.98 ± 0.31 | 65.12 ± 0.85 | 54.71 ± 0.33 | 9.25 |
| RobustGCN | 82.29 ± 1.29 | 60.61 ± 1.97 | 78.15 ± 0.26 | 69.83 ± 1.37 | 63.67 ± 0.51 | 55.63 ± 0.22 | 77.73 ± 0.41 | 73.08 ± 0.67 | 2.88 |
| GCORN | 79.21 ± 0.53 | 54.70 ± 0.84 | 72.06 ± 0.70 | 62.45 ± 1.18 | 57.51 ± 0.46 | 64.15 ± 0.47 | 74.47 ± 0.50 | 63.62 ± 0.48 | 6.00 |
| NoisyGCN | 79.46 ± 1.43 | 58.41 ± 1.03 | 78.50 ± 0.83 | 67.81 ± 1.22 | 58.85 ± 1.30 | 64.67 ± 0.76 | 76.24 ± 0.48 | 74.12 ± 0.75 | 3.38 |
| GRAND | 82.08 ± 2.39 | 46.92 ± 1.41 | 69.79 ± 0.40 | 47.50 ± 0.37 | 64.58 ± 0.98 | 60.92 ± 0.21 | 69.08 ± 0.62 | 55.84 ± 0.50 | 7.38 |
| SoftmedianGDC | 68.57 ± 0.63 | 51.15 ± 0.39 | 69.32 ± 0.74 | 54.46 ± 1.76 | 51.94 ± 0.42 | – | 63.31 ± 3.21 | – | 9.17 |
| EvenNet | 75.46 ± 1.34 | 48.33 ± 1.85 | 74.37 ± 0.68 | 51.86 ± 1.37 | 59.58 ± 0.73 | 56.55 ± 0.36 | 72.73 ± 0.44 | 75.15 ± 1.57 | 6.00 |
| ElasticGNN | 76.75 ± 1.96 | 56.11 ± 1.89 | 78.57 ± 0.64 | 51.80 ± 0.97 | 61.45 ± 1.02 | 60.19 ± 0.60 | 75.86 ± 0.73 | 60.90 ± 1.07 | 5.38 |
| GNNGuard | 52.34 ± 0.57 | 39.66 ± 0.56 | 65.82 ± 0.39 | 45.25 ± 0.34 | 51.53 ± 2.12 | 51.53 ± 0.77 | 56.90 ± 0.51 | 43.72 ± 0.22 | 12.50 |
| RUNG | 55.78 ± 0.63 | 39.76 ± 1.03 | 66.60 ± 0.34 | 46.45 ± 0.39 | 57.72 ± 1.00 | 53.22 ± 1.01 | – | – | 11.17 |

Table 36: Accuracy under the inductive/evasion setting against textual attack. (ptb_rate=0.4, def_emb=RoBERTa)

| Method | Cora | CiteSeer | PubMed | WikiCS | Instagram | Reddit | History | Photo | Avg Rank |
|---|---|---|---|---|---|---|---|---|---|
| GCN | 81.12 ± 1.61 | 59.14 ± 0.96 | 80.13 ± 0.39 | 73.27 ± 0.93 | 63.57 ± 1.47 | 65.42 ± 0.34 | 78.68 ± 0.22 | 74.01 ± 0.33 | 2.50 |
| GAT | 81.49 ± 1.58 | 57.84 ± 3.32 | 81.94 ± 0.97 | 67.54 ± 7.56 | 64.29 ± 1.30 | 63.29 ± 0.12 | 79.10 ± 1.36 | 79.17 ± 0.80 | 2.62 |
| APPNP | 59.29 ± 1.84 | 42.58 ± 0.96 | 67.57 ± 0.55 | 48.21 ± 0.37 | 58.35 ± 1.28 | 55.12 ± 0.72 | 68.43 ± 0.51 | 54.96 ± 0.91 | 10.25 |
| GPRGNN | 66.30 ± 1.54 | 41.80 ± 0.27 | 67.43 ± 0.71 | 48.40 ± 0.79 | 57.39 ± 1.73 | 54.69 ± 0.37 | 62.84 ± 1.09 | 55.86 ± 2.28 | 10.25 |
| RobustGCN | 83.03 ± 2.11 | 60.24 ± 1.07 | 79.07 ± 1.13 | 69.81 ± 2.56 | 64.51 ± 0.57 | 60.51 ± 0.60 | 78.51 ± 0.10 | 68.58 ± 1.45 | 3.25 |
| GCORN | 79.21 ± 1.28 | 56.06 ± 0.74 | 74.51 ± 0.58 | 66.75 ± 1.32 | 58.80 ± 1.18 | 64.97 ± 0.42 | 76.91 ± 0.67 | 64.81 ± 0.94 | 5.50 |
| NoisyGCN | 80.75 ± 1.43 | 59.72 ± 1.91 | 79.97 ± 0.64 | 73.25 ± 0.97 | 62.85 ± 1.84 | 65.60 ± 0.63 | 78.74 ± 0.30 | 73.90 ± 0.53 | 2.88 |
| GRAND | 69.43 ± 2.72 | 46.08 ± 1.56 | 72.62 ± 0.75 | 52.44 ± 0.50 | 66.05 ± 1.24 | 63.42 ± 1.38 | 74.01 ± 0.47 | 57.19 ± 1.09 | 6.38 |
| SoftmedianGDC | 65.99 ± 0.31 | 49.69 ± 0.22 | 69.09 ± 0.37 | 51.45 ± 0.88 | 55.00 ± 1.00 | – | 63.03 ± 2.97 | – | 9.50 |
| EvenNet | 68.82 ± 0.75 | 44.15 ± 1.07 | 73.50 ± 0.48 | 54.32 ± 2.56 | 58.86 ± 0.72 | 56.70 ± 0.48 | 72.50 ± 0.66 | 72.42 ± 1.18 | 7.25 |
| ElasticGNN | 78.91 ± 3.48 | 52.19 ± 0.56 | 78.19 ± 0.28 | 56.07 ± 1.37 | 63.32 ± 2.08 | 60.48 ± 0.30 | 76.95 ± 0.37 | 59.24 ± 2.94 | 5.88 |
| GNNGuard | 51.91 ± 1.14 | 38.82 ± 0.63 | 65.73 ± 0.52 | 47.12 ± 0.13 | 54.32 ± 0.65 | 53.94 ± 1.22 | 61.69 ± 1.77 | 54.98 ± 0.08 | 12.25 |
| RUNG | 62.12 ± 3.13 | 38.56 ± 2.11 | 66.49 ± 0.66 | 48.07 ± 0.42 | 58.80 ± 1.09 | 54.07 ± 0.80 | – | – | 11.17 |

Table 37: Accuracy under the transductive/poisoning setting against textual attack. (ptb_rate=0.8, def_emb=BoW)

| Method | Cora | CiteSeer | PubMed | WikiCS | Instagram | Reddit | History | Avg Rank |
|---|---|---|---|---|---|---|---|---|
| GCN | 83.63 ± 0.61 | 65.20 ± 1.19 | 85.16 ± 0.09 | 79.85 ± 0.25 | 65.81 ± 0.29 | 61.18 ± 0.61 | 79.69 ± 0.33 | 6.14 |
| GAT | 83.74 ± 0.12 | 65.54 ± 0.90 | 84.98 ± 0.05 | 79.05 ± 0.97 | 65.55 ± 0.34 | 60.92 ± 0.30 | 80.52 ± 0.16 | 6.86 |
| APPNP | **84.59 ± 0.69** | 66.58 ± 1.11 | 86.48 ± 0.28 | 79.88 ± 0.49 | 65.65 ± 0.38 | 60.60 ± 0.17 | **80.72 ± 0.13** | 3.57 |
| GPRGNN | 84.43 ± 0.45 | 66.93 ± 1.32 | 85.75 ± 0.07 | 78.83 ± 0.14 | 65.56 ± 0.44 | 59.83 ± 0.40 | 80.12 ± 0.22 | 6.43 |
| RobustGCN | 83.65 ± 0.48 | 64.60 ± 1.05 | 85.53 ± 0.16 | **80.39 ± 0.37** | 65.59 ± 0.12 | 59.26 ± 0.31 | 80.62 ± 0.10 | 6.00 |
| GCORN | 74.71 ± 1.80 | 64.35 ± 1.03 | 83.32 ± 0.19 | 77.63 ± 0.41 | 64.79 ± 0.51 | **65.30 ± 0.16** | 77.94 ± 0.17 | 10.86 |
| NoisyGCN | 83.13 ± 0.70 | 65.36 ± 0.83 | 85.26 ± 0.12 | 79.77 ± 0.26 | 65.81 ± 0.35 | 63.27 ± 0.65 | 80.09 ± 0.09 | 6.00 |
| GRAND | 83.93 ± 0.30 | 67.70 ± 0.68 | **86.58 ± 0.25** | 79.46 ± 0.46 | 65.53 ± 0.25 | 62.91 ± 0.08 | 80.28 ± 0.15 | 4.43 |
| SoftmedianGDC | 82.23 ± 0.88 | 60.27 ± 1.34 | 83.70 ± 0.04 | 70.02 ± 0.11 | **66.09 ± 0.31** | 61.00 ± 0.58 | 77.21 ± 0.35 | 10.71 |
| EvenNet | 83.39 ± 1.02 | 59.98 ± 0.16 | 85.98 ± 0.17 | 80.07 ± 0.59 | 65.30 ± 0.46 | 64.56 ± 0.53 | 80.19 ± 0.13 | 6.71 |
| ElasticGNN | 83.62 ± 0.53 | **68.37 ± 0.93** | 86.39 ± 0.21 | 79.78 ± 0.14 | 65.53 ± 0.27 | 60.59 ± 0.38 | 80.52 ± 0.05 | 5.43 |
| GNNGuard | 79.23 ± 0.36 | 17.98 ± 0.47 | 85.14 ± 0.19 | 72.80 ± 0.55 | 64.37 ± 0.14 | 58.73 ± 0.30 | 78.93 ± 0.09 | 13.29 |
| RUNG | 82.88 ± 0.59 | 61.48 ± 1.49 | 85.01 ± 0.24 | 77.69 ± 0.51 | 65.28 ± 0.25 | 60.49 ± 0.11 | – | 10.67 |
| Cosine-GCN | 78.59 ± 0.51 | 59.74 ± 0.21 | 82.06 ± 0.30 | 72.68 ± 0.18 | 64.72 ± 0.11 | 58.66 ± 0.34 | 77.44 ± 0.22 | 14.86 |
| Jaccard-GCN | 79.17 ± 1.96 | 60.25 ± 0.58 | 82.36 ± 0.22 | 77.38 ± 0.36 | 64.94 ± 0.59 | 61.16 ± 0.27 | 77.72 ± 0.03 | 12.14 |
| Stable | 77.86 ± 0.62 | 61.11 ± 0.40 | 82.22 ± 0.24 | 77.51 ± 0.16 | 64.59 ± 0.22 | 62.95 ± 0.34 | 74.80 ± 0.35 | 12.57 |
| ProGNN | 76.08 ± 0.34 | 66.07 ± 0.84 | – | – | – | – | – | 10.50 |

Table 38: Accuracy under the transductive/poisoning setting against textual attack. (ptb_rate=0.8, def_emb=MiniLM)

| Method | Cora | CiteSeer | PubMed | WikiCS | Instagram | Reddit | History | Avg Rank |
|---|---|---|---|---|---|---|---|---|
| GCN | 83.65 ± 1.28 | 69.52 ± 0.53 | 86.01 ± 0.31 | 80.65 ± 0.28 | 65.16 ± 0.58 | 65.36 ± 0.38 | 82.46 ± 0.07 | 8.14 |
| GAT | 84.33 ± 0.67 | 70.46 ± 0.75 | 85.38 ± 0.28 | 81.47 ± 0.53 | 65.29 ± 0.07 | 63.69 ± 0.43 | 82.45 ± 0.18 | 6.71 |
| APPNP | 84.17 ± 0.90 | 70.94 ± 0.62 | 87.35 ± 0.05 | 81.13 ± 0.16 | 65.70 ± 0.69 | 61.86 ± 0.27 | **83.13 ± 0.06** | 4.57 |
| GPRGNN | 84.23 ± 0.56 | 70.35 ± 0.40 | **87.47 ± 0.40** | 80.76 ± 0.25 | 64.76 ± 0.47 | 61.40 ± 0.48 | 82.31 ± 0.08 | 7.86 |
| RobustGCN | 84.60 ± 0.61 | 69.77 ± 0.28 | 86.04 ± 0.21 | 81.41 ± 0.13 | **65.98 ± 0.23** | 64.06 ± 0.39 | 82.81 ± 0.08 | 4.57 |
| GCORN | 82.66 ± 1.20 | 71.26 ± 0.77 | 85.30 ± 0.22 | 78.91 ± 0.10 | 65.72 ± 0.25 | 67.37 ± 0.13 | 82.41 ± 0.10 | 7.14 |
| NoisyGCN | 83.53 ± 0.75 | 68.72 ± 0.22 | 86.04 ± 0.38 | 80.35 ± 0.40 | 65.87 ± 0.63 | **67.86 ± 0.28** | 82.69 ± 0.10 | 6.57 |
| GRAND | **84.85 ± 0.55** | **71.60 ± 0.90** | 85.17 ± 0.24 | 78.54 ± 0.65 | 65.48 ± 0.19 | 65.24 ± 0.34 | 82.07 ± 0.07 | 7.14 |
| SoftmedianGDC | 84.22 ± 0.37 | 65.87 ± 1.06 | 85.78 ± 0.60 | 77.88 ± 0.43 | 65.45 ± 0.41 | – | 82.99 ± 0.10 | 9.33 |
| EvenNet | 84.09 ± 0.40 | 68.51 ± 0.60 | 86.90 ± 0.05 | **82.00 ± 0.34** | 65.59 ± 0.18 | 61.70 ± 0.09 | 82.94 ± 0.18 | 6.43 |
| ElasticGNN | 84.37 ± 0.70 | 69.94 ± 0.80 | 87.28 ± 0.16 | 81.36 ± 0.11 | 65.17 ± 0.75 | 62.62 ± 0.17 | 83.05 ± 0.18 | 5.71 |
| GNNGuard | 79.70 ± 0.25 | 17.98 ± 0.47 | 83.85 ± 0.35 | 76.18 ± 0.36 | 65.39 ± 0.92 | 58.95 ± 0.32 | 80.99 ± 0.15 | 14.29 |
| RUNG | 81.96 ± 0.79 | 67.09 ± 0.48 | 86.45 ± 0.13 | 80.41 ± 0.56 | 64.93 ± 0.04 | 61.32 ± 0.16 | – | 11.33 |
| Cosine-GCN | 79.45 ± 1.23 | 67.50 ± 1.22 | 82.62 ± 0.11 | 76.23 ± 0.31 | 65.10 ± 0.68 | 58.91 ± 0.29 | 80.64 ± 0.14 | 14.86 |
| Jaccard-GCN | 81.63 ± 1.34 | 67.09 ± 0.59 | 84.15 ± 0.25 | 80.11 ± 0.29 | 65.51 ± 0.16 | 66.19 ± 0.12 | 81.46 ± 0.17 | 10.43 |
| Stable | 81.62 ± 0.95 | 70.35 ± 0.66 | 84.06 ± 0.32 | 80.54 ± 0.58 | 65.35 ± 0.56 | 64.67 ± 0.22 | – | 9.67 |
| ProGNN | 81.88 ± 0.89 | 69.30 ± 0.58 | – | – | – | – | – | 11.50 |

Table 39: Accuracy under the transductive/poisoning setting against textual attack. (ptb_rate=0.8, def_emb=RoBERTa)

| Method | Cora | CiteSeer | PubMed | WikiCS | Instagram | Reddit | History | Avg Rank |
|---|---|---|---|---|---|---|---|---|
| GCN | 84.29 ± 0.68 | 68.67 ± 0.21 | 86.38 ± 0.15 | 81.70 ± 0.11 | 66.56 ± 0.22 | 63.80 ± 0.27 | 83.21 ± 0.13 | 8.14 |
| GAT | 84.97 ± 0.51 | 67.58 ± 1.60 | 85.54 ± 0.24 | 81.81 ± 0.30 | 66.37 ± 0.49 | 63.84 ± 0.48 | 83.31 ± 0.10 | 8.57 |
| APPNP | 85.03 ± 0.31 | 70.97 ± 0.35 | **87.83 ± 0.27** | 81.39 ± 0.06 | 66.86 ± 0.66 | 63.65 ± 0.28 | 83.57 ± 0.23 | 5.43 |
| GPRGNN | 85.00 ± 0.59 | 70.00 ± 1.38 | 87.82 ± 0.47 | 81.65 ± 0.07 | 66.76 ± 0.18 | 62.64 ± 0.46 | 83.53 ± 0.22 | 6.43 |
| RobustGCN | 84.28 ± 0.24 | 68.05 ± 0.23 | 86.28 ± 0.08 | 81.48 ± 0.16 | 65.89 ± 0.51 | 63.47 ± 0.12 | 83.00 ± 0.13 | 10.57 |
| GCORN | 83.40 ± 0.44 | 71.00 ± 1.12 | 85.69 ± 0.05 | 81.09 ± 0.37 | 66.58 ± 0.20 | **67.71 ± 0.11** | 83.67 ± 0.03 | 7.29 |
| NoisyGCN | 84.29 ± 0.23 | 68.35 ± 1.19 | 86.41 ± 0.08 | 81.47 ± 0.21 | 66.40 ± 0.33 | 63.65 ± 0.26 | 83.31 ± 0.03 | 8.71 |
| GRAND | **85.54 ± 0.35** | 71.40 ± 0.17 | 87.15 ± 0.21 | **82.89 ± 0.20** | **67.29 ± 0.20** | 67.02 ± 0.54 | **84.31 ± 0.09** | 2.14 |
| SoftmedianGDC | 83.33 ± 0.56 | 64.50 ± 0.98 | 86.25 ± 0.18 | 76.51 ± 0.37 | 66.10 ± 0.27 | – | 83.41 ± 0.08 | 12.67 |
| EvenNet | 84.96 ± 0.15 | 67.87 ± 0.59 | 87.64 ± 0.31 | 82.77 ± 0.27 | 66.78 ± 0.58 | 63.28 ± 1.04 | 83.63 ± 0.02 | 6.29 |
| ElasticGNN | 85.25 ± 0.81 | **72.02 ± 0.50** | 87.62 ± 0.09 | 82.33 ± 0.29 | 66.75 ± 0.34 | 63.88 ± 0.38 | 83.99 ± 0.05 | 3.43 |
| GNNGuard | 80.23 ± 0.78 | 17.98 ± 0.47 | 84.35 ± 0.30 | 77.26 ± 0.38 | 66.88 ± 0.36 | 58.93 ± 0.69 | 82.19 ± 0.17 | 13.00 |
| RUNG | 83.37 ± 0.56 | 68.71 ± 0.64 | 87.37 ± 0.28 | 81.56 ± 0.69 | 65.95 ± 1.54 | 63.01 ± 0.08 | – | 9.83 |
| Cosine-GCN | 77.33 ± 0.56 | 66.41 ± 0.66 | 83.29 ± 0.23 | 77.03 ± 0.54 | 66.58 ± 0.70 | 57.61 ± 1.08 | 81.77 ± 0.18 | 14.43 |
| Jaccard-GCN | 82.36 ± 0.85 | 66.92 ± 0.91 | 84.95 ± 0.22 | 81.04 ± 0.27 | 66.87 ± 0.31 | 63.82 ± 0.19 | 82.50 ± 0.18 | 10.86 |
| Stable | 83.83 ± 0.35 | 71.31 ± 0.67 | 84.87 ± 0.12 | 80.68 ± 0.53 | 66.88 ± 0.18 | 65.14 ± 0.38 | – | 7.67 |
| ProGNN | 82.48 ± 1.08 | 71.61 ± 2.06 | – | – | – | – | – | 8.00 |

## J.3 GRAPHLLM RESULTS

Results are in Tables 40, 41, 42, 43, 44, 45.

Table 40: GraphLLM's clean results in the inductive/evasion setting.

| Method | Cora | CiteSeer | PubMed | WikiCS | Instagram | Reddit | History | Photo | Computer | ArXiv |
|---|---|---|---|---|---|---|---|---|---|---|
| GraphGPT | 81.06 ± 2.33 | 74.35 ± 2.51 | 94.14 ± 0.23 | 82.31 ± 1.31 | 66.93 ± 1.44 | 61.87 ± 0.52 | 85.34 ± 1.07 | 84.77 ± 0.34 | 85.80 ± 0.60 | 74.78 |
| GraphGPT-noise | 80.63 ± 2.24 | 74.56 ± 3.07 | 94.09 ± 0.33 | 82.32 ± 1.97 | 67.59 ± 0.67 | 60.89 ± 1.61 | 85.70 ± 0.72 | 83.55 ± 0.58 | – | – |
| GraphGPT-noisetxt | 66.79 ± 3.84 | 57.37 ± 5.20 | 86.15 ± 1.88 | 64.56 ± 2.94 | – | – | – | – | – | – |
| LLaGA | 86.29 ± 1.76 | 75.81 ± 0.63 | 90.23 ± 0.59 | 84.88 ± 1.60 | 68.06 ± 0.49 | 68.53 ± 1.34 | 85.96 ± 0.53 | 86.96 ± 0.21 | 90.00 ± 0.20 | 74.52 |
| LLaGA-noise | 86.04 ± 2.18 | 75.55 ± 0.57 | 90.55 ± 0.57 | 84.14 ± 0.49 | 68.46 ± 0.59 | 68.62 ± 0.77 | 85.81 ± 0.38 | 86.57 ± 0.50 | – | – |
| LLaGA-noisefull | 83.70 ± 2.23 | 75.29 ± 1.01 | 89.13 ± 0.59 | 83.15 ± 1.64 | 68.31 ± 0.87 | 68.91 ± 0.48 | 85.52 ± 0.34 | 86.60 ± 0.39 | – | – |
| LLaGA-noisetxt | 83.64 ± 1.66 | 73.72 ± 1.89 | 86.37 ± 3.70 | 80.89 ± 1.20 | 68.15 ± 0.34 | 67.99 ± 2.25 | 84.11 ± 1.22 | 84.27 ± 1.78 | – | – |
| LLaGA-sim | 83.76 ± 1.03 | 75.08 ± 0.96 | 90.27 ± 0.19 | 82.39 ± 2.59 | 67.40 ± 0.46 | 56.75 ± 1.04 | 85.48 ± 0.85 | 79.51 ± 0.04 | – | – |
| SFT-auto | 83.27 ± 2.34 | 74.56 ± 1.19 | 94.74 ± 0.24 | 86.40 ± 0.69 | 68.39 ± 1.04 | 66.61 ± 1.44 | 86.08 ± 0.50 | 84.58 ± 0.09 | 83.58 ± 0.11 | – |
| SFT (w/o neighbor) | 82.96 ± 1.65 | 74.19 ± 0.48 | 94.98 ± 0.18 | 87.30 ± 0.20 | 69.27 ± 0.40 | 62.70 ± 0.50 | 86.51 ± 0.80 | 80.78 ± 0.35 | 77.19 ± 0.21 | 76.83 |
| SFT-neighbor | 88.25 ± 1.29 | 76.17 ± 1.81 | 95.08 ± 0.51 | 87.75 ± 0.46 | 69.24 ± 0.54 | 66.74 ± 1.06 | 86.81 ± 0.77 | 87.51 ± 0.07 | 89.74 ± 0.14 | 77.84 |
| SFT-noise | 85.30 ± 2.58 | 75.65 ± 1.89 | 95.13 ± 0.27 | 87.78 ± 0.30 | 69.78 ± 0.88 | 67.33 ± 1.09 | 86.53 ± 0.67 | 87.52 ± 0.13 | 89.65 ± 0.11 | 77.93 |
| SFT-noisefull | 87.45 ± 1.87 | 78.00 ± 1.42 | 94.96 ± 0.19 | 86.59 ± 0.20 | 69.39 ± 0.54 | 67.13 ± 0.68 | 86.31 ± 0.88 | 86.76 ± 0.12 | 89.37 ± 0.07 | 77.35 |
| SFT-noisetxt | 86.29 ± 1.26 | 76.18 ± 1.34 | 94.96 ± 0.23 | 86.70 ± 0.43 | 69.59 ± 0.50 | 66.99 ± 0.69 | 86.12 ± 0.47 | 86.78 ± 0.12 | 89.37 ± 0.19 | 77.60 |
| SFT-simp | 88.19 ± 1.58 | 76.38 ± 1.33 | 95.00 ± 0.08 | 87.84 ± 0.29 | 69.37 ± 0.33 | 66.23 ± 1.30 | 86.61 ± 0.71 | 87.53 ± 0.32 | 89.71 ± 0.04 | 77.68 |
| SFT-simf | 85.30 ± 2.04 | 76.02 ± 1.96 | 95.16 ± 0.39 | 87.61 ± 0.49 | 69.66 ± 0.46 | 51.78 ± 1.14 | 86.72 ± 0.87 | 83.12 ± 0.18 | – | – |

Table 41: GraphLLM's clean results in the transductive/poisoning setting.

| Method | Cora | CiteSeer | PubMed | WikiCS | Instagram | Reddit | History | Photo |
|---|---|---|---|---|---|---|---|---|
| GraphGPT | 68.97 ± 2.30 | 67.28 ± 1.56 | 92.13 ± 0.69 | 75.35 ± 0.16 | 63.21 ± 1.83 | 59.92 ± 0.06 | 83.46 ± 0.13 | 80.86 ± 0.55 |
| GraphGPT-noise | 69.71 ± 1.63 | 66.07 ± 0.06 | 91.87 ± 0.79 | 75.66 ± 0.12 | 55.44 ± 3.63 | 58.51 ± 0.58 | 83.09 ± 0.25 | 80.14 ± 0.89 |
| LLaGA | 78.94 ± 0.47 | 70.51 ± 2.98 | 85.60 ± 3.83 | 79.65 ± 1.15 | 61.56 ± 4.71 | 68.44 ± 1.39 | 83.55 ± 0.30 | – |
| LLaGA-noise | 79.53 ± 2.37 | 71.40 ± 1.76 | 84.71 ± 3.98 | 77.08 ± 1.59 | 66.65 ± 0.15 | 68.59 ± 0.37 | 82.94 ± 0.34 | 82.72 ± 0.52 |
| LLaGA-noisefull | 78.39 ± 3.82 | 71.82 ± 1.81 | 86.10 ± 1.46 | 78.39 ± 0.81 | 63.04 ± 4.14 | 68.18 ± 0.90 | 83.40 ± 0.42 | 83.40 ± 1.20 |
| LLaGA-noisetxt | 80.11 ± 3.95 | 71.26 ± 0.70 | 86.39 ± 1.17 | 79.60 ± 0.93 | 65.28 ± 2.14 | 67.31 ± 1.06 | 83.71 ± 0.22 | 84.38 ± 0.67 |
| SFT (w/o neighbor) | 75.45 ± 2.03 | 72.10 ± 0.94 | 93.93 ± 0.35 | 82.31 ± 0.49 | 66.53 ± 1.03 | 60.49 ± 1.02 | 84.78 ± 0.50 | – |
| SFT-neighbor | 79.40 ± 1.97 | 72.03 ± 2.03 | 93.80 ± 0.31 | 82.65 ± 0.83 | 66.85 ± 0.96 | 66.01 ± 0.57 | 85.72 ± 0.09 | 85.05 ± 0.57 |
| SFT-noise | 81.46 ± 2.39 | 71.88 ± 1.44 | 94.14 ± 0.35 | 83.12 ± 1.27 | 65.95 ± 1.18 | 66.02 ± 0.42 | 85.47 ± 0.12 | – |
| SFT-noisefull | 80.34 ± 1.32 | 70.79 ± 3.08 | 93.53 ± 0.49 | 82.86 ± 0.51 | 66.09 ± 1.83 | 66.17 ± 0.17 | 85.38 ± 0.13 | – |
| SFT-noisetxt | 76.92 ± 2.00 | 72.31 ± 0.49 | 93.46 ± 0.26 | 82.48 ± 0.12 | 66.23 ± 0.40 | 65.96 ± 0.74 | 85.07 ± 0.58 | – |
| SFT-simp | 78.80 ± 1.46 | 72.77 ± 1.22 | 94.06 ± 0.37 | 83.59 ± 0.61 | 66.37 ± 1.17 | 63.52 ± 2.06 | 85.54 ± 0.13 | – |

Table 42: GraphLLM's results under the inductive/evasion setting against structural attacks (ptb_rate=0.20).

| Method | Cora | CiteSeer | PubMed | WikiCS | Instagram | Reddit | History | Photo | Computer | ArXiv |
|---|---|---|---|---|---|---|---|---|---|---|
| GraphGPT | 77.68 ± 4.77 | 73.51 ± 0.72 | 93.62 ± 0.77 | 76.51 ± 5.85 | 65.31 ± 1.03 | 58.01 ± 0.74 | 81.59 ± 3.79 | 77.47 ± 3.32 | 76.12 ± 5.52 | 70.61 |
| GraphGPT-noise | 78.60 ± 5.05 | 73.98 ± 1.90 | 93.81 ± 0.44 | 78.87 ± 4.39 | 65.24 ± 1.37 | 58.01 ± 0.88 | 83.20 ± 2.69 | – | – | – |
| LLaGA | 75.21 ± 3.73 | 66.35 ± 1.25 | 87.38 ± 1.15 | 66.99 ± 5.07 | 66.33 ± 1.44 | 73.48 ± 5.39 | 81.24 ± 1.66 | 63.91 ± 4.11 | 62.32 ± 2.80 | 70.68 |
| LLaGA-noise | 76.26 ± 0.75 | 68.03 ± 1.57 | 88.29 ± 0.65 | 74.95 ± 1.94 | 67.73 ± 0.53 | 73.65 ± 6.02 | 82.34 ± 1.26 | 71.88 ± 0.63 | – | – |
| LLaGA-noisefull | 74.17 ± 2.05 | 66.35 ± 0.59 | 85.38 ± 0.84 | 62.40 ± 2.24 | 66.92 ± 1.73 | 75.16 ± 3.43 | 73.41 ± 2.06 | 66.48 ± 1.69 | – | – |
| LLaGA-noisetxt | 72.02 ± 1.26 | 65.41 ± 1.25 | 82.67 ± 0.57 | 51.17 ± 1.70 | 66.01 ± 1.31 | 74.44 ± 4.75 | 68.14 ± 0.39 | 57.96 ± 1.41 | – | – |
| LLaGA-sim | 81.92 ± 2.61 | 73.77 ± 0.48 | 89.43 ± 0.58 | 82.91 ± 0.46 | 68.21 ± 0.49 | 55.20 ± 1.68 | 85.57 ± 0.39 | 76.32 ± 0.42 | 73.99 ± 0.40 | 73.73 |
| SFT-auto | 82.59 ± 3.53 | 74.24 ± 2.14 | 92.29 ± 0.45 | 84.05 ± 0.69 | 66.93 ± 0.99 | 66.18 ± 2.74 | 85.54 ± 0.69 | 83.99 ± 0.14 | 83.62 ± 0.13 | – |
| SFT-neighbor | 82.35 ± 1.94 | 71.53 ± 1.86 | 94.73 ± 0.41 | 86.00 ± 1.40 | 68.25 ± 0.20 | 66.25 ± 3.31 | 85.98 ± 0.51 | 81.72 ± 0.34 | 82.39 ± 0.20 | 77.21 |
| SFT-noise | 79.52 ± 2.72 | 73.14 ± 0.36 | 94.51 ± 0.31 | 84.38 ± 0.60 | 68.45 ± 0.48 | 54.54 ± 2.95 | 85.31 ± 0.73 | 81.75 ± 0.12 | – | 77.49 |
| SFT-noisefull | 78.04 ± 0.85 | 70.17 ± 0.86 | 91.53 ± 0.55 | 82.07 ± 1.23 | 68.62 ± 1.01 | 62.39 ± 2.69 | 76.16 ± 2.83 | 79.00 ± 0.84 | – | 73.89 |
| SFT-noisetxt | 77.80 ± 2.51 | 68.18 ± 1.44 | 90.35 ± 0.51 | 79.98 ± 2.08 | 68.37 ± 0.72 | 62.07 ± 2.06 | 79.52 ± 2.36 | 78.20 ± 1.84 | 78.24 ± 1.07 | 74.38 |
| SFT-simp | 81.12 ± 2.77 | 72.26 ± 1.63 | 94.61 ± 0.17 | 86.39 ± 0.86 | 68.87 ± 0.36 | 64.27 ± 1.27 | 85.69 ± 0.23 | 81.63 ± 0.53 | 82.45 ± 0.42 | 77.11 |
| SFT-simf | 84.19 ± 2.72 | 75.29 ± 2.40 | 94.96 ± 0.36 | 86.00 ± 1.36 | 69.21 ± 0.60 | 64.88 ± 2.96 | 86.50 ± 0.89 | 81.23 ± 0.09 | – | – |

## J.4 RESULTS WITH DIFFERENT PERTURB RATIOS

In Tables 46, 47, 48 and 49, we present results against different attack ratios. In Figures 12 and 13, we further visualize the accuracy–perturbation curves across different datasets under structural and textual attacks in the inductive setting. The trends align with our observations in the main paper.

Table 43: GraphLLM's results under the transductive/poisoning setting against structural attacks. (ptb_rate=0.30)

| Method | Cora | CiteSeer | PubMed | WikiCS | Instagram | Reddit | History | Photo |
|---|---|---|---|---|---|---|---|---|
| GraphGPT | 64.53 ± 1.81 | 64.91 ± 1.55 | 90.02 ± 2.33 | 76.28 ± 0.95 | 58.95 ± 0.69 | 45.73 ± 0.66 | 79.82 ± 0.38 | 73.58 ± 1.47 |
| GraphGPT-noise | 65.70 ± 0.76 | 64.46 ± 2.67 | 90.60 ± 1.77 | 75.06 ± 0.70 | 58.13 ± 0.94 | 45.78 ± 0.35 | 80.82 ± 0.30 | 74.02 ± 0.37 |
| LLaGA | 71.02 ± 4.70 | 69.41 ± 1.42 | 81.63 ± 1.58 | 76.70 ± 1.25 | 58.94 ± 1.72 | 45.07 ± 0.76 | 79.00 ± 0.72 | 72.54 ± 2.18 |
| LLaGA-noise | 72.39 ± 1.59 | 71.41 ± 1.61 | 82.91 ± 1.36 | 79.07 ± 1.79 | 55.43 ± 10.44 | 46.90 ± 0.70 | 80.16 ± 0.28 | 73.43 ± 1.48 |
| LLaGA-noisefull | 64.22 ± 2.80 | 68.69 ± 2.53 | 81.11 ± 0.64 | 74.85 ± 3.57 | 59.83 ± 0.96 | 45.29 ± 0.57 | 73.68 ± 1.58 | 73.14 ± 0.59 |
| LLaGA-noisetxt | 61.92 ± 9.72 | 68.12 ± 1.98 | 81.05 ± 1.11 | 76.37 ± 1.81 | 57.36 ± 5.04 | 45.63 ± 1.36 | 72.64 ± 2.52 | 73.04 ± 0.93 |
| SFT-neighbor | 69.65 ± 4.42 | 71.99 ± 1.28 | 92.33 ± 0.62 | 82.00 ± 1.18 | 62.48 ± 2.60 | 51.90 ± 1.02 | 82.82 ± 1.61 | 80.51 ± 0.51 |
| SFT-noise | 70.56 ± 2.34 | 71.56 ± 1.82 | 92.84 ± 0.71 | 82.73 ± 0.61 | 60.95 ± 4.73 | 51.25 ± 0.86 | 83.38 ± 0.60 | 81.25 ± 0.82 |
| SFT-noisefull | 71.48 ± 4.09 | 71.78 ± 0.71 | 92.35 ± 0.86 | 81.92 ± 0.56 | 63.90 ± 0.24 | 51.39 ± 0.47 | 83.71 ± 0.29 | 81.16 ± 0.52 |
| SFT-noisetxt | 71.10 ± 1.19 | 70.52 ± 3.10 | 92.02 ± 0.47 | 81.35 ± 0.71 | 60.77 ± 5.23 | 53.60 ± 0.32 | 83.69 ± 0.67 | 80.74 ± 0.63 |
| SFT-simp | 70.45 ± 5.39 | 70.89 ± 1.16 | 92.65 ± 1.18 | 82.82 ± 1.41 | 63.89 ± 0.23 | 51.73 ± 0.63 | 83.48 ± 0.83 | 80.76 ± 0.48 |

Table 44: GraphLLM's results under the inductive/evasion setting against textual attacks. (ptb_rate=0.40)

| Method | Cora | CiteSeer | PubMed | WikiCS | Instagram | Reddit | History | Photo |
|---|---|---|---|---|---|---|---|---|
| GraphGPT | 66.48 ± 15.39 | 59.51 ± 13.47 | 71.21 ± 14.22 | 66.45 ± 17.17 | 63.38 ± 3.93 | 57.27 ± 3.57 | 70.53 ± 14.86 | 54.23 ± 2.63 |
| GraphGPT-noise | 51.66 ± 4.54 | 59.87 ± 13.78 | 70.36 ± 15.72 | 65.91 ± 18.43 | 62.99 ± 4.04 | 57.56 ± 2.34 | 70.26 ± 15.90 | – |
| LLaGA | 81.73 ± 1.76 | 61.65 ± 2.30 | 73.84 ± 1.08 | 61.14 ± 3.87 | 63.40 ± 3.54 | 68.83 ± 1.61 | 72.55 ± 1.70 | 77.58 ± 1.62 |
| LLaGA-noise | 80.88 ± 1.67 | 61.23 ± 3.46 | 70.82 ± 1.00 | 51.29 ± 1.55 | 59.44 ± 1.15 | 68.21 ± 2.07 | 71.15 ± 1.53 | 67.24 ± 1.28 |
| LLaGA-noisefull | 79.89 ± 2.13 | 61.65 ± 1.04 | 77.12 ± 0.21 | 63.40 ± 1.99 | 62.14 ± 2.41 | 68.59 ± 0.56 | 78.06 ± 0.94 | 74.93 ± 1.80 |
| LLaGA-noisetxt | 78.54 ± 0.93 | 59.19 ± 0.80 | 79.51 ± 1.73 | 67.92 ± 2.92 | 64.70 ± 1.35 | 69.10 ± 1.38 | 78.15 ± 0.69 | 77.00 ± 1.73 |
| LLaGA-sim | 52.21 ± 1.66 | 39.08 ± 0.77 | 65.68 ± 0.42 | 47.10 ± 0.36 | 55.12 ± 2.67 | 52.03 ± 0.71 | 60.24 ± 4.61 | 47.08 ± 0.58 |
| SFT-auto | 76.75 ± 1.76 | 65.36 ± 3.29 | 77.86 ± 1.38 | 77.83 ± 0.55 | 61.70 ± 0.84 | 65.68 ± 0.87 | 73.55 ± 0.83 | 77.96 ± 1.76 |
| SFT-neighbor | 75.65 ± 1.33 | 43.84 ± 2.14 | 72.27 ± 3.53 | 51.53 ± 1.24 | 64.33 ± 1.46 | 65.52 ± 1.24 | 64.84 ± 0.98 | 63.76 ± 1.87 |
| SFT-noise | 71.71 ± 8.20 | 43.00 ± 3.42 | 72.10 ± 0.98 | 50.63 ± 0.50 | 63.67 ± 0.12 | 66.00 ± 0.86 | 59.50 ± 4.01 | 62.88 ± 1.47 |
| SFT-noisefull | 77.06 ± 5.97 | 52.46 ± 5.80 | 74.72 ± 1.06 | 66.11 ± 6.06 | 64.20 ± 1.13 | 65.64 ± 0.81 | 67.22 ± 3.19 | 72.36 ± 0.70 |
| SFT-noisetxt | 75.64 ± 0.49 | 51.67 ± 1.22 | 76.35 ± 1.50 | 68.27 ± 6.29 | 64.78 ± 0.93 | 65.26 ± 1.23 | 67.49 ± 0.76 | 74.71 ± 1.30 |
| SFT-simp | 75.03 ± 4.71 | 43.57 ± 3.46 | 74.66 ± 2.67 | 51.16 ± 0.86 | 63.54 ± 1.87 | 64.49 ± 1.37 | 66.37 ± 3.40 | 64.79 ± 0.70 |
| SFT-simf | 53.38 ± 1.29 | 40.28 ± 1.88 | 71.66 ± 3.38 | 51.52 ± 1.24 | 63.70 ± 2.10 | 65.92 ± 0.86 | 59.42 ± 1.24 | 51.24 ± 1.35 |

Table 45: GraphLLM's results in the transductive/poisoning setting against textual attacks. (ptb_rate=0.80)

| Method | Cora | CiteSeer | PubMed | WikiCS | Instagram | Reddit | History |
|---|---|---|---|---|---|---|---|
| GraphGPT | 63.54 ± 2.39 | 40.82 ± 11.87 | 84.52 ± 1.46 | 61.96 ± 3.44 | 42.13 ± 3.52 | 46.42 ± 2.04 | 73.18 ± 1.18 |
| GraphGPT-noise | 62.94 ± 1.39 | 41.01 ± 5.32 | 86.25 ± 2.95 | 62.19 ± 1.42 | 38.71 ± 1.55 | 46.83 ± 2.65 | 70.51 ± 0.98 |
| LLaGA | 78.96 ± 2.60 | 52.52 ± 1.31 | 72.16 ± 8.62 | 75.60 ± 3.10 | 36.33 ± 0.26 | 56.57 ± 6.62 | 75.57 ± 4.36 |
| LLaGA-noise | 79.05 ± 3.07 | 51.87 ± 10.89 | 78.33 ± 5.89 | 71.42 ± 2.64 | 36.82 ± 0.78 | 47.41 ± 3.08 | 71.54 ± 2.20 |
| LLaGA-noisefull | 77.66 ± 2.19 | 50.24 ± 5.85 | 80.16 ± 1.53 | 70.41 ± 7.15 | 42.64 ± 4.51 | 62.39 ± 2.86 | 75.22 ± 5.33 |
| LLaGA-noisetxt | 76.27 ± 4.31 | 51.96 ± 4.09 | 81.46 ± 2.41 | 73.94 ± 3.49 | 41.82 ± 2.11 | 63.95 ± 1.00 | 76.16 ± 3.77 |
| SFT-neighbor | 72.40 ± 4.62 | 46.44 ± 4.32 | 93.07 ± 0.55 | 79.33 ± 1.10 | 62.11 ± 2.91 | 59.13 ± 6.06 | 79.88 ± 3.15 |
| SFT-noise | 73.99 ± 3.44 | 47.89 ± 4.41 | 93.49 ± 0.26 | 80.04 ± 0.41 | 61.86 ± 3.34 | 60.59 ± 2.97 | 82.07 ± 0.86 |
| SFT-noisefull | 74.59 ± 1.40 | 47.33 ± 10.74 | 92.94 ± 0.43 | 79.11 ± 1.12 | 63.88 ± 0.22 | 63.26 ± 0.57 | 81.46 ± 1.58 |
| SFT-noisetxt | 72.73 ± 4.19 | 43.87 ± 4.94 | 93.07 ± 0.47 | 79.21 ± 1.17 | 63.35 ± 0.64 | 61.62 ± 1.54 | 80.24 ± 0.88 |
| SFT-simp | 76.64 ± 1.70 | 46.92 ± 6.90 | 93.04 ± 0.22 | 79.51 ± 1.76 | 55.40 ± 14.53 | 60.41 ± 2.40 | 79.38 ± 1.78 |

Table 46: Accuracy under the inductive/evasion setting against structural attack. (ptb_rate=0.1, atk_emb=BoW, def_emb=RoBERTa)

| Method | Cora | CiteSeer | PubMed | WikiCS | Instagram | Reddit | History | Photo | Computer | ArXiv | Avg Rank |
|---|---|---|---|---|---|---|---|---|---|---|---|
| GCN | 77.24 ± 2.39 | 64.99 ± 1.81 | 84.58 ± 0.38 | 45.55 ± 0.84 | 59.41 ± 0.81 | 62.89 ± 1.72 | 70.36 ± 0.50 | 66.13 ± 0.85 | 64.39 ± 0.03 | 52.21 ± 0.00 | 10.00 |
| GAT | 77.80 ± 0.97 | 67.35 ± 1.41 | 84.41 ± 0.66 | 58.89 ± 0.37 | 62.70 ± 0.78 | 60.87 ± 1.07 | 72.41 ± 1.17 | 66.73 ± 0.30 | 64.82 ± 0.59 | 60.07 ± 0.00 | 7.80 |
| APPNP | 81.80 ± 2.05 | 72.73 ± 1.44 | 89.78 ± 0.28 | 78.76 ± 1.08 | 64.15 ± 1.01 | 56.53 ± 0.59 | 80.84 ± 0.39 | 70.76 ± 0.44 | 64.35 ± 0.23 | 62.73 ± 0.00 | 5.10 |
| GPRGNN | 81.80 ± 2.32 | 72.15 ± 0.70 | 89.96 ± 0.20 | 78.29 ± 0.98 | 60.95 ± 0.38 | 59.97 ± 1.01 | 82.50 ± 0.32 | 71.98 ± 0.51 | 69.51 ± 0.67 | 68.59 ± 0.00 | 4.20 |
| RobustGCN | 76.75 ± 2.27 | 65.57 ± 1.07 | 83.72 ± 0.36 | 51.94 ± 1.23 | 64.79 ± 0.79 | 57.29 ± 1.36 | 69.21 ± 0.40 | 64.10 ± 0.36 | 60.71 ± 0.99 | 57.20 ± 0.00 | 10.00 |
| GCORN | 77.86 ± 1.05 | 66.46 ± 0.66 | 85.38 ± 0.08 | 46.55 ± 0.26 | 59.69 ± 1.00 | 66.91 ± 1.50 | 65.15 ± 0.69 | 62.61 ± 0.11 | 56.28 ± 0.23 | 49.74 ± 0.00 | 9.70 |
| NoisyGCN | 77.31 ± 1.74 | 65.78 ± 1.58 | 84.46 ± 0.47 | 46.65 ± 0.79 | 60.95 ± 1.02 | 63.26 ± 1.37 | 70.52 ± 0.47 | 66.29 ± 0.72 | 64.75 ± 0.21 | 51.40 ± 0.00 | 8.80 |
| GRAND | 81.73 ± 1.45 | 72.68 ± 0.83 | 87.04 ± 0.25 | 75.78 ± 1.08 | 65.29 ± 0.25 | 63.45 ± 1.51 | 79.41 ± 0.33 | 74.37 ± 0.07 | 71.06 ± 0.33 | 70.93 ± 0.00 | 3.70 |
| SoftmedianGDC | 81.80 ± 2.70 | 71.47 ± 1.55 | 87.82 ± 0.08 | 81.57 ± 0.75 | 62.18 ± 0.65 | – | 80.89 ± 0.39 | – | – | – | 5.00 |
| EvenNet | 80.75 ± 1.06 | 72.78 ± 1.22 | 88.44 ± 0.26 | 70.94 ± 0.60 | 66.80 ± 0.16 | 62.01 ± 2.06 | 76.49 ± 0.12 | 69.84 ± 0.30 | 68.87 ± 0.20 | 59.48 ± 0.00 | 4.90 |
| ElasticGNN | 78.23 ± 1.38 | 65.46 ± 1.03 | 85.42 ± 0.27 | 66.61 ± 0.63 | 60.05 ± 3.30 | 56.66 ± 0.77 | 74.21 ± 0.59 | 70.20 ± 0.15 | 63.96 ± 0.58 | 60.48 ± 0.00 | 8.30 |
| GNNGuard | 83.64 ± 1.94 | 73.67 ± 0.22 | 90.26 ± 0.33 | 83.10 ± 0.31 | 63.93 ± 0.57 | 58.52 ± 0.42 | 84.78 ± 0.42 | 69.12 ± 0.65 | 68.60 ± 0.44 | 71.26 ± 0.00 | 3.40 |
| RUNG | 84.75 ± 1.15 | 74.09 ± 0.70 | 90.64 ± 0.19 | 83.38 ± 0.59 | 65.18 ± 1.08 | 58.50 ± 0.53 | – | – | – | – | 2.67 |

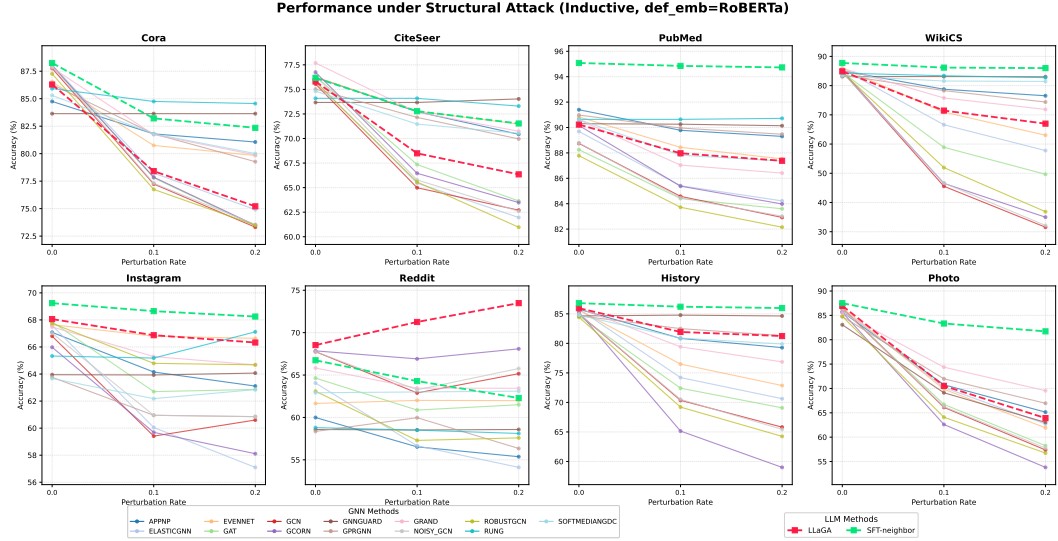

Figure 12: Performance under structural attacks (inductive setting). Accuracy trends on eight datasets as the perturbation rate increases. Each curve corresponds to a different GNN/RGNN/-GraphLLM method.

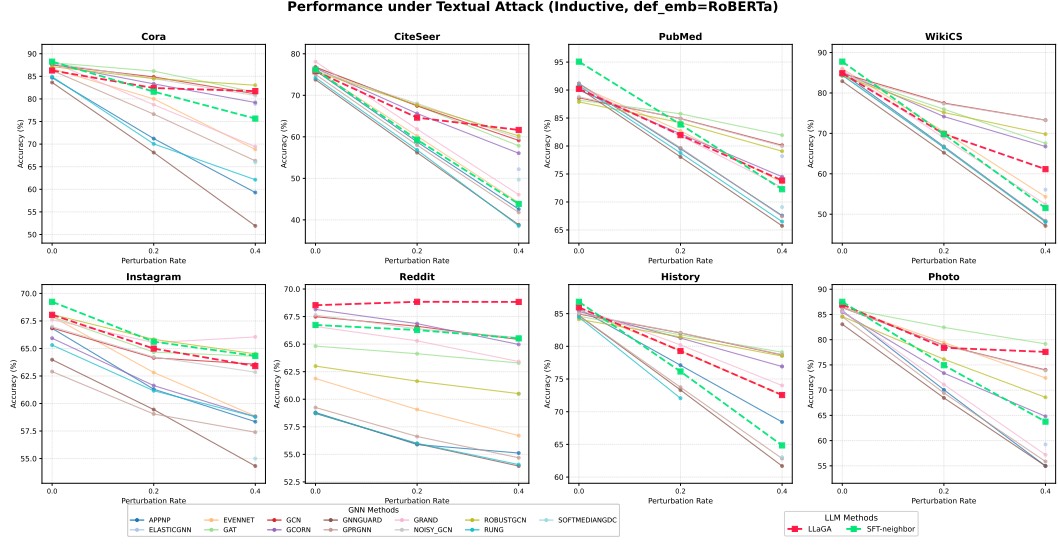

Figure 13: Performance under textual attacks (inductive setting). Accuracy trends on eight datasets as the perturbation rate increases. Each curve corresponds to a different GNN/RGNN/GraphLLM method.

Table 47: Accuracy under the inductive/evasion setting against textual attack. (ptb_rate=0.2, def_emb=RoBERTa)

| Method | Cora | CiteSeer | PubMed | WikiCS | Instagram | Reddit | History | Photo | Avg Rank |
|--------|------|----------|--------|--------|-----------|--------|---------|-------|----------|
| GCN | 84.87 ± 0.40 | 67.45 ± 0.63 | 84.90 ± 0.30 | **77.50 ± 0.45** | 64.15 ± 0.45 | 66.62 ± 0.67 | 82.05 ± 0.28 | 78.92 ± 0.19 | 2.62 |
| GAT | **86.16 ± 1.83** | 67.55 ± 1.34 | **85.76 ± 0.74** | 75.98 ± 2.19 | 64.67 ± 1.06 | 64.13 ± 0.20 | 81.76 ± 0.93 | **82.44 ± 0.33** | 2.50 |
| APPNP | 71.22 ± 1.20 | 58.78 ± 0.66 | 79.51 ± 0.75 | 66.75 ± 0.56 | 61.30 ± 0.42 | 55.89 ± 0.67 | 77.11 ± 0.26 | 70.07 ± 0.05 | 8.62 |
| GPRGNN | 76.63 ± 0.71 | 57.99 ± 0.46 | 79.68 ± 0.60 | 66.62 ± 0.67 | 59.05 ± 2.20 | 56.62 ± 0.84 | 73.74 ± 1.08 | 69.45 ± 0.69 | 8.88 |
| RobustGCN | 84.50 ± 0.84 | 67.66 ± 0.52 | 84.17 ± 0.57 | 75.17 ± 1.68 | **65.83 ± 0.16** | 61.64 ± 0.71 | 81.44 ± 0.66 | 76.13 ± 0.83 | 3.62 |
| GCORN | 83.33 ± 0.53 | 65.62 ± 0.37 | 82.18 ± 0.05 | 74.16 ± 0.55 | 61.63 ± 0.54 | **66.86 ± 0.49** | 81.26 ± 0.66 | 73.37 ± 0.16 | 5.00 |
| NoisyGCN | 84.50 ± 0.40 | **68.03 ± 1.48** | 84.84 ± 0.48 | 77.33 ± 0.49 | 64.23 ± 0.94 | 66.39 ± 1.07 | **82.08 ± 0.38** | 78.86 ± 0.12 | 2.62 |
| GRAND | 78.72 ± 0.38 | 61.86 ± 1.10 | 81.83 ± 0.53 | 68.86 ± 0.82 | 65.53 ± 0.67 | 65.30 ± 1.00 | 80.21 ± 0.43 | 71.11 ± 0.15 | 5.75 |
| EvenNet | 80.01 ± 1.48 | 60.19 ± 1.12 | 82.69 ± 0.78 | 70.41 ± 1.53 | 62.82 ± 1.36 | 59.07 ± 0.90 | 79.32 ± 0.63 | 79.44 ± 0.21 | 5.75 |
| GNNGuard | 68.14 ± 1.32 | 56.22 ± 0.07 | 78.05 ± 0.97 | 65.23 ± 0.41 | 59.45 ± 0.77 | 55.96 ± 0.29 | 73.32 ± 0.50 | 68.47 ± 0.25 | 10.50 |
| RUNG | 70.05 ± 1.26 | 56.84 ± 0.53 | 78.76 ± 0.38 | 66.47 ± 0.50 | 61.16 ± 0.57 | 56.00 ± 0.77 | – | – | 9.67 |

Table 48: GraphLLM's results under the inductive/evasion setting against structural attacks. (ptb_rate=0.10)

| Method | Cora | CiteSeer | PubMed | WikiCS | Instagram | Reddit | History | Photo |
|---|---|---|---|---|---|---|---|---|
| LLaGA | $78.41 \pm 0.64$ | $68.49 \pm 1.50$ | $87.97 \pm 1.28$ | $71.46 \pm 3.37$ | $66.87 \pm 1.03$ | $71.26 \pm 5.87$ | $81.93 \pm 1.56$ | $70.53 \pm 2.44$ |
| SFT-neighbor | $83.21 \pm 1.21$ | $72.78 \pm 2.28$ | $94.85 \pm 0.49$ | $86.17 \pm 0.90$ | $68.65 \pm 0.28$ | $64.29 \pm 2.96$ | $86.21 \pm 0.64$ | $83.33 \pm 0.60$ |

Table 49: GraphLLM's results under the inductive/evasion setting against textual attacks. (ptb_rate=0.20)

| Method | Cora | CiteSeer | PubMed | WikiCS | Instagram | Reddit | History | Photo |
|---|---|---|---|---|---|---|---|---|
| LLaGA | $82.41 \pm 1.86$ | $64.63 \pm 1.81$ | $81.94 \pm 1.12$ | $69.86 \pm 2.27$ | $64.99 \pm 1.35$ | $68.84 \pm 1.97$ | $79.29 \pm 1.14$ | $78.40 \pm 0.64$ |
| SFT-neighbor | $81.61 \pm 0.21$ | $59.30 \pm 2.28$ | $83.85 \pm 1.25$ | $69.80 \pm 0.64$ | $65.62 \pm 1.10$ | $66.28 \pm 1.35$ | $76.15 \pm 1.15$ | $74.94 \pm 0.51$ |

# K  ADDITIONAL BASELINE EVALUATION

To further validate the persistence of the text-structure robustness trade-off, we evaluate three recent methods: GPRGNN-AT Gosch et al. (2023) (with PR-BCD training), GOOD-AT Li et al. (2024), and GPR-GAE Lee & Park (2025). Tables 50 and 51 present results under inductive structural and textual attacks, respectively.

Table 50: Performance under inductive structural attacks (PGD 20%). Format: Clean/Attacked (%).

| Method | Cora | CiteSeer | PubMed | WikiCS |
|---|---|---|---|---|
| GCN | 87.76/73.31 | 76.07/62.70 | 88.76/82.92 | 84.86/31.54 |
| GPRGNN | 86.29/79.27 | 74.19/69.96 | 91.05/89.47 | 84.38/74.38 |
| GPRGNN-AT | 86.10/81.92 | 74.03/71.32 | 90.61/90.49 | 83.11/81.43 |
| GOOD-AT | 88.44/73.68 | 76.18/63.27 | 88.83/82.99 | 84.85/32.09 |
| GPR-GAE | 87.82/79.03 | 76.44/**74.71** | 88.48/88.39 | 84.19/41.73 |
| SFT-Auto | 83.27/**82.59** | 74.56/74.24 | 94.74/**92.29** | 86.40/**84.05** |

Table 51: Performance under inductive textual attacks (GPT 40%). Format: Clean/Attacked (%).

| Method | Cora | CiteSeer | PubMed | WikiCS |
|---|---|---|---|---|
| GCN | 87.76/81.12 | 76.07/59.14 | 88.76/80.13 | 84.86/73.27 |
| GPRGNN | 86.29/66.30 | 74.19/41.80 | 91.05/67.43 | 84.38/48.40 |
| GPRGNN-AT | 86.10/65.68 | 74.03/39.71 | 90.61/66.78 | 83.11/47.20 |
| GOOD-AT | 88.44/**83.03** | 76.18/59.93 | 88.83/**80.35** | 84.85/75.35 |
| GPR-GAE | 87.82/67.10 | 76.44/40.07 | 88.48/70.86 | 84.19/66.38 |
| SFT-Auto | 83.27/76.75 | 74.56/**65.36** | 94.74/77.86 | 86.40/**77.83** |

**Key Observations:** The results confirm that the text-structure trade-off persists in adversarial training methods. GPR-GAE and GPRGNN-AT significantly improve structural robustness but *degrade* textual robustness. Conversely, GOOD-AT excels under textual attacks (83.03% on Cora) but collapses under structural attacks (32.09% on WikiCS). This validates our finding that existing defenses exhibit inherent trade-offs, while SFT-Auto achieves balanced robustness across both modalities.

