# OpenReview forum: "Robustness in Text-Attributed Graph Learning: Insights, Trade-offs, and New Defenses"
_ICLR.cc/2026/Conference — ICLR 2026 Poster_

### Official Review · Reviewer_jx3n · 2025-10-17

**Soundness:** 2
**Presentation:** 3
**Contribution:** 2
**Rating:** 2
**Confidence:** 4

**Summary:**

This paper provides a comprehensive evaluation framework to assess the adversarial robustness of various models on Text-Attributed Graphs. Their large-scale analysis reveals the text-structure robustness trade-off in RGNNs and GraphLLMs, showing that models are typically strong against either structural or textual attacks, but not both. To overcome the trade-off and achieve a balanced robustness, the authors propose SFT-auto, a 2-staged framework that first detect the structural and textual attacks, then proceeds to an adaptive recovery.

**Strengths:**

* Authors provide a comprehensive, large-scale empirical evaluation across a wide range of models, datasets, and attack settings.
* The writing and the figures are well-structured and easy to follow.
* Various experiments provided in appendix.

**Weaknesses:**

* Some of the statements are wrongly addressed:

     * In Section 3.1, it says that the spectral methods demonstrate superior performance against poisoning attacks while referencing [1]. However, [1] does not include any experiments or analysis on poisoning setting.
     * In Section 3.2, GNNGuard and RUNG are marked as spectral methods, while both are spatial methods.

* More recent work such as [2,3] could be included for the "improving structure" category for a more reliable empirical evaluation Also, while the authors reference [1] within the paper, it is not included as a baseline.


* My primary concern is the novelty of SFT-auto. While the authors state that it overcomes the trade-off, it seems more like an incremental mergence of two simple defense methods against structure and text attacks. Moreover, while the authors provide a comprehensive evaluation framework for both poisoning and evasion setting, their suggested method to overcome the trade-off, SFT-auto, is only applicable to the evasion setting.




[1] Adversarial Training for Graph Neural Networks: Pitfalls, Solutions, and New Directions \
[2] Boosting the Adversarial Robustness of Graph Neural Networks: An OOD Perspective \
[3] Self-supervised Adversarial Purification for Graph Neural Networks

**Questions:**

* Could the authors clarify on how SFT-auto leverages the LLM's reasoning to defend against structural attacks? The paper describes the detection mechanism for structure attacks as using embedding-based cosine similarity, which seems separate from the LLM's reasoning as it would have to use another text encoding model. This appears to contradict the claim that the LLM's capabilities are used to defend against both attack types within a single model.

---

> ### Author Response · Authors · 2025-11-17
> **Response to Reviewer jx3n**
>
> We appreciate the constructive feedback and address each concern below.
>
> ---
>
> ### **W1: Incorrect References and Categorization**
>
> Both errors have been corrected in the **revised PDF**:
>
> **Section 3.1:** We incorrectly referenced poisoning attacks—our finding concerns **transductive scenarios**. We revised the PDF for a more accurate statement.
>
> **Section 3.2:** GNNGuard and RUNG are correctly categorized as spatial methods in Table 2. The typo is fixed.
>
> ---
>
> ### **W2: Missing Recent Baselines**
>
> We evaluated **GOOT-AT** and **GPR-GAE**. Results confirm the trade-off persists:
>
> **Inductive Structure Attack (PGD 20%)**
>
> |Method|Cora|CiteSeer|PubMed|WikiCS|
> |-|-|-|-|-|
> |GCN|87.76/73.31|76.07/62.70|88.76/82.92|84.86/31.54|
> |GNNGuard|83.64/**83.64**|74.03/74.03|90.27/90.14|83.07/83.03|
> |GPRGNN|86.29/79.27|74.19/69.96|91.05/89.47|84.38/74.38|
> |SFT-Auto|83.27/82.59|74.56/74.24|94.74/**92.29**|86.40/**84.05**|
> |GOOT-AT|88.44/73.68|76.18/63.27|88.83/82.99|84.85/32.09|
> |GPR-GAE|87.82/79.03|76.44/**74.71**|88.48/88.39|84.19/41.73|
> |GPRGNN-AT|86.10/81.92|74.03/71.32|90.61/90.49|83.11/81.43|
>
> **Inductive Text Attack (GPT 40%)**
>
> |Method|Cora|CiteSeer|PubMed|WikiCS|
> |-|-|-|-|-|
> |GCN|87.76/81.12|76.07/59.14|88.76/80.13|84.86/73.27|
> |GNNGuard|83.64/51.91|74.03/38.82|90.27/65.73|83.07/47.12|
> |GPRGNN|86.29/66.30|74.19/41.80|91.05/67.43|84.38/48.40|
> |SFT-Auto|83.27/76.75|74.56/**65.36**|94.74/77.86|86.40/**77.83**|
> |GOOT-AT|88.44/**83.03**|76.18/59.93|88.83/**80.35**|84.85/75.35|
> |GPR-GAE|87.82/67.10|76.44/40.07|88.48/70.86|84.19/66.38|
> |GPRGNN-AT|86.10/65.68|74.03/39.71|90.61/66.78|83.11/47.20|
>
>
> *Format: Clean/Attacked. Bold: best robustness.*
>
> **Key finding:** GOOT-AT excels on text (83.03%) but **collapses on structure** (32.09%); GPRGNN-AT/GPR-GAE handle structure well but **fail on text** (39.71%/40.07%). SFT-Auto achieves balanced performance.
>
> ---
>
> ### **W3: Limited Novelty of SFT-auto**
>
> SFT-Auto's novelty emerges from a **complete discovery-to-solution chain**:
>
> **Discovery (§3.1-3.2, §5.1):** We reveal that structure robustness is achievable via simple and efficient filtering given good text encodings (GNNGuard, SFT-simf succeed; and SFT-neighbor is not bad inherently), but **these SFT-variants still fail against text attacks**—even naive GraphLLM adaptations (Figure 5).
>
> **Natural Solution (§5.2-5.3):** Since filtering solves structure but text remains vulnerable, we exploit LLM's **unique text reasoning capability** for attack detection and recovery. This design is novel—**no prior work leverages LLM reasoning specifically for text attack defense**.
>
> **Key Insight (§5.3):** We demonstrate that **LLMs achieve 6.2-17.4× better text attack detection** than GNNs (vs. GCN-auto) while maintaining $O(|V|)$ complexity. This finding establishes LLMs as a promising direction for balanced robustness.
>
> Although SFT-auto is straightforward, our **initial endeavor** provides valuable insights and serves as a proof of concept to motivate future, more sophisticated explorations.
>
> **Regarding poisoning:** Our trade-off discovery stems from §3.1-3.2's **inductive/evasion observations**, where the phenomenon is most pronounced. While poisoning reveals other insights (§3.2: GraphLLM vulnerability), the trade-off is less distinct there. We've clarified this in the revised pdf.
>
> ---
>
> ### **Q: How does SFT-Auto leverage LLM reasoning for structural attacks?**
>
> SFT-Auto uses **embedding-based filtering** for structure for three reasons:
>
> 1. **Simplicity suffices:** §3.1 shows filtering with quality embeddings works well; text is the remaining challenge. Complex structural reasoning would be unnecessary based on our analysis.
>
> 2. **Efficiency:** Structural reasoning (e.g., LLM4RGNN) requires $O(|E|)$ or $O(|V|^2)$ LLM queries—prohibitively expensive.
>
> 3. **Multi-task interference:** Fine-tuning a single LLM to simultaneously perform text and edge filtering risks compromising performance on both tasks, while using separate models (LLM4RGNN trains both LLM+GNN) defeats the purpose of building a unified defense framework.
>
> **Additional structural design:** Detected text-attacked nodes also inform structure defense—we prevent them from appearing in others' neighborhoods, yielding:
>
> |Dataset|Text (with→w/o)|Structure (with→w/o)|
> |-|-|-|
> |Cora|76.75→76.47|82.59→82.57|
> |CiteSeer|65.36→65.05|74.24→74.26|
>
> When texts are attacked, this structural mitigation provides additional benefits while maintaining performance under non-attacked conditions.
>
> ---
>
> ### **Summary**
>
> Thanks for your time and effort in reviewing. We've revised the PDF following your advice.
>
> Beyond SFT-Auto, our work contributes: **(1)** Comprehensive evaluation (Table 1); **(2)** Novel observations on vulnerability (§3.1, §3.2, App. G) and the text-structure trade-off; **(3)** RGNN+LLM exploration revealing architectural advantages (§5.1, §5.3).
>
> We believe these findings advance understanding of TAG robustness and hope you will consider the breadth of our contributions.

---

### Official Review · Reviewer_Bb1t · 2025-10-26

**Soundness:** 2
**Presentation:** 2
**Contribution:** 3
**Rating:** 4
**Confidence:** 4

**Summary:**

The paper presents a comprehensive empirical study of GNNs, robust GNNs and Graph LLMs under a unified adversarial setup allowing fair comparions. In particular, the dimensions of perturbing structure vs perturbing text have been investigated. Several insights have been derived such as a text-structure robustnes tradeoff. Further, a method is proposed that has a superior text-structure tradeoff compared to other methods.

**Strengths:**

* Investigating the disting effect of textual perturbations (compared to small-epsilon perturbations) for node features is interesting and more realistic compared to pervious studies.
* The text-structure tradeoff is an interesting insight and the proposed method a good remedy against.
* There are many empirical results and insights. However, given its a purely empirical paper, this is also in a way a necessity.
* Code provided.
* Comparing robust GNNs with Graph LLMs in a (unified) adversarial setting is interesting and important.

**Weaknesses:**

1. Comparison Table 1 feels outdates and slightly misleading. On GNNs & RGNNs, the table only mentions the quite old graph robustness benchmark, while they don't mention recent works on GNNs & RGNNs. Exemplary, the cited work by Gosch et al. 2023 in the paper includes 8 datasets, GMA, evasion & inductive & adaptive attacks but is not mentioned, instead the 5 datsets of the GRB highlighted. I do think that the breath of experiments in submitted work is quite good, but I don't think for metrics such as num. datasets or num. domains the work did a comprehensive and fair survey of existing works, which would be necessary if one quantitavely compares these numbers to previous works.
2. Lines 160-183 mentions sufficiently strong attacks to study model robustness but does not mention adaptive attacks, which are the gold-standard "sufficiently strong attacks" to evaluate methods [1, 2]. I also do not understand the focus in the threat model on transfer attacks, as they have been shown to lead to misleading robustness evaluations in prior work.
3. Regarding robust training, I'm missing a comparison to adversarially training. In particular, the in the paper chosen GNN model GPR-GNN was shown to be particularly strong against structure perturbations after adversarial training with PR-BCD (Gosch et al. 2023). Thus, a comparison would be interesting.
4. Evasion textual attacks replaced 40% of test set nodes with LLM-text and poisoning textual attacks 80% fo the training nodes. These chooses seem quite strong and I think a gradual investigation (e.g., against 1%, 5%, 10%, 20%, ...) would be more interesting then setting one fixed attack budget.
5. Confusing Paper Structure. Results are presented before the method. E.g., the first results paragraph presents findings on "SFT-neighbor", though the method is not introduced. Even though the findings motivate the development of "SFT-neighbor", I do think that a more classic structure Intro -> Method -> Results would be sufficient with some motivating results for the method mentioned in the intro / beginning of methods, or at least introducing the method in some way more than just a short mention in the introduction.
6. I'm unsure how representative the results of comparing structure attacks for GNNs with an applied adaptive attack (PGD/GRBCD) is compared to a GraphLLM without an adaptive attack and thus, don't know if the statement "GraphLLMs demonstrate inehrent robustnes against evasion attacks." truly holds under an adaptive attack setting.
7. I think the main draft would benefit from a figure that doesn't only compare "rankings" but also some classic "Accuracy vs Attack Budget" plot, one for evasion and one for poisoning.

[1] Tramèr et al. "On adaptive attacks to adversarial example defenses", NeurIPS 2020
[2] Mujkanovic et al. "Are defenses for graph neural networks robust?", NeurIPS 2022

**Questions:**

1. Why did the authors focus on transfer attacks and not adaptive attacks?
2. Line 272: I would say the results are well "complementing the findings in Gosch et al. (2023)" (or something similar) and not "aligning with" as Gosch et al. (2023) investigated evasion and not poisoning. Or alternatively, make explicit that Gosch et al. (2023) shows high robustness of these models for evasion, e.g. through stating "aligning with findings in (Gosch et al., 2023) that show strong performance of these models against evasion attacks".
3. How would SFT-Auto perform in Figure 5?
4. Given many investigated datasets (e.g., the citation networks) are usually distributed with classic BoW embedding, how have the original texts been obtained for computing the rankings against textual attacks such as in Figure 3?

*Minor:*
* Adjust Line 124 (e.g. through reducing spaces) to fit the margins of the text.
* Line 161: Mettack -> Metaattack
* Provide References in line 80/81 for "proven effectivenss" of noise-injection & similarity-filterings

---

> ### Author Response · Authors · 2025-11-17
> **Response to Reviewer Bb1t**
>
> We appreciate your detailed and constructive feedback and address each concern below.
>
> ---
>
> ### **W1: Comparison Table 1 and Related Work**
>
> Our work focuses on TAGs, so Table 1 initially compares TAG-specific or explicit non-text graph benchmarks. We have **revised Table 1** to properly acknowledge Gosch et al. (2023)'s excellent work while clarifying the scope difference.
>
> Regarding domain diversity: our findings in Figure 4(b) and Appendix I.2 show that robustness patterns can be **domain-dependent**. Results in social networks, review networks, and citation networks slightly differ from each other. We believe validation across domains provides more solid results.
>
> ---
>
> ### **W2/Q1/W6: Transfer vs. Adaptive Attacks**
>
> We believe both adaptive attacks and transfer attacks are important and worth studying. Our emphasis on transfer attacks stems from:
>
> 1. **Practical relevance**: In real-world TAGs (social networks, citation graphs), adversaries typically poison data (fake posts/papers) without knowing the deployed model. Defenders can also deploy personalized models and switch easily. Transfer attacks capture this realistic threat model.
>
> 2. **Foundational necessity**: We view robustness against transfer attacks as a fundamental prerequisite; models that fail in this setting are likely vulnerable under more intensive adaptive attacks. Furthermore, designing effective adaptive attacks for GraphLLMs is highly non-trivial, a challenge underscored by the lack of such explorations in prior TAG robustness benchmarks. Since existing TAG works remain incomplete even in the transfer setting (as shown in Table 1), we prioritized transfer attacks first.
>
> We acknowledge the importance of adaptive attacks and provide an initial exploration in **Appendix H** with novel text-based adaptive attacks and Guard experiments.
> We believe our work also provides a foundation for future GraphLLM attack research, as we've already identified distinct preferences across GraphLLM variants that could inform adaptive strategies.
>
> **We've revised our claims** to specify: "GraphLLMs show inherent robustness against **non-adaptive** inductive/evasion attacks."
>
> ---
>
> ### **W3: Adversarial Training Comparison**
>
> We evaluated **GPRGNN with PR-BCD adversarial training** (extended version with more baselines in reply to reviewer jx3n). Results confirm **the text-structure trade-off persists**:
>
> **Inductive Structure Attack (PGD 20%)**
>
> |Method|Cora|CiteSeer|PubMed|WikiCS|
> |-|-|-|-|-|
> |GPRGNN|86.29/79.27|74.19/69.96|91.05/89.47|84.38/74.38 |
> |GPRGNN-AT|86.10/81.92|74.03/71.32|90.61/**90.49**|83.11/81.43|
> |SFT-Auto|83.27/**82.59**|74.56/**74.24**|94.74/92.29|86.40/**84.05**|
>
> **Inductive Text Attack (GPT 40%)**
>
> |Method|Cora|CiteSeer|PubMed|WikiCS|
> |-|-|-|-|-|
> |GPRGNN|86.29/66.30|74.19/41.80|91.05/67.43|84.38/48.40|
> |GPRGNN-AT|86.10/65.68|74.03/39.71|90.61/66.78|83.11/47.20|
> |SFT-Auto|83.27/76.75|74.56/**65.36**|94.74/77.86|86.40/77.83|
>
> *Format: Clean/Attacked. Bold: most robust.*
>
> **Key finding:** Adversarial training significantly improves structural robustness (81.92% vs 79.27% on Cora) but **degrades text robustness** (39.71% vs 41.80% on CiteSeer). This validates our core observation that existing defenses exhibit inherent text-structure trade-offs.
>
> ---
>
> ### **W4/W7: Perturbation Rate Analysis**
>
> Our perturbation rate exploration was limited by the combinatorial experimental space: (structure/text) × (GNN+LLM variants) × (encoders) × (evasion/poisoning) × (10 datasets).
>
> For the 80% poisoning rate: early experiments at 40% showed <1% accuracy drops for GCN-family models under transfer attacks—too subtle for meaningful comparison. We opted for higher rates to ensure observable effects. Note that in semi-supervised settings, 80% × 10% labeled nodes = 8% total training data, which remains reasonable.
>
> Appendix J provides results across varying perturbation rates. **We've now added line plots** for representative models with a detailed rationale in **Appendix K**.
>
> ---
>
> ### **W5: Paper Structure**
>
> SFT-neighbor builds on Wang et al.'s "Exploring graph tasks with pure LLM"—it's an existing baseline. Due to space constraints, detailed prompts and setup are left to the Appendix, like other baselines. For SFT-auto, we provide details about how it is set up.
>
> ---
>
> ### **Q4: Text Sources**
>
> TAG datasets have open-source text available (e.g., Cora/CiteSeer from github.com/CurryTang/Graph-LLM). Full data sources are detailed in Appendix C.
>
> ---
>
> ### **Q2 & Minor Issues**
>
> We adopted "Mettack" following DeepRobust and GNNGuard. We also noticed that the official codebase uses variants "Meta-attack" and "metattack".
> Could you please advise which variant is better: "Metaattack", "Meta-attack", or "Metattack"?
>
> Other issues are fixed in the revised PDF.
>
> ---
>
> We sincerely thank you for your valuable feedback, which has significantly improved our paper. We look forward to your response and hope these revisions address your concerns.

---

### Official Review · Reviewer_UTSY · 2025-10-31

**Soundness:** 3
**Presentation:** 4
**Contribution:** 3
**Rating:** 6
**Confidence:** 4

**Summary:**

This paper investigates the robustness of text-attributed graph (TAG) learning models against textual, structural, and hybrid adversarial attacks. It conducts a large-scale empirical study comparing GNNs, RGNNs, and GraphLLMs, and introduces SFT-auto, an LLM-based defense that automatically detects and recovers perturbed nodes. The results show that SFT-auto achieves more balanced robustness across attack types than prior methods, though the approach is primarily heuristic and lacks strong theoretical grounding.

**Strengths:**

1. The paper includes extensive baselines and experiments.
2. It offers novel insights into robustness in text-attributed graph learning.
3. The topic is highly relevant and timely.

**Weaknesses:**

1. The experiments use a fixed perturbation rate; it would be more informative to select a few representative models and show robustness across different perturbation rates.
2. Although effective, the strategies in the SFT-auto pipeline are not very novel.
3. The study mainly reports average ranks across datasets rather than raw accuracy or significance tests. This approach may obscure real performance gaps and overstate the robustness of SFT-auto.

**Questions:**

1. How is the perturbation for textual attacks defined within a single node? The proportion of perturbed nodes (40%–80%) seems quite large. My central question is: how do the authors justify that these perturbations are not noticeable?
2. Could the authors show the performance degradation of several representative models under varying perturbation rates for textual and combined textual–structural attacks?
3. Could the authors provide more detailed ablations on the inference process of SFT-auto to verify the contribution of each stage?

---

> ### Author Response · Authors · 2025-11-17
> **Response to Reviewer UTSY**
>
> We sincerely thank the reviewer for the thorough evaluation and constructive feedback.
>
> ---
>
> ### **W1 & Q2: Fixed Perturbation Rates**
>
> We provide **comprehensive multi-rate results in Appendix F & J.3**: structural attacks at rates 0.10 (Tables 46, 48) and 0.20 (Tables 27-30); textual attacks at rates 0.20 (Tables 47, 49) and 0.40 (Tables 34-36); combined attacks (WTGIA) at rates 0.20 and 0.40 (Tables 10-11). **Key findings remain consistent across all rates**.
>
> We chose higher rates following our principle (Section 2.2): "*weak perturbations fail to differentiate defense models*." At lower rates (e.g., 20-40%), GNNs show minimal degradation—GCN maintains 83-85% accuracy even at 80% perturbation (Table 37). Without sufficient attack strength, rankings become dominated by clean accuracy rather than robustness. Note that 80% of training nodes = **only ~8% of total nodes** in our semi-supervised split (10% training set).
>
> ---
>
> ### **W2: Limited Novelty of SFT-auto**
>
> SFT-auto's novelty stems from three fundamental contributions:
>
> **1. First to identify the text-structure robustness trade-off** (Section 4, Figures 2-4): This previously unknown phenomenon affects **all existing methods**.
>
> **2. Demonstrating failure of naive adaptations** (Section 5.1, Figure 5): Direct application of RGNN techniques (noise injection, similarity filtering) to GraphLLMs **fails to resolve the trade-off**—all variants remain polarized.
>
> **3. First effective solution**:
> - Prior LLM-based TAG defenses focus solely on structure purification [Zhang et al., 2025b; Guo et al., 2024b], offering no guarantee for mitigating textual attacks
> - SFT-auto achieves **unified multi-modal defense** within a single model, **the only method with balanced robustness** where all 15+ baselines fail (Figure 4)
> - As a pioneering work, SFT-auto **establishes LLM-based approaches as viable for robust TAG learning** and demonstrates their advantages over GNNs (Section 5.3)
>
> ---
>
> ### **W3: Reporting Average Ranks vs. Raw Accuracy**
>
> We display **raw accuracies** in Appendix J (20+ tables, 10 datasets × 4+ settings) due to space constraints. In the main paper, we report **both** performance rank (absolute) and drop rank (relative) to provide complementary perspectives. Besides, rankings offer two advantages:
>
> 1. **Fairness**: Methods have different scalability (ProGNN/RUNG fail on large graphs). Rankings only consider datasets where methods run successfully, while averaging raw accuracy unfairly penalizes methods with limited scalability.
> 2. **Interpretability**: Clean accuracy varies dramatically (90%+ on PubMed vs. 65% on Instagram). Rankings normalize this variation to reveal consistent trends.
>
> In our response to Reviewer jx3n, we include results comparing SFT-auto with other newly added baselines in terms of performance. SFT-auto achieves the best regarding the trade-off.
>
> ---
>
> ### **Q1: Noticeability of Textual Perturbations**
>
> **Our attack design philosophy**: We prioritize **rigorous evaluation** over imperceptibility. The 40/80% perturbation rate ensures sufficient attack strength to differentiate defenses. At lower rates, GNNs show minimal degradation, making evaluation indistinguishable from a clean accuracy comparison (see reply to W1 & Q2).
>
> **Content**: Our LLM-based attack (Section D.2) are required to generate **fluent, semantically coherent text** with constraints on word count (*keep similar length*) and topic consistency (*"Available classes"*), modeling realistic adversarial content injection. This constraint ensures better unnoticeability than gradient-based methods like textfooler, following WTGIA.
>
> **More realistic settings**: We additionally evaluate **WTGIA** (Appendix F, Tables 10-11), a graph injection attack that aligns with practical settings.
>
> ---
>
> ### **Q3: Detailed Ablations on SFT-auto Inference**
>
> We provide comprehensive ablations from multiple perspectives:
>
> **Stage-wise contributions** (Section 5.1, Figure 5):
> - **Without text recovery** (SFT-simf): Achieves only **40-70%** accuracy against textual attacks
> - **Without structure filtering** (SFT-neighbor): Suffers significant degradation under structural attacks (Figures 3 & 4)
> - **Combined pipeline** (SFT-auto): Achieves balanced robustness superior to all individual strategies (Figure 4)
>
> **Backbone comparison** (Section 5.3, Figure 6):
> - **LLM vs. GNN**: Replacing LLM with GNN (AutoGCN) degrades detection from **60-80% to 5-12%** (Figure 6b) and robustness from **75-85% to 50-65%** (Figure 6a), demonstrating LLM's advantage in identifying and recovering from adversarial perturbations
>
> ---
>
> We hope these clarifications adequately address the reviewer's concerns. We would be happy to provide additional details if needed.

---

### Author Response · Authors · 2025-11-27
**General Response to all Reviewers**

Dear Reviewers and Area Chair,

We sincerely thank all reviewers for their dedicated time and constructive feedback. We are encouraged that the reviewers recognized the value of our work, specifically highlighting the **comprehensive nature of our empirical study** [Reviewer UTSY, Bb1t], the **novel insights regarding the text-structure robustness trade-off** [Reviewer UTSY, jx3n], and the **extensive evaluation across diverse datasets and settings** [Reviewer Bb1t, jx3n].

We have carefully addressed all questions and concerns in our individual responses and have updated the PDF accordingly. **All modifications are highlighted in blue in the revised pdf.** We summarize the key improvements below:

---

**1. Refinements and Corrections**

We have corrected the categorization of baselines (e.g., clarifying spatial vs. spectral methods) and updated **Table 1** to accurately reflect recent benchmarks. We have also revised the text to ensure precise terminology regarding "transfer" vs. "adaptive" attacks and "inductive" settings. Additionally, we have updated **Table 2** to include more defense methods for a more comprehensive baseline coverage.

**2. Expanded Experimental Analysis**

In the previous version, we included tabular results across varying perturbation rates in **Appendix J**. We further included illustrative line plots in **Appendix K** to visualize these trends. These results confirm that our findings on model rankings and trade-offs remain consistent regardless of attack intensity.

**3. New Baselines and Validations**

We evaluated strong baselines (GPRGNN-AT, GOOT-AT, GPR-GAE) as requested. The detailed tables for these results have been added to **Appendix K** in the revised PDF. These new results confirm that the text-structure trade-off persists even in adversarially trained models (e.g., GOOT-AT excels on text but degrades on structure), further validating our core discovery. We also provided detailed ablations demonstrating that SFT-auto is the only solution achieving balanced robustness where other naive adaptations fail, highlighting the necessity of LLM-based reasoning for text defense.

---

**Scope of Our Contribution**

We would like to emphasize the breadth and depth of our work through the following key aspects:

1. **Comprehensive TAG Robustness Evaluation.** Compared to existing studies summarized in Table 1, our framework is the first to unify the evaluation of GNNs, RGNNs, and GraphLLMs under a consistent threat model for TAGs.

2. **In-depth Analysis of Adaptive Attacks.** While our main experiments focus on transfer attacks, we provide extensive adaptive attack analysis in **Appendix H**. Notably, we investigate how embedding alignment affects attack transferability (**Appendix H.2**), revealing that attacks transfer more effectively within the same embedding family (e.g., MiniLM ↔ RoBERTa) and that mismatched embeddings can serve as an effective defense strategy.

3. **Understanding and Improving GNNGuard.** We conduct a thorough analysis of why GNNGuard excels with advanced text encoders (**Appendix G**), demonstrating that embedding quality fundamentally determines its filtering effectiveness. Based on these insights, we propose **Guardual** (**Appendix G.3**), a novel dual-threshold extension that eliminates hyperparameter tuning and achieves SOTA RGNN performance against structural evasion attacks.

4. **Discovery of the Text-Structure Trade-off.** Our systematic evaluation reveals the inherent text-structure robustness trade-off (**Section 4**), where models excelling against one attack type consistently struggle against the other. This novel finding has significant practical implications for robust TAG system design.

5. **SFT-auto: A Unified Defense Framework.** In **Section 5.1**, we extensively explore combining existing robust GNN techniques with GraphLLMs, demonstrating that naive adaptations (noise injection, similarity filtering) fail to overcome the trade-off. Our proposed **SFT-auto** (**Section 5.2**) is the first unified framework that leverages LLM reasoning for both attack detection and recovery without training separate models, achieving efficiency and balanced robustness simultaneously. Furthermore, our analysis in **Section 5.3** shows that GNN-based alternatives (AutoGCN) fail due to limited semantic understanding, encouraging future research to build upon our findings using LLM backbones for robust TAG learning.

---

We believe these revisions have substantially strengthened the paper and addressed the concerns raised during the initial review. We have made every effort to respond thoroughly to each point and hope the reviewers will find our responses satisfactory. We remain fully committed to addressing any remaining questions and would be deeply grateful for the opportunity to continue this constructive discussion.

Best regards,

The Authors

---

### Meta-Review · Area_Chair_92dQ · 2026-01-06

**Summary:**

The paper presents a comprehensive and timely empirical study of adversarial robustness in text-attributed graphs, evaluating GNNs, RGNNs, and GraphLLMs under a unified framework. All reviewers acknowledged the breadth of experiments and the novel identification of the text-structure robustness trade-off. While opinions differed on the novelty of the proposed method, the overall contribution extends beyond a single defense and includes a large-scale evaluation framework, systematic robustness insights, and extensive experimental evidence. After rebuttal, most concerns were addressed, supporting acceptance.

**Reviewer Concerns:**

Addressed concerns:
Technical inaccuracies and baseline categorization issues were corrected. Additional baselines were added. Multiple perturbation experiments were provided.The focus on transfer attacks was clarified, and adaptive attack analyses were added. Additional ablations clarified the components and behavior of SFT-auto.

Remaining concerns:
The novelty depth of SFT-auto. This concern is largely subjective and does not negate the broader empirical and analytical contributions of the work.

**Reviewer Scores:**

Reviewer UTSY (6 → 6 or 8): Concerns were addressed.

Reviewer Bb1t (4 → 6): Likely increased after the addition of adaptive attack analysis, new baselines, and clarifications.

Reviewer jx3n (2 → 4 or 6): While concerns about novelty may persist, the correction of technical issues and expanded analysis would reasonably warrant a higher score.

---

### Decision · Program_Chairs · 2026-01-26

Accept (Poster)